# Empirical Guidelines for Deploying LLMs onto Resource-constrained Edge Devices

## Abstract

The scaling laws have become the de facto guidelines for designing large language models (LLMs), but they were studied under the assumption of unlimited computing resources for both training and inference. As LLMs are increasingly used as personalized intelligent assistants, their customization (i.e., learning through fine-tuning) and deployment onto resource-constrained edge devices will become more and more prevalent. An urgent but open question is how a resource-constrained computing environment would affect the design choices for a personalized LLM. We study this problem empirically in this work. In particular, we consider the tradeoffs among a number of key design factors and their intertwined impacts on learning efficiency and accuracy. The factors include the learning methods for LLM customization, the amount of personalized data used for learning customization, the types and sizes of LLMs, the compression methods of LLMs, the amount of time afforded to learn, and the difficulty levels of the target use cases. Through extensive experimentation and benchmarking, we draw a number of surprisingly insightful guidelines for deploying LLMs onto resource-constrained devices. For example, an optimal choice between parameter learning and RAG may vary depending on the difficulty of the downstream task, the longer fine-tuning time does not necessarily help the model, and a compressed LLM may be a better choice than an uncompressed LLM to learn from limited personalized data.

## 1 Introduction

The world has witnessed the growing interest in applying Large Language Models (LLMs) as a potential solution to personalized AI assistants (Qin et al., 2024c). This is particularly true when LLMs are deployed onto edge devices (edge LLMs) to meet the increasing needs of people's daily life assistance (Bevilacqua et al., 2023; Luo et al., 2023), personalized companionship (et al., 2023b; Qin et al., 2023), and real-time work assistance (et al., 2023a), where data privacy is of high priority Xu et al. (2023) and constant internet connections may not be possible (Lin et al., 2022). Some exemplar edge LLMs include *LLM on Nvidia IGX* (NVIDIA, 2024b), *Chat with RTX* (NVIDIA, 2024a), and *TinyChat* MIT (2024). edge LLMs can keep locally generated data from users and learn from that data locally. Such a high assurance of privacy coupled with LLMs' strong reasoning capabilities can induce even more user interaction with the personal assistant at a deeper personal level than otherwise (Hämäläinen et al., 2023).

The scaling law (Kaplan et al., 2020) has emerged as the standard for creating large language models (LLMs), but a key assumption behind it is the reliance on unlimited computing power for both training and inference. However, the design of edge LLMs is no easy feat (Lin et al., 2024c) because of the limited resources available on edge devices. It requires allocating limited resources to accommodate multiple needs. An ideal resource allocation strategy needs to balance many key, yet sometimes conflicting, factors, such as the the types and sizes of pre-trained LLMs, the learning methods and hyper-parameters for LLM customization, the amount of historical data used for learning personalization, the compression methods of LLMs, the amount of time afforded to learn, and the difficulty levels of the target user cases. For example, models can be selected from a wide range of candidates with different model structures and sizes; while learning methods include parameter-efficient fine-tuning (PEFT) and retrieval-augmented generation (RAG). A poorly designed edge LLM may render a poor user experience as it cannot learn well from user locally-generated content. As such, a pressing yet unresolved issue is how those constraints of limited computing resources

Figure 1: Overview of our empirical study, filling the missing part of scaling law cycled by pink dash-line in right coordinate graph. Each numbered annotation in red corresponds to an empirical guideline, detailed in the respective subsections of section 3.

influence the design decisions for personalized LLMs on edge devices, and what principles should guide the deployment of these edge LLMs. In this work, we empirically investigate this problem, focusing on the trade-offs between key design factors and their combined effects on learning efficiency and accuracy.

We formalize our empirical design guidelines for edge LLMs through extensive experimentation and benchmarking as shown in Figure 1, covering a wide range of possibilities for the key factors and their combinations. We draw a number of surprisingly insightful guidelines for deploying LLMs onto resource-constrained devices. These guidelines will be elaborated in six parts in Section 3. As a heads-up, a high-level summary of some interesting guidelines is as follows:

- The difficulty of the downstream task is a main factor for choosing the optimal edge LLM types and the learning method. Tasks that are either too easy or too hard tend to favor parameter learning, while medicore tasks tend to favor RAG.

- More historical user data for training is not always better. In most cases, the fine-tuning process does not necessarily need to use all the data and should stop early, especially when the model converges to some stable position (even at a very shallow local optimum).

- Among the three most popular model compression methods, distillation provides the most stable performance, quantization is less stable but has the highest peak performance, while pruning is not suitable for the edge LLMs.

To the best of our knowledge, this work is the first exploratory study that tries to address the full range of constraints on deploying LLMs on edge devices. By systematically examining the trade-offs among various factors, we offer guidelines and insights for the community and future research in this field. We hope that this work will both increase awareness of the limitations that LLMs will encounter in future edge deployments and shed light on the opportunities for future LLM designs.

## 2 PREPARATION FOR EVALUATION

In this section, we outline the settings for our evaluation. Due to space limitations, detailed content is provided in Appendix A. Specifically, Appendix A.1 emphasizes the constraints of edge devices. Following this, Appendix A.2 elaborates on how we pick and quantify resource constraints for comprehensive experiments. Appendix A.3 then describes the datasets used to evaluate edge LLM customization for personalization. With datasets and edge devices selected for experiments, it's crucial to choose a wide range of LLMs suitable for edge deployment. The selection rationale and a detailed list of chosen LLMs are presented in Appendix A.4. Beyond model selection, it's essential to consider how to compress an LLM to fit an edge device, effectively creating an edge LLM. Appendix A.5 provides details on the compression methods used in our empirical study. Having established the edge devices, models, datasets, and compression methods, the final critical component is the customization method. This method can enhance edge LLM personalization for individual users, as detailed in Appendix A.6. Lastly, Appendix A.7 covers the default experimental settings used in our study. In addition, we also include some background information about LLMs in Appendix D.

## 3 RESULTS ANALYSIS AND EMPIRICAL GUIDELINES

In this section, we present analyses of our experimental results and explain how these findings inform our empirical guidelines for edge LLMs. Additionally, we provide a perspective **remark** at the end of each subsection to highlight the significant finding and future potential research question.

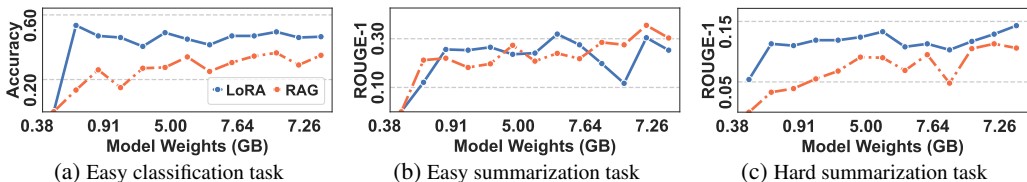

(a) Easy classification task      (b) Easy summarization task      (c) Hard summarization task

Figure 2: Performance comparisons on multiple models on tasks with different difficulties. The model size in the figures increases from left to right. The easy classification task refers to LaMP-2, the easy summarization task refers to LaMP-6, and the hard summarization task refers to LaMP-4.

Due to space limitations, some of the detailed experimental results are placed in Appendix B and Appendix C.

For the datasets we used, their difficulties can be ranked as LaMP-2 < LaMP-3 < LaMP-1 in classification task type, and LaMP-6 < LaMP-5 < LaMP-7 < LaMP-4 in generation task type. The details are included in Appendix A.3. In addition, we find that models often exceed human performance on classification tasks (Goh et al., 2020), whereas summarization tasks remain an open problem (Yadav et al., 2022). Thus, we argue that classification tasks are generally easier than summarization tasks. In the following analyses, we will use "difficulties" defined here loosely in multiple guidelines in the context of better choices.

### 3.1 OPTIMAL MODEL AND CUSTOMIZATION FOR LLMS ON THE EDGE

**Study target.** As illustrated in Figure 1, two primary questions need to be addressed within the constraints of edge device resources: 1) Which LLM is best suited for the edge device? and 2) What appropriate customization method should be applied to the chosen LLM? The performance of LLMs of various sizes, combined with different customization methods, can vary significantly depending on user tasks. These questions are particularly crucial for edge LLMs, as resource constraints necessitate inevitable trade-offs between model size and customization methods.

**Experiments and guidelines.** To investigate these questions, we first compiled a collection of over 30 LLMs with diverse architectures and sizes, as described in Section A.2. We then selected five widely used customization methods: LoRA(Hu et al., 2021), Prompt Tuning(Lester et al., 2021), Prefix Tuning(Li & Liang, 2021), IA3(Liu et al., 2022), and RAG(Lewis et al., 2020). We evaluated these LLMs on datasets designed for personalization tasks. It's worth noting that LoRA, Prompt Tuning, Prefix Tuning, and IA3 are categorized as Parameter-efficient Fine-tuning (PEFT) methods(Ding et al., 2023).

Table 1 presents results for 10 representative LLMs across the five customization methods on four tasks of increasing difficulty. The featured models include Pythia(Py)-2.8B, OPT-2.7B, Llama(Ll)-2-3B, StableLM(Sta)-3B, Gemma(Ge)-2B, Phi-2, Mistral(Mis)-7B quantized by GPTQ(G) (Frantar et al., 2022), OpenChat(OpCt)-G, Gemma-7B-G, and Sheared(S)-Llama-2.7B pruned. Our experiments reveal that LoRA consistently outperforms the other three PEFT methods, although it occasionally underperforms compared to RAG. Consequently, we focus our subsequent investigations on LoRA and RAG. Additional experimental results can be found in Appendix B.1.1.

Figure 2 showcases 13 models of varying sizes, ranging from 160M to 13B parameters, and their performance on three tasks of increasing difficulty. The models, arranged on the x-axis based on their weight sizes, include Pythia-160m, OPT-125m, OPT-350m, Pythia-410m, Phi-1.5, Gemma-2b-GPTQ, Phi-2, Sheared-Llama-2.7b, Llama2-3b-GPTQ, Phi-3, Llama2-7b-GPTQ, OpenChat-3.5-GPTQ, and Llama-13b-GPTQ. In our analysis, we selected LaMP-2 as the easy classification task, LaMP-6 as the easy summarization task, and LaMP-4 as the hard summarization task. Through our extensive experiments, we conclude that the optimal choice of model and customization method depends on the complexity of the downstream task. We provide the general guidelines below:

- For edge LLM customizations across different tasks, LoRA and RAG should be primarily considered. Other PEFT methods, including Prefix Tuning, IA3, and Prompt Tuning, can be resource-inefficient and less effective compared to LoRA and RAG.

- For simple classification tasks, the optimal choice should be a small LLM with LoRA. However, when an LLM is excessively small, like Pythia-70m or Pythia-160m shown in Appendix B.1.1, the

Table 1: Performance comparisons between parameter learning and RAG, across ten relatively big models on four datasets of increasing difficulty. PfixT stands for prefix tuning, and PmptT stands for prompt tuning.

| Models | | Py-2.8b | OPT-2.7b | Ll2-3b | Sta-3b | Ge-2b | Phi-2 | Mis-7b-G | OpCt-3.5-G | Ge-7b-G | S-Ll-2.7b-P |
|---|---|---|---|---|---|---|---|---|---|---|---|
| LaMP-2 | PfixT | 0.223 | 0.105 | 0.035 | 0.145 | 0.130 | 0.122 | 0.205 | 0.145 | 0.185 | 0.033 |
| | PmptT | 0.055 | 0.025 | 0.045 | 0.087 | 0.070 | 0.204 | 0.085 | 0.215 | 0.170 | 0.050 |
| | IA3 | 0.055 | 0.080 | 0.085 | 0.090 | 0.108 | 0.115 | 0.189 | 0.105 | 0.155 | 0.055 |
| | LoRA | **0.475** | **0.480** | **0.430** | **0.480** | **0.430** | **0.450** | **0.485** | **0.460** | **0.420** | **0.455** |
| | RAG | 0.320 | 0.110 | 0.295 | 0.245 | 0.205 | 0.340 | 0.375 | 0.290 | 0.365 | 0.035 |
| LaMP-3 | PfixT | 0.490 | 0.392 | 0.520 | 0.412 | 0.500 | 0.451 | 0.206 | 0.627 | 0.275 | 0.265 |
| | PmptT | 0.373 | 0.304 | 0.461 | 0.471 | 0.451 | 0.324 | 0.353 | 0.647 | 0.324 | 0.451 |
| | IA3 | 0.480 | 0.314 | 0.451 | 0.343 | 0.539 | 0.567 | 0.425 | 0.605 | 0.314 | 0.382 |
| | LoRA | **0.716** | 0.627 | **0.765** | **0.784** | **0.784** | **0.745** | **0.814** | 0.647 | **0.775** | **0.725** |
| | RAG | 0.716 | **0.696** | 0.461 | 0.667 | 0.765 | 0.627 | 0.755 | **0.814** | 0.480 | 0.578 |
| LaMP-6 | PfixT | 0.043 | 0.017 | 0.000 | 0.064 | 0.026 | 0.075 | 0.039 | 0.119 | 0.074 | 0.015 |
| | PmptT | 0.062 | 0.039 | 0.056 | 0.058 | 0.053 | 0.093 | 0.084 | 0.072 | 0.038 | 0.065 |
| | IA3 | 0.063 | 0.049 | 0.070 | 0.084 | 0.057 | 0.081 | 0.102 | 0.098 | 0.079 | 0.046 |
| | LoRA | **0.285** | **0.155** | **0.264** | **0.280** | 0.295 | **0.241** | 0.296 | 0.304 | 0.198 | **0.319** |
| | RAG | 0.186 | 0.134 | 0.253 | 0.256 | **0.299** | 0.208 | **0.327** | **0.355** | **0.223** | 0.240 |
| LaMP-7 | PfixT | 0.103 | 0.105 | 0.008 | 0.104 | 0.083 | 0.115 | 0.106 | 0.140 | 0.093 | 0.117 |
| | PmptT | 0.118 | 0.092 | 0.041 | 0.089 | 0.084 | 0.102 | 0.080 | 0.075 | 0.034 | 0.145 |
| | IA3 | 0.118 | 0.140 | 0.011 | 0.099 | 0.103 | 0.135 | 0.117 | 0.103 | 0.081 | 0.137 |
| | LoRA | 0.154 | **0.168** | **0.204** | 0.108 | 0.201 | **0.220** | **0.199** | **0.290** | 0.030 | 0.124 |
| | RAG | **0.170** | 0.156 | 0.149 | **0.166** | **0.270** | 0.197 | 0.181 | 0.231 | **0.126** | **0.188** |

model may be incapable of handling even simple classification tasks. It is noteworthy to avoid selecting such overly small models.

- As task difficulty increases, such as with complex classification tasks and simple summarization tasks, the choice should gradually shift to RAG with the strongest model. Here, the strongest models are (quantized) LLMs that excel at general benchmarks and fit within the RAM constraint.

- As the difficulty of downstream tasks further increases to challenging summarization tasks, RAG will not be sufficient, and LoRA with strong and quantized models will stand out as the optimal choice.

- When the difficulty of a task cannot be readily assessed, the Phi family combined with LoRA provides a safe option, offering decent performance across various scenarios.

***Remark 1: Why does RAG work best with moderately complex tasks?*** We hypothesize that this behavior stems from RAG's reliance on the base LLM's semantic understanding, which limits the potential for performance improvement. Given the emergent abilities of LLMs (Wei et al., 2022), where certain skills (particularly those required for solving more complex problems) only appear in larger models, it is expected that smaller LLMs perform well on simpler tasks but struggle with more complex ones. Furthermore, LLMs suited for edge deployment tend to be smaller in size, meaning that even 7B-parameter models may lack the capacity to address highly complex tasks effectively using RAG. In these cases, fine-tuning is more advantageous, as it teaches the models the semantic structure of the expected answers, making parameter learning more effective than RAG for such difficult tasks.

## 3.2 CHOICE OF LoRA STRATEGIES

**Study target.** In Section 3.1, we identified the generally superior performance of LoRA and RAG, and determined the situations where LoRA is more suitable than RAG. This section delves deeper into LoRA settings, aiming to establish efficient strategies for optimizing LoRA configurations across various edge LLM use cases. This investigation is particularly crucial for edge LLMs, given the limited resources available for conducting multiple experiments to determine optimal settings. In this context, we address three key questions. **First**, is it necessary to experiment with different LoRA settings to enhance model performance across various models and datasets? **Second**, is there an optimal range or a universal LoRA setting that is effective for all edge LLM applications? Answering these two questions can significantly benefit future edge LLM research and deployment by conserving time and resources in performance optimization. **Finally**, we aim to understand the factors that cause LoRA to behave differently in edge LLMs compared to LLMs hosted on cloud servers.

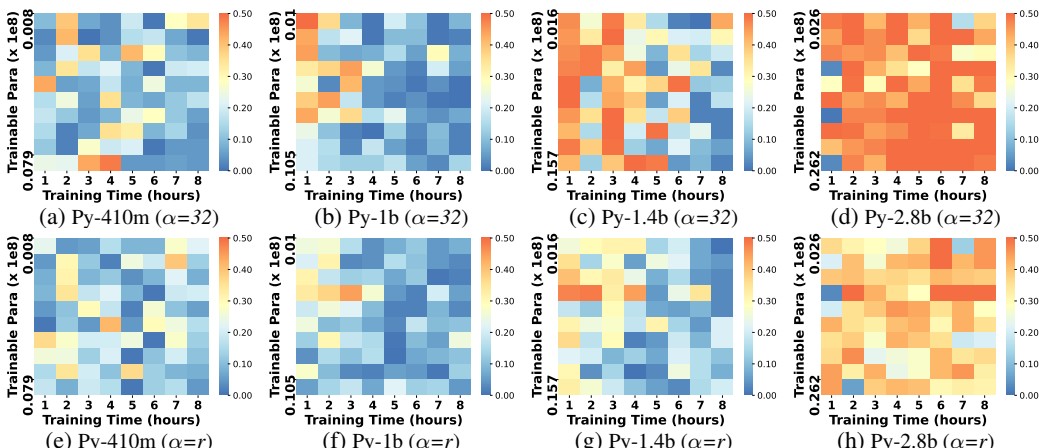

Figure 3: For (a)-(h): performance comparisons for Pythia(Py) models on LaMP-1. From (a) to (d), we set *alpha* ($\alpha$) to 32. From (e) to (h), we set $\alpha = rank$(r). More are in Appendix B.1.2 and Appendix B.1.3.

**Experiments and guidelines.** Among the hyper-parameters of LoRA, *rank* and *alpha* are most closely related to the number of trainable parameters, which directly impact resource usage. Moreover, these two hyper-parameters significantly influence edge LLM customization (Zhang et al., 2024a). While inappropriate combinations of *rank* and *alpha* can degrade edge LLM performance, identifying the optimal combination can be challenging. Different edge LLMs, operating under varying resource constraints, may require different optimal values for *rank* and *alpha*. Therefore, we investigate the impact of LoRA settings from two perspectives: the number of trainable parameters and the specific combinations of *rank* and *alpha*.

In Figure 3, we examine the performance of different-sized Pythia models given the value of rank equal 8, 16, 32, 40, 48, 56, 64, 72, and 80, under the training time from one to eight hours. Additionally in Figure 4, we examine the performance of StableLM-2-1.6b and the larger StableLM-3b on eight commonly used *rank* and *alpha* combinations, under the training time 1 to 8 hours. Our findings are as follows:

- Within the limited edge device resource, through experiments over various combinations of *alpha* and *rank*, varying the value of *alpha* benefits less than fixing the value of *alpha*.

- Increasing the training time might either have negligible LLM performance improvement or even lower the performance (A more detailed analysis on the training time will be provided in Section 3.3). This situation remains consistent even when the value of *rank* is increased.

- As shown in Figure 3 and Figure 4 and its supplemental results in Appendix B.1.2, Appendix B.1.3, and Appendix B.1.4, even in larger models like StableLM-3b and Pythia-2.8b deployed on edge devices with 10G or 16G RAM, increasing *rank* or *alpha* does not necessarily improve the model performance. Setting *alpha* and *rank* to (16, 16) or (16, 32) can work in most cases.

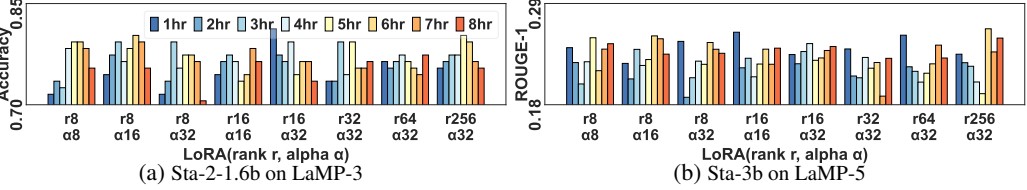

Figure 4: Performance comparisons for StableLM(Sta) models on LaMP-3 and LaMP-5 over eight combinations of alpha and rank. More results are in Appendix B.1.4

***Remark 2: Why is fixing alpha necessary but not rank for LoRA fine-tuning?*** We hypothesize that fixing $\alpha$ serves as an implicit mechanism to mitigate overfitting. As increasing $r$ enhances the model's adaptation to the training dataset, fixing the impact of LoRA on the original model with a constant $\frac{a}{r}$ could cause the model to focus excessively on the specifics of the training data, potentially leading to overfitting. However, we acknowledge that these findings may seem counterintuitive, and further research is needed to explore this topic, particularly in the context of resource-constrained fine-tuning.

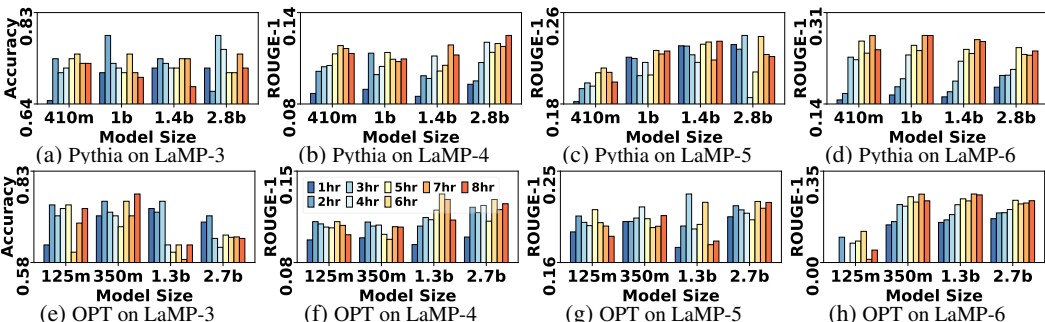

Figure 5: Performance comparisons on multiple sized Pythia and OPT models on different amounts of training data. More performance comparisons can be found in Appendix B.1.5.

## 3.3 TREND OF LoRA TRAINING TIME FOR EDGE LLMS

**Study target.** In Section 3.2, we identified appropriate LoRA settings for various user cases. Building upon this, it is crucial to determine whether extending training time under each setting yields additional benefits. For instance, if we can conclusively establish that longer training periods offer negligible performance improvements, we should consistently keep edge LLM training brief for resource efficiency—a critical consideration for edge devices. Notably, longer training times allow for the incorporation of more user history data in edge LLM training. Intuitively, due to their limited pre-trained knowledge stemming from smaller pre-trained weights, certain edge LLMs may exhibit learning behaviors from user history data that differ from those of larger edge LLMs with more extensive pre-trained knowledge. Considering these two aspects, studying the LoRA performance trend over extended training periods is of paramount importance, allowing us to explore the edge LLM's learning boundary.

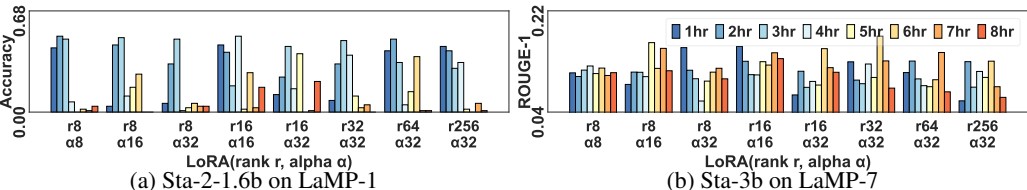

Figure 6: Performance comparisons for StableLM(Sta) models on LaMP-1 and LaMP-7 over eight combinations of alpha and rank. More results are in Appendix B.1.4

**Experiments and guidelines.** To conduct our investigations, we first quantify training time and correlate it with the amount of training data processed. For each model, we measure the number of data samples it can process within one hour. Detailed information can be found in Appendix A.2. Longer training times allow for more data samples to be processed by the edge LLM. After establishing this quantitative relationship between training time and data volume, we proceed to select edge LLMs for various edge devices. To minimize potential differences arising from varied model architectures, we choose four Pythia models and four OPT models of different sizes, deploying them on edge devices with RAM requirements ranging from 4G to 16G. Detailed specifications are provided in Appendix A.1. For LoRA settings, we adhere to the guidelines derived from Section 3.2, setting the *rank* to 8 and *alpha* to 32. We vary the training time from 1 to 8 hours, corresponding to typical idle periods for edge devices such as smartphones when users are asleep and the device is charging. The experimental results are presented in Figure 5 and Figure 6. Our findings are as follows:

- More training data does not necessarily mean better performance. Under most tasks, training with 3-4 hours is usually enough for customizing the LLMs towards downstream tasks on the A14 Bionic chip, and the time could be adjusted accordingly based on the edge device.

- While fine-tuning with LoRA is useful for almost all cases, increasing the training time only shows consistent improvement over time on the LaMP-2 dataset, the easiest task out of all seven, as shown in Figure 20 and Figure 21.

***Remark 3: Why does test accuracy not always improve after training for some time?*** We hypothesize that this behavior may result from two factors. First, since the evaluated tasks are based on personalized datasets, the training data might lack diversity, increasing the risk of overfitting. In such cases, the LLM may learn patterns tied to specific tokens rather than capturing the desired features

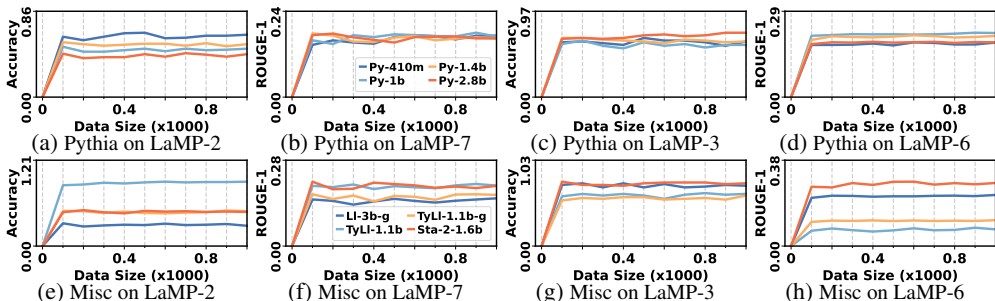

Figure 7: RAG performance comparison on four Pythia(Py) models and four miscellaneous(Misc) models including Llama(Ll)-3b-GPTQ(G), StableLM(Sta)-2-1.6B, TinyLlama(TyLl)-1.1B-G, and TyLl-1.1B across different sizes of user history data (**Data Size**). More in Appendix B.2.1.

of the broader target population. Second, due to resource constraints on edge devices, we simulate practical scenarios by limiting the training data size. Recent work (Muennighoff et al., 2024; Clark et al., 2022; Lin et al., 2024a) discusses the phenomenon of the "pre-scale law", which suggests that loss during LLM fine-tuning does not decrease linearly. Early in fine-tuning, test loss may not reduce quickly. Our observations in the experiments may align with the "pre-power law" stage, or even an earlier stage, where the LLM is learning an initial adaptation to the downstream task. In the first stage, the model improves as it learns the task representation. However, it then enters the pre-power-law stage, where performance oscillates. Finally, the model would reach the power-law stage, which requires at least 10K-100K samples, as noted by Lin et al. (2024a), a scale that is nearly impossible to achieve under the constraints of edge devices.

## 3.4 IMPACT OF USER HISTORY DATA VOLUME ON RAG PERFORMANCE

**Study target.** Unlike LoRA, RAG involves simpler hyper-parameters, primarily related to max inner product search (MIPS), its core functionality (Zhao et al., 2024). While LoRA's challenge lies in parameter determination for edge LLMs, RAG's difficulty stems from increasing stored user history data and search latency (Qin et al., 2024c). Cloud-hosted LLMs benefit from this data growth due to their strong reasoning capabilities. However, edge LLMs, with significantly smaller pre-trained knowledge bases and lower reasoning capabilities (Lin et al., 2024b), may not consistently gain the same advantages. This study aims to investigate whether expanding stored user history data consistently enhances RAG performance for edge LLMs or if there's a threshold beyond which data volume no longer improves performance, thus elucidating the relationship between data volume and RAG performance in edge LLMs.

**Experiments and guidelines.** We select 100 users and each user has up to 1000 samples of user history data. For the history data, we randomly maintain 0% to 100% of them, where 0% corresponds to the case where no RAG is used. RAG takes an LLM as its content generator. We examine four Pythia models with different sizes and four miscellaneous models including Llama-3b-v2-GPTQ, TinyLlama-1.1b, TinyLlama-1.1b-GPTQ, and StableLM-2-1.6b. We calculate the performance improvement brought by RAG using different amounts of user history data.

As shown in Figure 7, we observe that increasing the amount of user history data to 1000 samples does not significantly enhance RAG performance compared to using only 100 samples. The RAG performance based on eight models is quite consistent across all different sizes of user history data. Noted, in Figure 7c, increasing user history data can even lower the RAG performance based on Pythia-1b and Pythia-1.4b. The performance of RAG can be more related to the internal reasoning capabilities of the LLM rather than the amount of user history data.

***Remark 4: Why does using less history data have marginal effects on RAG performance?*** These results are not surprising, particularly when considering that personalized data may not be as diverse as general domain data. Additionally, connecting back to Section 3.1, we hypothesize that RAG elevates the LLM close to a stage just before the "pre-power law" by providing grounded context on the target's format and basic information. Such a hypothesis might be the reason why RAG only performs better than LoRA on tasks with moderate difficulty. For simple tasks, LoRA might be able to reach a stage beyond "pre-power law" (which is the upper bound of RAG). For complex tasks, RAG on edge LLMs cannot understand anything, whereas LoRA at least can learn some semantic structure, which helps the ROGUE metric.

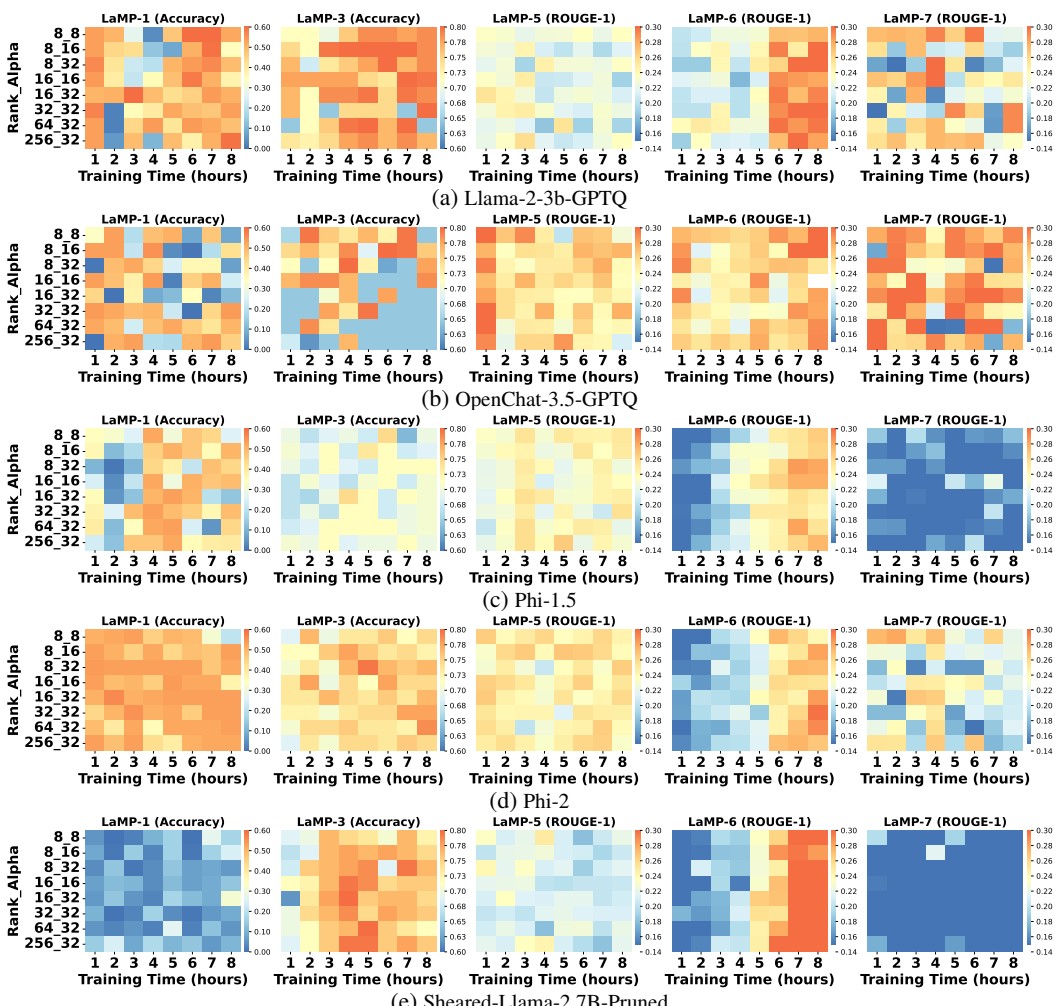

Figure 8: Performance comparisons of quantization, knowledge distillation and pruning models over eight commonly used LoRA (*rank_alpha*) settings. More results can be found in Appendix C.4.

### 3.5 COMPARISON OF MODEL COMPRESSION TECHNIQUES ON EDGE LLMS

**Study target.** Other than customization methods, the model itself is the other key component in our empirical study, shown in the right dashed rectangle in Figure 1. When deploying an LLM onto edge devices, it normally can be expected that the model needs to be compressed to fit the devices with various resource constraints. Hence, the critical question is: what is the best way to compress it? As such large-scale LLMs have demonstrated promising reasoning abilities when they are deployed on cloud servers, their compressed versions are also expected to demonstrate similar reasoning abilities when they are deployed on edge devices.

**Experiments and guidelines.** Among various model compression techniques, there are three main approaches including quantization, pruning, and knowledge distillation. While each of them may have the potential to conduct a decent compressed LLM performance, it remains unknown which one of the methods can be more promising and appropriate under the constraints of edge devices. We take the representative quantization method *GPTQ*, pruning method *structure pruning*, and knowledge distillation model *Phi* to make investigations. Our findings through the investigations are as follows (full experiments in Appendix C.1 to C.4):

- Using distilled models such as Phi is a safe option when the type and difficulty of the downstream task cannot be determined. Phi-1.5 shows robustness towards both classification and summarization tasks, where it generally has a "brighter" performance heatmap shown in Figure 8, whereas Phi-3 excels in classification tasks.

- Different compression techniques are good at different types of tasks. Summarization tasks require larger (and better) LLMs to achieve emergent ability, thus GPTQ models such as OpenChat-3.5-GPTQ work better than the other two compression techniques.

- Shearing is a very promising technique that preserves better performance, but they are less efficient at saving RAM compared to quantization. Thus, shearing is generally not preferred for edge LLMs where RAM capacity is a critical constraint.

***Remark 5: What are the foundations behind each compression technique?*** From the results, we observe that model distillation is the most stable method for fitting a large LLM on edge devices, although it rarely (if ever) stands out as the best option. In contrast, GPTQ is less stable but often achieves the best performance when combined with certain LoRA parameters and training data. Shearing and pruning, however, perform poorly in comparison to these two methods. It should be clear why sheared and pruned models are not ideal for edge devices. Pruning removes specific weights from the model, but since the models must be fine-tuned for downstream tasks, the missing weights are likely to be reintroduced during fine-tuning. As a result, pruned models offer no advantages in RAM usage and do not retain the semantic understanding abilities of larger models. Additionally, we hypothesize that the quality and memorization of pre-training datasets influence the fine-tuning of these models. It is well known that LLMs memorize parts of their training datasets (Yin et al., 2024; Carlini et al., 2023), but two uncertainties remain. First, we do not know the quality of the memorized data (i.e., do they memorize only high-quality/significant data, or do they memorize anything?). Second, we do not know **where** they memorize the data. It is possible that models trained on low-quantity pre-training data exhibit unstable behavior in downstream tasks, but further investigation is needed to confirm this.

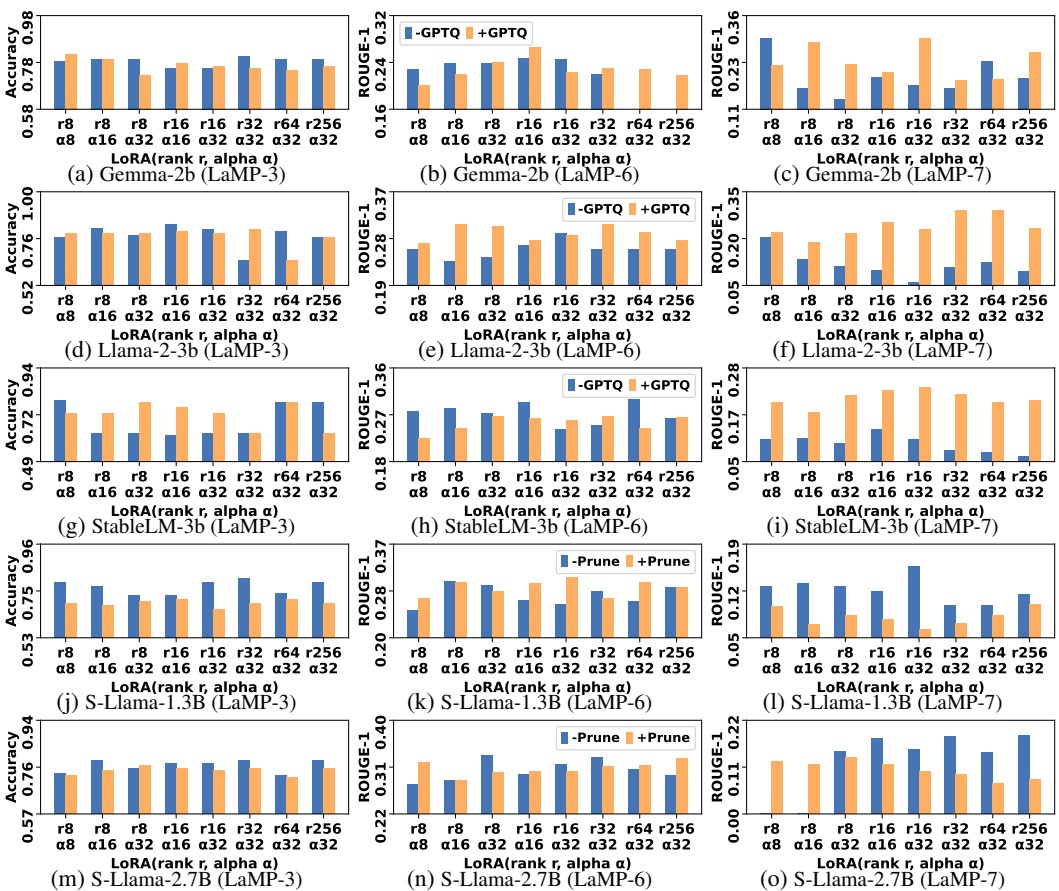

Figure 9: Performance comparisons of four quantized models, two pruned models with their corresponding original models on eight commonly used LoRA settings. More experiments are presented in Appendix C.5.

### 3.6 Comparison between Compressed and Uncompressed edge LLMs

**Study target.** After we compare the three compression techniques for their adaption on edge devices, we further need to investigate the impact of such compression: Consider an edge device with 16G RAM which is sufficient to host some uncompressed large-scale LLMs. While the compressed models can learn and infer faster than their larger original versions, will such benefits compromise the model's performance? Other than model performance, learning efficiency under compressed and uncompressed models is also important to consider.

**Experiments and guidelines:** Based on the experiments and their settings in Section 3.5, we concentrate on two groups of comparisons: between quantized and un-quantized models, and between pruned and un-pruned models. We select Gemma, Llama, and StableLM models. The results are shown in Figure 9. Through our investigations, our findings are as follows:

- While it is true that quantization will lead to some accuracy drops in most cases (Yin et al., 2024), it is noteworthy that with limited user history data for fine-tuning, the quantized model might perform better compared to unquantized counterparts on more challenging tasks.
- Different from quantization which can benefit edge LLMs, pruning as shown from Figure 9j to Figure 9o indicates a clear performance drop in most LoRA settings and various datasets. Hence, pruning in edge LLM fine-tuning is not recommended.
- A larger model is not always better. Llama3-8B is not performing well on many of the benchmarks, maybe because it is too big and not able to adapt to downstream tasks with limited user history data. Especially during classification tasks, we see that 2B to 3B models (quantized) are good enough. On the other hand, referring to Table 7, bigger models work better with summarization tasks that require more semantic understanding of the context.

***Remark 6: Why is a compressed model sometimes better?*** We hypothesize that such a phenomenon shares the same intuition with the Section 3.5. An LLM memorizes its pre-training data, and they are in the model's weights in a numerical form that we cannot interpret. Fine-tuning toward the downstream task is essentially the process of letting the LLM forget some of the pre-trained data and memorize data from the target domain. Thus, quantization removes a lot of the pre-training information that is stored in the less significant bits of the LLM and makes the fine-tuning faster.

## 4 Conclusion and Future Directions

In conclusion, we present an empirical study for deploying large language models onto edge devices with multiple resource constraints, and we provide guidelines for choosing the optimal strategy for the deployment. Through experiments, we show that the optimal choice of LLM and customization depends on the difficulty of the downstream task; during PEFT fine-tuning, the largest possible parameters and time are not the optimal settings; and compressed models are sometimes better than the original models on the edge due to their faster adaptation speed.

The insights provided by this study not only offer empirical guidelines for future deployment efforts but also highlight key avenues for further research into adapting LLMs in resource-constrained environments. In addition to these findings, we put forth several recommendations for the LLM community, drawn from our experimental results:

- Small LLMs could sometimes solve domain-specific problems better than larger models. The community, especially the industry, should consider private small LLMs as alternatives to larger models on many edge services.
- When fine-tuning small LLMs on the edge, the tradeoff between time and performance shall be carefully considered. Larger trainable parameters and more training time are not always the best. Different from the cloud LLMs where there are often no validation datasets, edge LLMs should employ a validation set to ensure the best fine-tuning performance.
- The RAG ability of edge LLMs is largely unexplored. Existing RAG frameworks heavily rely on the semantic understanding ability of LLMs, which might not exist on the edge.
- Model distillation is shown to be an effective solution that migrates the larger model's semantic understanding ability to edge devices, but more research shall be directed into this area.

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

# APPENDIX

# A  SUPPLEMENTS TO EVALUATIONS AND EXPERIMENTS

## A.1  EDGE DEVICE CONSTRAINTS

**Overview.** Clusters and cloud servers that provide computing-as-a-service (CaaS) consist of millions of computation units like GPU with nearly unlimited RAM, storage, and energy (Qin et al., 2024a). Common edge devices such as smartphones, on the other hand, are usually equipped with a multi-core CPU with limited resources. Emerging technologies are beginning to integrate GPUs into edge devices, exemplified by the Jetson Orin or phones powered by the Qualcomm Snapdragon 8 Gen, which offer computational powers nearly half that of a cluster-level GPU like the Nvidia RTX 2070. As computational power in edge devices increases, the feasibility of running edge LLMs is significantly enhanced.

**Edge Devices for Experiments.** In this paper, to prepare evaluations as we mentioned in Section 2, we selected some common edge devices with different RAM capacities and computation densities for evaluation as shown in Table 2. RAM is a critical resource for enabling edge LLM learning. The model weights must be loaded into RAM so that they can be updated and optimized to adapt to user needs. In other words, when choosing to fine-tune an edge LLM, we should consider not only the size of its weights but also the additional RAM usage due to parameter optimization and gradient calculation. Conversely, when opting to use RAG to facilitate edge LLM learning, the consideration shifts to only the weights size and data embedding size, as RAG requires storing all data embeddings in RAM to perform MIPS.

Table 2: Profile of five common edge devices evaluated in this paper

| Device | RAM (GB) |
|---|---|
| iPhone 12 (A14) | 4 |
| iPhone 15 (A16) | 6 |
| NVIDIA Jetson Orin Nano | 8 |
| Samsung Galaxy S24 Ultra | 10 |
| NVIDIA Jetson Orin NX | 16 |

## A.2  RUNNING LLMS ON EDGE DEVICES

In this paper, to correspond the evaluation preparation in Section 2, we select five commonly used edge devices with different RAM capacities, as shown in Table 2. Besides computational power, RAM is a critical resource for enabling edge LLM learning. We categorize the edge devices by RAM capacity, as it provides a more concise classification than using computational power. We use data processing capability to quantify the computational power of each edge device. By deploying LLMs to each edge device and determining the optimal processing time per data sample, we can calculate how many data samples each edge device can process per hour for all selected LLMs. The evaluation results are shown in Table 3.

In addition to the two primary resource constraints - computational power and RAM, some constraints such as temperature and energy consumption may also be of concern. However, these are considered secondary issues. Without addressing computational power and RAM requirements, an LLM cannot be deployed on edge devices at all. Once an LLM is deployed, temperature and energy consumption are correlated with computational power and RAM usage. In our work, we focus on the primary constraints. Additionally, we assume the edge devices are charging, connected to a power source, and training the deployed LLMs while users are not using these devices. Hence, we can minimize the impact of temperature and power constraints.

To efficiently conduct comprehensive evaluations, we use a high-performance NVIDIA A10 GPU to simulate the training time on edge devices. We set the corresponding number of data samples processed by the A10 GPU to represent the amount of time required for training on an edge device.

For example, the first entry in Table 3 indicates that 825 data samples can be processed when the Pythia-70m model is deployed on an edge device with 4GB RAM.

This approach allows us to run experiments to assess how Pythia-70m performs given different training times by simply assigning different numbers of data samples to the model under various settings, such as different LoRA hyperparameters. It also closely simulates practical scenarios where a user employs an edge device with a specific LLM and wants the edge LLM to learn from available data.

Table 3: Selected models with their initial model weights size and the training peak RAM. For the data size per hour, it indicates the estimated amount of data each model can learn or train from given one hour.

| Total RAM | Pretrained LLM (ID from Hugging Face) | Training Peak memory (GB) | Weights Size (GB) | Data Samples per Hour |
|---|---|---|---|---|
| 4G | EleutherAI/pythia-70m | 1.15 | 0.17 | 825 |
| | EleutherAI/pythia-160m | 1.61 | 0.38 | 415 |
| | EleutherAI/pythia-410m | 2.69 | 0.91 | 215 |
| | facebook/opt-125m | 1.78 | 0.25 | 550 |
| | facebook/opt-350m | 2.25 | 0.66 | 320 |
| | TheBloke/TinyLlama-1.1B-Chat-v0.3-GPTQ | 1.72 | 0.77 | 110 |
| | TheBloke/open-llama-3b-v2-GPTQ | 3.39 | 2.09 | 95 |
| 6G | facebook/opt-1.3b | 5.68 | 2.63 | 245 |
| | TheBloke/stablelm-zephyr-3b-GPTQ | 3.17 | 1.84 | 85 |
| | TechxGenus/gemma-2b-GPTQ | 5.31 | 2.08 | 145 |
| | EleutherAI/pythia-1b | 4.98 | 2.09 | 250 |
| | TinyLlama/TinyLlama-1.1B-step-50K-105b | 5.32 | 2.20 | 165 |
| 8G | microsoft/phi-1_5 | 6.76 | 2.84 | 153 |
| | EleutherAI/pythia-1.4b | 6.68 | 2.93 | 182 |
| | TheBloke/Llama-2-7B-Chat-GPTQ | 6.07 | 3.90 | 82 |
| | TheBloke/Mistral-7B-v0.1-GPTQ | 6.61 | 4.16 | 75 |
| | TheBloke/Synthia-7B-v1.3-GPTQ | 6.61 | 4.16 | 75 |
| | TheBloke/openchat_3.5-GPTQ | 6.16 | 4.16 | 76 |
| 10G | princeton-nlp/Sheared-LLaMA-1.3B-Pruned | 6.89 | 2.69 | 222 |
| | stabilityai/stablelm-2-1_6b | 8.12 | 3.29 | 156 |
| | TechxGenus/Meta-Llama-3-8B-GPTQ | 8.59 | 5.74 | 85 |
| | princeton-nlp/Sheared-LLaMA-1.3B | 6.89 | 5.38 | 222 |
| | facebook/opt-2.7b | 6.89 | 5.30 | 120 |
| | TechxGenus/gemma-7b-GPTQ | 9.45 | 7.18 | 71 |
| | TheBloke/Llama-2-13B-GPTQ | 9.95 | 7.26 | 50 |
| 16G | microsoft/phi-2 | 11.97 | 5.00 | 94 |
| | microsoft/Phi-3-mini-4k-instruct | 13.57 | 7.64 | 84 |
| | EleutherAI/pythia-2.8b | 12.97 | 5.68 | 106 |
| | google/gemma-2b | 13.45 | 4.95 | 132 |
| | princeton-nlp/Sheared-LLaMA-2.7B-Pruned | 12.26 | 5.40 | 89 |
| | stabilityai/stablelm-3b-4e1t | 12.61 | 5.59 | 100 |
| | openlm-research/open_llama_3b_v2 | 15.11 | 6.85 | 89 |
| | princeton-nlp/Sheared-LLaMA-2.7B | 12.26 | 9.95 | 89 |
| | meta-llama/Meta-Llama-3-8B* | 23.00 | 16.07 | 50 |

* for comparison with quantized Llama-3-8B

## A.3 Datasets

### A.3.1 Background

To prepare for evaluation in Section 2, it is crucial to utilize appropriate datasets. The edge LLM is primarily employed for handling prompts made by a single user. Consequently, user-specific datasets are necessary. Such datasets should feature data that is correlated and includes user-specific information.

We have selected the LaMP datasets (Salemi et al., 2023) to evaluate our edge LLM. In LaMP, each user has history data as shown in Table 5. The history data for each user consists of numerous utterances that include a label and textual content. The edge LLM will primarily learn from this historical data. The user prompts in the LaMP datasets are presented in Table 6. A well-trained edge LLM should be capable of accurately responding to the prompts using the history data.

The LaMP datasets consist of **seven datasets**. The first three datasets involve text classification, including LaMP-1 for citation identification, LaMP-2 for movie tagging, and LaMP-3 for product rating. The remaining four datasets pertain to text generation, including LaMP-4 for news headline generation, LaMP-5 for scholarly title generation, LaMP-6 for email subject generation, and LaMP-7 for tweet paraphrasing.

### A.3.2 Data Preparation

As we mentioned in Section 2, we prepare the data for our edge LLM evaluation from two perspectives: user-level and task-level.

**User-level**:

Each dataset contains many users, and each user has many samples of user history data and user query data. One sample of user history data and user query data are shown in Table 5 and in Table 6. In terms of generalization, we randomly select 100 users and calculate the average performance of edge LLMs over the 100 users in our evaluation.

**Task-level**:

When evaluating various edge LLMs across multiple datasets (tasks), it is important to account for the latent factor of task difficulty, which can influence the evaluation results (Ge et al., 2024). Within a single dataset, we assume that utterances exhibit similar levels of difficulty (Balyan et al., 2020). However, difficulty can vary significantly across different datasets, making it a critical consideration for accurate assessment. Previous research emphasizes the importance of considering task difficulty in evaluating machine learning models, as this factor can substantially impact performance comparisons across tasks and models. In particular, zero-shot learning has been used as an effective tool to estimate the inherent difficulty of datasets on strong pre-trained models, allowing for more refined assessments (Raina et al., 2024). To this end, we evaluate the performance of models such as GPT-4 (Achiam et al., 2023), Claude 3 Opus (Kevian et al., 2024), Gemini 1.0 Pro (Team et al., 2023), and Llama 3 70B (Meta, 2023) on selected test data from each dataset, with the results summarized in Table 4.

For classification tasks, where the number of choices varies across tasks, we adopt a more equitable assessment method inspired by (De Diego et al., 2022; Chen et al., 2020), which normalizes accuracy relative to human expert performance. Since obtaining human performance benchmarks on these datasets was not feasible, we instead normalized accuracy by dividing the LLMs' achieved accuracy by the accuracy expected from random guessing, a metric we refer to as "normalized accuracy." This is mathematically expressed as $\frac{\text{accuracy}}{1/\text{No. choices}}$. This approach allows us to address task difficulty disparities across datasets, as normalizing the model's accuracy against random guessing helps isolate performance differences related to the models themselves rather than the intrinsic difficulty of the task. Similar methods for task difficulty normalization have been discussed in literature (Yin et al., 2024), highlighting the importance of using a baseline such as random guessing or human experts when comparing performance across datasets of varying complexities.

For summarization tasks, normalized accuracy is equivalent to the raw accuracy, since random guessing does not apply in such contexts. This metric allows us to gauge task difficulty, where a higher normalized accuracy reflects a less challenging task.

Table 4: Difficulty (weighted average) comparisons of different datasets. The mean of four cloud-based LLMs' zero-shot performance can be used as an evaluator of task difficulty.

| Dataset | GPT-4 | Claude 3 Opus | Gemini 1.0 Pro | Llama 3 70B | Average | **Normalized Accuracy** | Task Type |
|---|---|---|---|---|---|---|---|
| LaMP-1 (Accuracy) | 0.539 | 0.539 | 0.490 | 0.422 | 0.4975 | 0.995 | Classification |
| LaMP-2 (Accuracy) | 0.355 | 0.320 | 0.300 | 0.400 | 0.3438 | 5.157 | Classification |
| LaMP-3 (Accuracy) | 0.667 | 0.657 | 0.510 | 0.755 | 0.6472 | 3.236 | Classification |
| LaMP-4 (ROUGE-1) | 0.143 | 0.171 | 0.139 | 0.131 | 0.1460 | 0.1460 | Generation |
| LaMP-5 (ROUGE-1) | 0.386 | 0.374 | 0.405 | 0.084 | 0.3123 | 0.3123 | Generation |
| LaMP-6 (ROUGE-1) | 0.351 | 0.356 | 0.405 | 0.278 | 0.3475 | 0.3475 | Generation |
| LaMP-7 (ROUGE-1) | 0.326 | 0.136 | 0.237 | 0.255 | 0.2385 | 0.2385 | Generation |

Our analysis yields normalized accuracy scores for the classification datasets as follows: LaMP-1 (0.995), LaMP-2 (5.157), and LaMP-3 (3.236). For the summarization datasets, the normalized accuracies are: LaMP-4 (0.1460), LaMP-5 (0.3123), LaMP-6 (0.3475), and LaMP-7 (0.2385). Detailed results for each dataset across the various cloud-based LLMs are presented in Table 4.

## A.4    SELECTED LLMS

In Section A.1 , we have explained the edge device constraints. As we mentioned in Section 2, we elaborate selected LLMs in this section. In Table 2, we categorize five RAM sizes based on the common edge devices. To run pre-trained LLMs under such RAM constraints, we first need to ensure the size of these LLMs can fit into their corresponding RAM. To satisfy 4G RAM, we pick five models from the family of Pythia (Biderman et al., 2023) and Llama (Touvron et al., 2023). To satisfy 6G RAM, we pick four models from the family of StableLM (Bellagente et al., 2024), Pythia (Biderman et al., 2023), and Gemma (Team et al., 2024). To satisfy 8G RAM, we pick six models from the family of Phi (Gunasekar et al., 2023), Pythia, Llama (Touvron et al., 2023), Mistral (Jiang et al., 2023), Synthia (Tissera, 2023), and OpenChat (Wang et al., 2023).To satisfy 10G RAM, we pick five models from the family of Llama, StableLM, and Gemma. To satisfy 16G RAM, we pick seven models from the family of Phi, Pythia, Gemma, Llama, and StableLM. The detailed model descriptions can be found in Table 3. For each model we also make profiling to find its peak RAM usage, showing as "Training Peak RAM" on Table  3. [1] The model with smaller training peak RAM can also run on larger RAMs.

## A.5    MODEL COMPRESSION

As we mentioned in Section 2, model compression is another component in our empirical study. The LLMs were originally designed and trained for cluster computing (Qin et al., 2024b). Their size can easily go beyond the size of edge device RAM size. Furthermore, the larger the model, the longer time it takes to train and make inferences (Hoffer et al., 2017). Model compression can help such LLMs deploy on edge devices (edge LLMs) and improve their training and inference efficiency. There are three common implementation methods for model compression: quantization, pruning, and distillation.

**Quantization** The default precision for model weights in LLMs is FP32 or FP16. By reducing the weight precision from 32 bits down to 3 or 4 bits (quantization), we can significantly decrease the model size and inference time due to simpler gradient computations. However, reducing the number of bits in weights can compromise their precision and, consequently, model performance. The primary goal of quantization is to reduce weight bits without degrading performance. The main strategy for LLM quantization involves quantizing and then calibrating the pre-trained model, a technique known as post-training quantization (PTQ) (Yao et al., 2024). Several implementations of PTQ for LLMs exist, including SmoothQuant (Xiao et al., 2023), AWQ (Lin et al., 2023), LLM-QAT (Liu et al., 2023), and GPTQ (Frantar et al., 2022).

---

[1]Phi-3 theoretically could fit in a 16G RAM during training, but typical optimizations such as HuggingFace's trainer require some extra RAM, which will cause overflow issues. Thus we do not include Phi-3 in Table 3 but we still use the model for some of the experiments.

For experimental consistency, in our experiments, we selected the widely-used GPTQ (W4A16), which has been applied to almost all LLMs and has proven effective. Furthermore, GPTQ has been specifically adapted for edge LLMs (Shen et al., 2024; Yi et al., 2023).

**Pruning** Different from quantization, pruning lightens the model by trimming off certain weights without lowering the model performance. Compared to quantization, pruning receives less attention in the beginning. One reason can be the high weight complexity of LLMs which makes it difficult to prune them while maintaining their performance. Furthermore, since the core modules of LLMs, the transformer blocks, make the structures of LLMs complex, pruning the LLMs can be more challenging than pruning on pure neural networks with simple structures. Within the existing pruning works including Wanda (Sun et al., 2023), LLM-Pruner (Ma et al., 2023), and Sheared-LLaMA (Xia et al., 2023), Sheared-LLaMA has demonstrated decent performance with published LLM weights.

To simplify the experiments and avoid potential issues with implementing Wanda or LLM-Pruner to prune LLMs, we use Sheared-LLaMA in our experiments.

**Distillation** Other than quantization and pruning, another method involves using a smaller model to emulate and learn from a larger model, a process known as distillation. Microsoft introduced a small language model (SLM) called Phi, which leverages synthetic datasets specifically created by GPT-3.5 to learn domains such as common sense reasoning, general knowledge, science, daily activities, and theory of mind (Gunasekar et al., 2023). This approach represents a novel departure from traditional large language models (LLMs), allowing SLMs to acquire dense knowledge from LLMs while maintaining a smaller size. Although other works such as Distilling Step-by-Step (Hsieh et al., 2023), MetaIE (Peng et al., 2024a), and MLFS (Kundu et al., 2024) have proposed distillation methods, they have not yielded a well-trained model comparable to Phi, which benefited from Microsoft's extensive resources and data. Therefore, we have chosen to use Phi in our experiments.

## A.6 METHODS FOR EDGE LLM CUSTOMIZATION

In Section 2, we also mention the customization. There are two methods for edge LLM customization: Parameter-Efficient Fine-Tuning (PEFT) (Hu et al., 2023) and Retrieval-Augmented Generation (RAG) (Chen et al., 2024). PEFT is an approach that improves edge LLM performance by tuning a small portion of the model's parameters. RAG, on the other hand, focuses on storing user history data and retrieving appropriate information to provide more context to user prompts. While both methods can generally enhance cloud-based LLM performance (Lakatos et al., 2024), their effectiveness can vary significantly depending on different settings and datasets. It is crucial to investigate whether similar variations in performance exist for edge LLMs.

**PEFT**: Compared to various PEFT implementations, low-rank adaption (LoRA) has demonstrated promising capabilities (Hu et al., 2021) in fine-tuning a wide range of LLMs and significantly improving their performance via updating a small portion of trainable parameters. The two hyperparameters *rank* and *alpha* can mostly impact the LoRA-tuning model performance (Lee et al., 2023). Hence, we set all other hyperparameters to default, as described in Appendix A.7, and explore a wide range of rank and alpha value combinations and their corresponding trainable parameters in the performance of the edge LLM.

**RAG**: RAG consists of a retriever that is commonly based on max inner product search (MIPS) and a generator which usually is an LLM. The retriever gets the appropriate query-relevant information from user history data to formalize the final prompt (Lewis et al., 2020). While this method does not consume resources to fine-tune the model, it requires saving all user history data embeddings for information retrieval. As user history data accumulates, the data embeddings can become a significant burden on the edge device (Qin et al., 2024c). In our experiments, we focus on the size of user history data and set all other hyperparameters to default, as shown in Appendix A.7.

Our experiments investigate how different settings in each method can impact model performance and study which method can outperform the other under certain circumstances.

## A.7 EXPERIMENTAL SETTINGS

In response to the mentioned experimental setting in Section 2, we provide detailed experimental settings in this section. For each LaMP dataset, we analyze 100 users, each containing up to 1000

documents in their user history data. Each document encapsulates a single piece of user information, as illustrated in Table 5. Unless specifically stated otherwise, we set the temperature to 0.1, top_p to 0.9, max_new_tokens to 100, and top_k to 10 for content generation by the LLM.

**Retrieval-Augmented Inference (RAG).** RAG operates with two main parts: a retriever that obtains the user-specific documents from his historical data using max inner product search (MIPS), and a generator backed by an LLM. This generator takes both the user query and the retrieved document as inputs to create an informative prompt and generate the corresponding content. For MIPS to function effectively, all history documents must be converted into embeddings via a sentence embedding model. In our experiments, we use *all-MiniLM-L6-v2* (Hugging Face). We select the highest-ranked data as the output in MIPS, setting the top_k parameter to 3.

**Parameter-Efficient Fine-Tuning (PEFT).** We selected LoRA (Hu et al., 2021), prefix tuning (Li & Liang, 2021), prompt tuning (Lester et al., 2021), and IA3 (Liu et al., 2022) as the Parameter-Efficient Fine-Tuning (PEFT) implementations in our study. For prompt tuning and prefix tuning, we set the virtual token size to 20. In the case of LoRA and IA3, we enabled fine-tuning of parameters for the *query*, *key*, and *value* layers. We configured the dropout rate for LoRA at *0.1*. The initial learning rate was set to *5e-4*, and we employed a linear learning rate scheduler to optimize it. For LoRA, we utilized the *AdamW* optimizer. Furthermore, we designated the fine-tuning task type as *CAUSAL_LM*. To explore a range of trainable parameter sizes, we established wide ranges for both rank and alpha, as these directly influence the number of trainable parameters. The rank was set to increment from *8* to *256*, while alpha ranged from *8* to *32*. This approach allowed us to investigate various trainable parameter configurations.

Table 5: One sample of user history data in each dataset

| Dataset | | User History Data |
|---------|---|-------------------|
| **LaMP-1** | [label] title
abstract | "DSP architectures: past, present and futures"
"As far as the future of communication is concerned, we have seen that there is great demand for audio and video data to complement text. Digital signal processing (DSP) is the science that enables traditionally analog audio and video signals to be processed digitally for transmission, storage, reproduction and manipulation. In this paper, we will explain the various DSP architectures and its silicon implementation. We will also discuss the state-of-the art and examine the issues pertaining to performance." |
| **LaMP-2** | [label] tag
description | "classic"
"Young Dorothy finds herself in a magical world where she makes friends with a lion, a scarecrow and a tin man as they make their way along the yellow brick road to talk with the Wizard and ask for the things they miss most in their lives. The Wicked Witch of the West is the only thing that could stop them." |
| **LaMP-3** | [label] score
text | "4"
"Amazing story of love that overcomes many obstacles. This book demonstrates that many of us are imprisoned by our views that have developed due to our upbringing, our culture, our environment and that it is possible to work through these problems of prejudice." |
| **LaMP-4** | [label] title
text | "Five Things Women Can Do Today to Move Past Divorce"
"I know it sounds trite, but now you have the freedom to be you and to let it reflect in your home, surroundings and daily activities. Embrace it, run with it and most of all enjoy it." |
| **LaMP-5** | [label] title
abstract | "Performability Studies of Hypercube Architectures"
"The authors propose a novel technique to study composite reliability and performance (performability) measures of hypercube systems using generalized stochastic Petri nets (GSPNs). This technique essentially consists of the following: (i) a GSPN reliability model; (ii) a GSPN performance model; and (iii) a way of combining the results from these two models. Models and performability results for an iPSC/2 hypercube system under the workload of concurrent matrix multiplication algorithm are presented." |
| **LaMP-6** | [label] title
abstract | "Get paid real $$$$$ to drive your own car!"
"You are receiving this exclusive promotion because you agreed to receive special offers from an emailYOUlike marketing partner. If you have received this email in error or would like to no longer receive these special offers, please follow the instructions at the end of the message. This message is brought to you by emailYOUlike. To find out more about emailYOUlike, visit http:www.emailyoulike.com or write us at 212 Technology Dr., Suite P, Irvine, CA 92602. If you would prefer not to receive future marketing messages from us, click here or visit http://www.emailyoulike.com/remove.asp, enter your email address, and click on the unsubscribe button. Only unsubscribe requests submitted to this page can be fulfilled. emailYOUlike cannot fulfill unsubscribe requests submitted elsewhere or to the email boxes of individuals." |
| **LaMP-7** | tweet | "Atleast Now, All fake political drama by parties (to get votes) in TN in the name of suffering Eelam Tamils will end, Thats it, Game over
HT Channel News: 25000 SL Tamils lie injured in NFZ w/o medical care and food. Thousands died in the final assault by SL army" |

Table 6: One sample of query data in each dataset

| Dataset | Prompt |
|---|---|
| LaMP-1 | "For an author who has written the paper with the title "An application-specific protocol architecture for wireless microsensor networks", which reference is related? Just answer with [1] or [2] without explanation. [1]: "End-to-end Internet packet dynamics" [2]: "Energy-Neutral Source-Channel Coding with Battery and Memory Size Constraints" |
| LaMP-2 | "Which tag does this movie relate to among the following tags? Just answer with the tag name without further explanation. tags: [sci-fi, based on a book, comedy, action, twist ending, dystopia, dark comedy, classic, psychology, fantasy, romance, thought-provoking, social commentary, violence, true story] description: A snobbish phonetics professor agrees to a wager that he can take a flower girl and make her presentable in high society." |
| LaMP-3 | "What is the score of the following review on a scale of 1 to 5? just answer with 1, 2, 3, 4, or 5 without further explanation. review: If You Were Me and Lived In...Germany: A Child's Introduction to Culture Around the World (If You Were Me and Lived) by Carole P. Roman, Kelsea Wienrenga (Illustrations) is a wonderful addition to the series. It's a delight to read and look at the illustrations! Plus this book provides so many facts about the culture and customs in Germany but not in a dry, boring way. This series is a terrific way to spark interest in the world for your child and maybe for you! With thanks to the author for my copy." |
| LaMP-4 | "Generate a headline for the following article: Being an ex wife was very unexpected. I went into it kicking and screaming. And drunk texting. Oh-and a little bit of stalking. To those going through it now I can tell you, you will survive." |
| LaMP-5 | "Generate a title for the following abstract of a paper: Web caching is the process in which web objects are temporarily stored to reduce bandwidth consumption, server load and latency. Web prefetching is the process of fetching web objects from the server before they are actually requested by the client. Integration of caching and prefetching can be very beneficial as the two techniques can support each other. By implementing this integrated scheme in a client-side proxy, the perceived latency can be reduced for not one but many users. In this paper, we propose a new integrated caching and prefetching policy called the WCP-CMA which makes use of a profit-driven caching policy that takes into account the periodicity and cyclic behaviour of the web access sequences for deriving prefetching rules. Our experimental results have shown a 10%-15% increase in the hit ratios of the cached objects and 5%-10% decrease in delay compared to the existing scheme." |
| LaMP-6 | "Generate a subject for the following email: You are receiving this exclusive promotion because you agreed to receive special offers from an emailYOUlike marketing partner. If you have received this email in error or would like to no longer receive these special offers, please follow the instructions at the end of the message. This message is brought to you by emailYOUlike. To find out more about emailYOUlike, visit http:www.emailyoulike.com or write us at 212 Technology Dr., Suite P, Irvine, CA 92618. If you would prefer not to receive future marketing messages from us, click here or visit http://www.emailyoulike.com/remove.asp, enter your email address, and click on the unsubscribe button. Only unsubscribe requests submitted to this page can be fulfilled. emailYOUlike cannot fulfill unsubscribe requests submitted elsewhere or to the email boxes of individuals." |
| LaMP-7 | "Paraphrase the following tweet without any explanation before or after it: I concur with @peyarili that there is animosity, and I believe that the Indian media is exacerbating the situation for the students. It seems as though the foolish media desires conflict with Australia." |

# B RESULTS OF EDGE LLM LEARNING

## B.1 EXPERIMENTAL RESULTS FOR PEFT

### B.1.1 EXPERIMENTS: EXAMINE RAG AND PEFT

The experiments contain a wide range of edge LLMs and compare their learning performance by PEFT and RAG. These experiments can correspond to Section 3.1.

Table 7: Performance comparisons of PEFT and RAG for selected LLMs. For PEFT we use the default experimental settings and let *rank* = 8, *alpha* =16. Additionally, we keep the training hours to 8 so the model can fully learn from the user history data. The PEFT performance compares with RAG. Their zero-shot learning performance can be found in Table 8.

| LLM | LaMP-1 | | LaMP-2 | | LaMP-3 | | LaMP-4 | | LaMP-5 | | LaMP-6 | | LaMP-7 | |
|---|---|---|---|---|---|---|---|---|---|---|---|---|---|---|
| | FT | RAG | FT | RAG | FT | RAG | FT | RAG | FT | RAG | FT | RAG | FT | RAG |
| Pythia-70m | 0.000 | 0.000 | 0.000 | 0.000 | 0.000 | 0.000 | 0.000 | 0.000 | 0.000 | 0.000 | 0.000 | 0.000 | 0.000 | 0.000 |
| Pythia-160m | 0.019 | 0.000 | 0.000 | 0.000 | 0.647 | 0.000 | 0.054 | 0.000 | 0.076 | 0.000 | 0.001 | 0.000 | 0.011 | 0.000 |
| Pythia-410m | 0.275 | 0.500 | 0.460 | 0.010 | 0.725 | 0.657 | 0.119 | 0.055 | 0.198 | 0.157 | 0.253 | 0.182 | 0.128 | 0.156 |
| Pythia-1b | 0.039 | 0.500 | 0.465 | 0.185 | 0.716 | 0.578 | 0.109 | 0.051 | 0.211 | 0.165 | 0.273 | 0.221 | 0.096 | 0.169 |
| Pythia-1.4b | 0.559 | 0.490 | 0.470 | 0.510 | 0.716 | 0.667 | 0.118 | 0.061 | 0.216 | 0.187 | 0.281 | 0.253 | 0.128 | 0.163 |
| Pythia-2.8b | 0.500 | 0.451 | 0.475 | 0.320 | 0.716 | 0.716 | 0.129 | 0.065 | 0.207 | 0.220 | 0.285 | 0.186 | 0.154 | 0.170 |
| opt-125m | 0.049 | 0.284 | 0.535 | 0.135 | 0.725 | 0.588 | 0.113 | 0.033 | 0.189 | 0.135 | 0.121 | 0.212 | 0.211 | 0.187 |
| opt-350m | 0.196 | 0.500 | 0.470 | 0.260 | 0.735 | 0.569 | 0.110 | 0.039 | 0.204 | 0.171 | 0.091 | 0.236 | 0.156 | 0.199 |
| opt-1.3b | 0.363 | 0.373 | 0.475 | 0.270 | 0.627 | 0.657 | 0.120 | 0.057 | 0.199 | 0.175 | 0.123 | 0.112 | 0.121 | 0.173 |
| opt-2.7b | 0.010 | 0.520 | 0.480 | 0.110 | 0.627 | 0.696 | 0.127 | 0.060 | 0.222 | 0.184 | 0.155 | 0.134 | 0.168 | 0.156 |
| TinyLlama-1.1b | 0.500 | 0.520 | 0.230 | 0.085 | 0.775 | 0.618 | 0.088 | 0.051 | 0.182 | 0.150 | 0.158 | 0.081 | 0.181 | 0.206 |
| Llama-v2-3b | 0.320 | 0.569 | 0.430 | 0.295 | 0.765 | 0.461 | 0.129 | 0.026 | 0.252 | 0.136 | 0.264 | 0.253 | 0.204 | 0.149 |
| Llama-v3-8b | 0.324 | 0.590 | 0.130 | 0.235 | 0.627 | 0.696 | 0.065 | 0.091 | 0.180 | 0.150 | 0.172 | 0.280 | 0.189 | 0.213 |
| StableLM-2-1.6b | 0.480 | 0.390 | 0.440 | 0.210 | 0.755 | 0.765 | 0.118 | 0.098 | 0.181 | 0.208 | 0.265 | 0.276 | 0.097 | 0.200 |
| StableLM-3b | 0.529 | 0.520 | 0.480 | 0.245 | 0.784 | 0.667 | 0.136 | 0.094 | 0.246 | 0.248 | 0.280 | 0.256 | 0.108 | 0.166 |
| Gemma-2b | 0.471 | 0.010 | 0.430 | 0.205 | 0.784 | 0.765 | 0.119 | 0.079 | 0.239 | 0.245 | 0.295 | 0.299 | 0.201 | 0.270 |
| Phi-1.5 | 0.255 | 0.451 | 0.405 | 0.270 | 0.696 | 0.422 | 0.119 | 0.068 | 0.252 | 0.203 | 0.265 | 0.197 | 0.181 | 0.270 |
| Phi-2 | 0.206 | 0.529 | 0.450 | 0.340 | 0.745 | 0.627 | 0.133 | 0.090 | 0.243 | 0.241 | 0.241 | 0.208 | 0.220 | 0.197 |
| Phi-3 | 0.471 | 0.333 | 0.470 | 0.345 | 0.784 | 0.647 | 0.103 | 0.048 | 0.211 | 0.242 | 0.198 | 0.285 | 0.199 | 0.158 |
| Llama-v2-3b-G[1] | 0.520 | 0.559 | 0.470 | 0.305 | 0.784 | 0.706 | 0.113 | 0.095 | 0.234 | 0.215 | 0.275 | 0.218 | 0.220 | 0.151 |
| StableLM-3b-G | 0.490 | 0.559 | 0.440 | 0.390 | 0.725 | 0.598 | 0.118 | 0.082 | 0.234 | 0.251 | 0.226 | 0.258 | 0.200 | 0.144 |
| TinyLlama-1.1B-G | 0.265 | 0.422 | 0.410 | 0.120 | 0.765 | 0.588 | 0.115 | 0.057 | 0.225 | 0.182 | 0.173 | 0.112 | 0.197 | 0.171 |
| Gemma-2b-G | 0.167 | 0.461 | 0.490 | 0.275 | 0.814 | 0.716 | 0.124 | 0.091 | 0.240 | 0.247 | 0.201 | 0.273 | 0.227 | 0.169 |
| Llama-v2-7b-G | 0.451 | 0.520 | 0.495 | 0.365 | 0.755 | 0.657 | 0.129 | 0.105 | 0.250 | 0.278 | 0.172 | 0.275 | 0.249 | 0.157 |
| Llama-v3-8b-G | 0.539 | 0.520 | 0.140 | 0.365 | 0.461 | 0.631 | 0.073 | 0.115 | 0.229 | 0.245 | 0.174 | 0.280 | 0.277 | 0.156 |
| Mistral-7b-G | 0.529 | 0.529 | 0.485 | 0.375 | 0.814 | 0.755 | 0.128 | 0.097 | 0.255 | 0.274 | 0.296 | 0.327 | 0.199 | 0.181 |
| OpenChat-3.5-G | 0.539 | 0.520 | 0.460 | 0.290 | 0.647 | 0.814 | 0.129 | 0.113 | 0.227 | 0.279 | 0.304 | 0.355 | 0.290 | 0.231 |
| Synthia-7b-G | 0.451 | 0.529 | 0.480 | 0.365 | 0.725 | 0.549 | 0.123 | 0.088 | 0.244 | 0.302 | 0.301 | 0.338 | 0.263 | 0.213 |
| Gemma-7b-G | 0.529 | 0.539 | 0.420 | 0.365 | 0.775 | 0.480 | 0.105 | 0.048 | 0.213 | 0.184 | 0.198 | 0.223 | 0.030 | 0.126 |
| Llama-13b-G | 0.529 | 0.569 | 0.465 | 0.350 | 0.804 | 0.657 | 0.143 | 0.106 | 0.259 | 0.264 | 0.253 | 0.303 | 0.121 | 0.138 |
| S[3]-Llama-1.3b-P[2] | 0.069 | 0.049 | 0.410 | 0.100 | 0.686 | 0.569 | 0.090 | 0.040 | 0.215 | 0.157 | 0.268 | 0.195 | 0.096 | 0.178 |
| S-Llama-1.3B | 0.461 | 0.471 | 0.510 | 0.255 | 0.784 | 0.559 | 0.113 | 0.064 | 0.205 | 0.197 | 0.247 | 0.234 | 0.126 | 0.183 |
| S-Llama-2.7B-P | 0.167 | 0.120 | 0.455 | 0.035 | 0.725 | 0.578 | 0.100 | 0.051 | 0.221 | 0.207 | 0.319 | 0.240 | 0.124 | 0.188 |
| S-Llama-2.7B | 0.294 | 0.529 | 0.415 | 0.250 | 0.775 | 0.627 | 0.108 | 0.069 | 0.216 | 0.242 | 0.218 | 0.260 | 0.234 | 0.181 |

---

[1]G represents GPTQ, the quantization technique

[2]P represents Pruned

[3]S represents Sheared

Table 8: Performance comparisons of zero-shot learning for all selected LLMs.

| LLM | LaMP-1 | LaMP-2 | LaMP-3 | LaMP-4 | LaMP-5 | LaMP-6 | LaMP-7 |
|---|---|---|---|---|---|---|---|
| Pythia-70m | 0.000 | 0.000 | 0.000 | 0.000 | 0.000 | 0.000 | 0.000 |
| Pythia-160m | 0.000 | 0.000 | 0.000 | 0.000 | 0.000 | 0.000 | 0.000 |
| Pythia-410m | 0.420 | 0.230 | 0.441 | 0.034 | 0.103 | 0.045 | 0.112 |
| Pythia-1b | 0.420 | 0.310 | 0.343 | 0.028 | 0.134 | 0.047 | 0.115 |
| Pythia-1.4b | 0.400 | 0.240 | 0.373 | 0.033 | 0.120 | 0.000 | 0.058 |
| Pythia-2.8b | 0.429 | 0.130 | 0.598 | 0.035 | 0.128 | 0.053 | 0.107 |
| opt-125m | 0.039 | 0.031 | 0.304 | 0.018 | 0.098 | 0.034 | 0.096 |
| opt-350m | 0.429 | 0.210 | 0.255 | 0.021 | 0.106 | 0.056 | 0.116 |
| opt-1.3b | 0.429 | 0.035 | 0.284 | 0.033 | 0.130 | 0.041 | 0.102 |
| opt-2.7b | 0.429 | 0.150 | 0.451 | 0.025 | 0.113 | 0.051 | 0.152 |
| TinyLlama-1.1b | 0.471 | 0.090 | 0.588 | 0.032 | 0.088 | 0.029 | 0.143 |
| Llama-v2-3b | 0.469 | 0.130 | 0.461 | 0.026 | 0.136 | 0.053 | 0.150 |
| Llama-v3-8b | 0.469 | 0.220 | 0.695 | 0.035 | 0.159 | 0.081 | 0.120 |
| StableLM-2-1.6b | 0.449 | 0.109 | 0.735 | 0.033 | 0.138 | 0.066 | 0.135 |
| StableLM-3b | 0.429 | 0.145 | 0.441 | 0.040 | 0.177 | 0.067 | 0.257 |
| Gemma-2b | 0.078 | 0.140 | 0.696 | 0.025 | 0.145 | 0.077 | 0.130 |
| Phi-1.5 | 0.500 | 0.155 | 0.382 | 0.023 | 0.094 | 0.056 | 0.062 |
| Phi-2 | 0.429 | 0.110 | 0.471 | 0.064 | 0.096 | 0.059 | 0.259 |
| Phi-3 | 0.441 | 0.175 | 0.549 | 0.049 | 0.108 | 0.104 | 0.098 |
| Llama-v2-3b-G[1] | 0.402 | 0.275 | 0.647 | 0.041 | 0.173 | 0.065 | 0.158 |
| StableLM-3b-G | 0.490 | 0.190 | 0.559 | 0.049 | 0.225 | 0.098 | 0.166 |
| TinyLlama-1.1B-G | 0.059 | 0.090 | 0.176 | 0.026 | 0.069 | 0.043 | 0.109 |
| Gemma-2b-G | 0.402 | 0.210 | 0.676 | 0.026 | 0.114 | 0.051 | 0.107 |
| Llama-v2-7b-G | 0.429 | 0.275 | 0.696 | 0.048 | 0.190 | 0.147 | 0.181 |
| Llama-v3-8b-G | 0.429 | 0.210 | 0.715 | 0.026 | 0.205 | 0.120 | 0.155 |
| Mistral-7b-G | 0.429 | 0.190 | 0.343 | 0.034 | 0.210 | 0.079 | 0.166 |
| OpenChat-3.5-G | 0.480 | 0.210 | 0.735 | 0.052 | 0.182 | 0.134 | 0.202 |
| Synthia-7b-G | 0.500 | 0.275 | 0.373 | 0.033 | 0.120 | 0.000 | 0.271 |
| Gemma-7b-G | 0.402 | 0.230 | 0.676 | 0.026 | 0.114 | 0.051 | 0.112 |
| Llama-13b-G | 0.429 | 0.155 | 0.696 | 0.027 | 0.252 | 0.068 | 0.174 |
| $S^3$-Llama-1.3b-P[2] | 0.294 | 0.113 | 0.255 | 0.028 | 0.111 | 0.038 | 0.176 |
| S-Llama-1.3B | 0.429 | 0.135 | 0.206 | 0.030 | 0.107 | 0.029 | 0.122 |
| S-Llama-2.7B-P | 0.010 | 0.120 | 0.422 | 0.027 | 0.128 | 0.043 | 0.124 |
| S-Llama-2.7B | 0.429 | 0.220 | 0.549 | 0.029 | 0.174 | 0.074 | 0.118 |

### B.1.2 EXPERIMENTS: TRAINABLE PARAMETERS BASED ON FIXED ALPHA AND VARYING RANK

The experiments for fixed value of alpha to 32 and changing value of *rank* from 8 to 80 have results shown in Figure 10, Figure 11, Figure 12, Figure 13 correspond to Section 3.2.

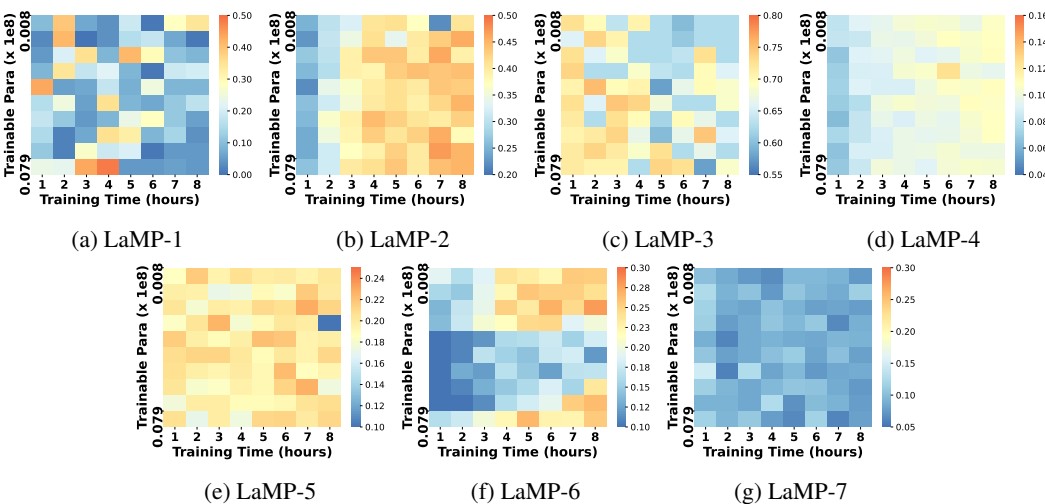

Figure 10: Performance heatmaps for Pythia-410m on dataset LaMP-1 to LaMP-7, given *alpha* = 32 and *rank* = 8, 16, 24, 32, 40, 48, 56, 72, 80. In each heatmap, we use the number of trainable parameters of different *rank* values as y-axis , and the training hours as the x-axis.

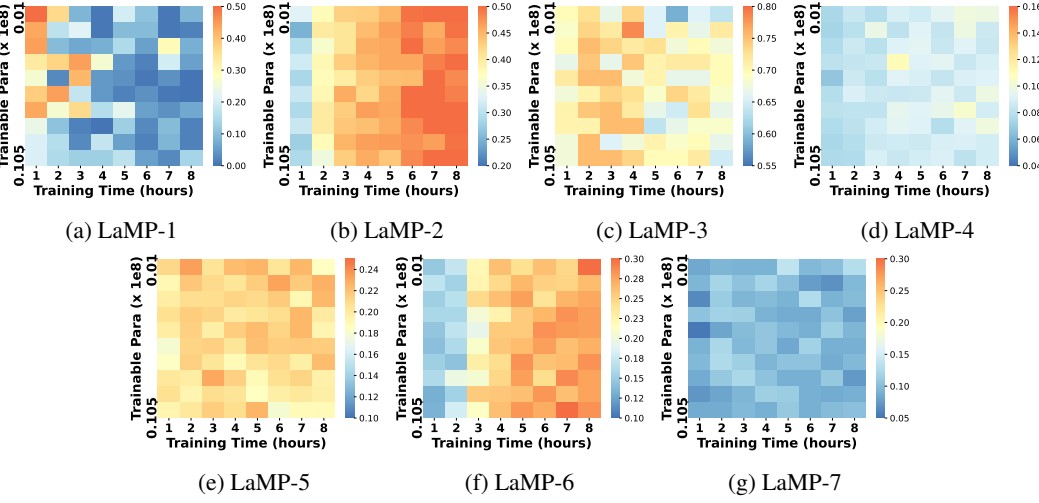

Figure 11: Performance heatmaps for Pythia-1b on dataset LaMP-1 to LaMP-7, given *alpha* = 32 and *rank* = 8, 16, 24, 32, 40, 48, 56, 72, 80. In each heatmap, we use the number of trainable parameters of different *rank* values as y-axis , and the training hours as the x-axis.

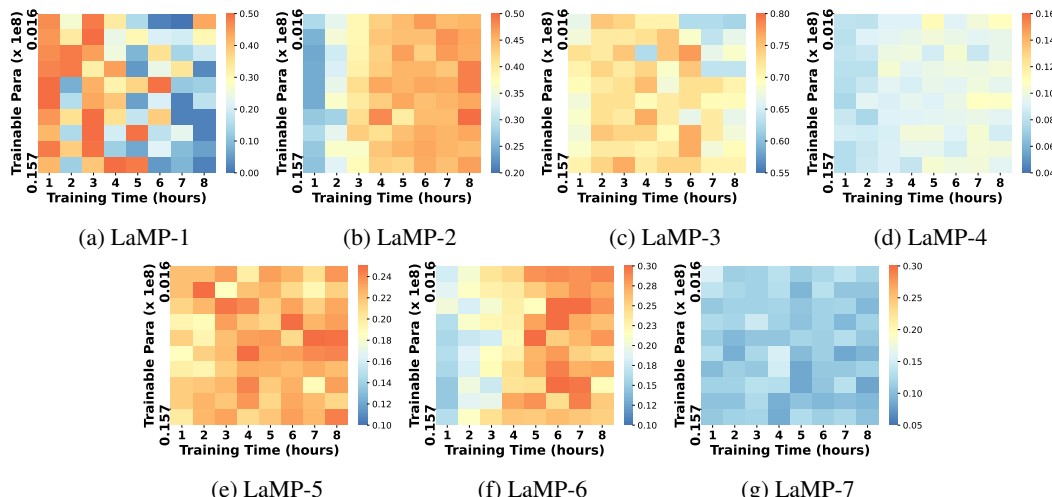

Figure 12: Performance heatmaps for Pythia-1.4b on dataset LaMP-1 to LaMP-7, given *alpha* = 32 and *rank* = 8, 16, 24, 32, 40, 48, 56, 72, 80. In each heatmap, we use the number of trainable parameters of different *rank* values as y-axis , and the training hours as the x-axis.

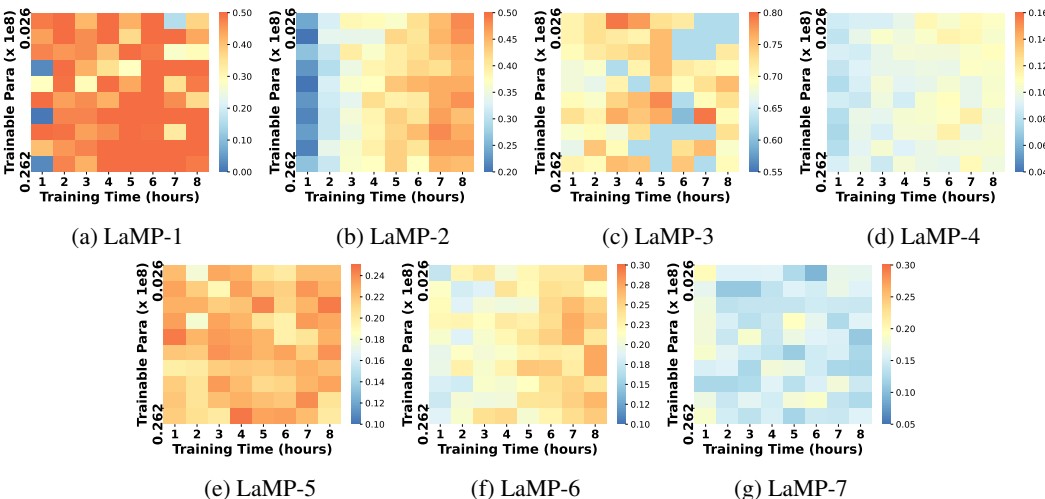

Figure 13: Performance heatmaps for Pythia-2.8b on dataset LaMP-1 to LaMP-7, given *alpha* = 32 and *rank* = 8, 16, 24, 32, 40, 48, 56, 72, 80. In each heatmap, we use the number of trainable parameters of different *rank* values as y-axis , and the training hours as the x-axis.

### B.1.3 EXPERIMENTS: TRAINABLE PARAMETERS BASED ON VARYING ALIGNED ALPHA AND RANK

The experiments for aligning alpha and rank and changing their values from 8 to 80 have results shown in Figure 14, Figure 15, Figure 16, Figure 17 correspond to Section 3.2.

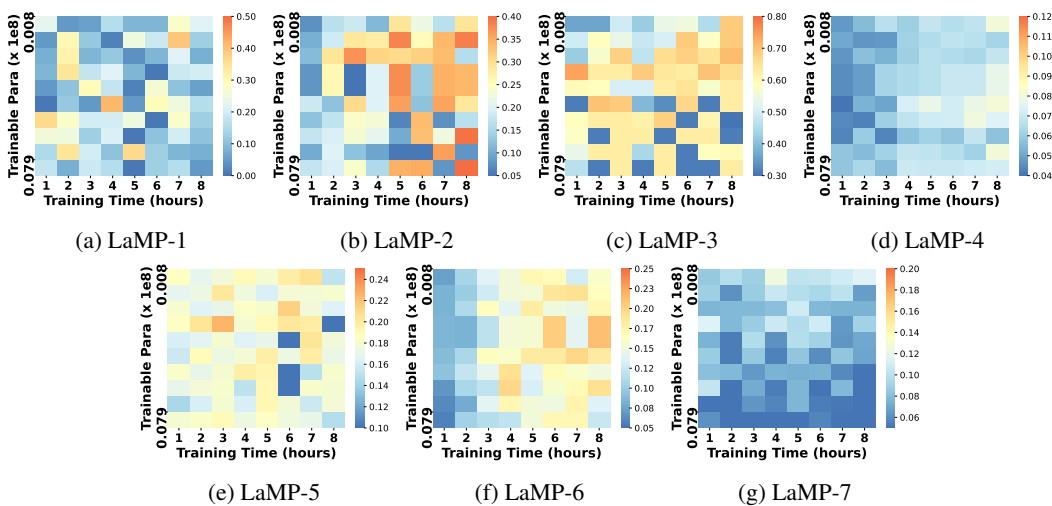

Figure 14: Performance heatmaps for Pythia-410m on dataset LaMP-1 to LaMP-7, given *alpha = rank* = 8, 16, 24, 32, 40, 48, 56, 72, 80. In each heatmap, we use the number of trainable parameters of different *rank* values as y-axis , and the training hours as the x-axis.

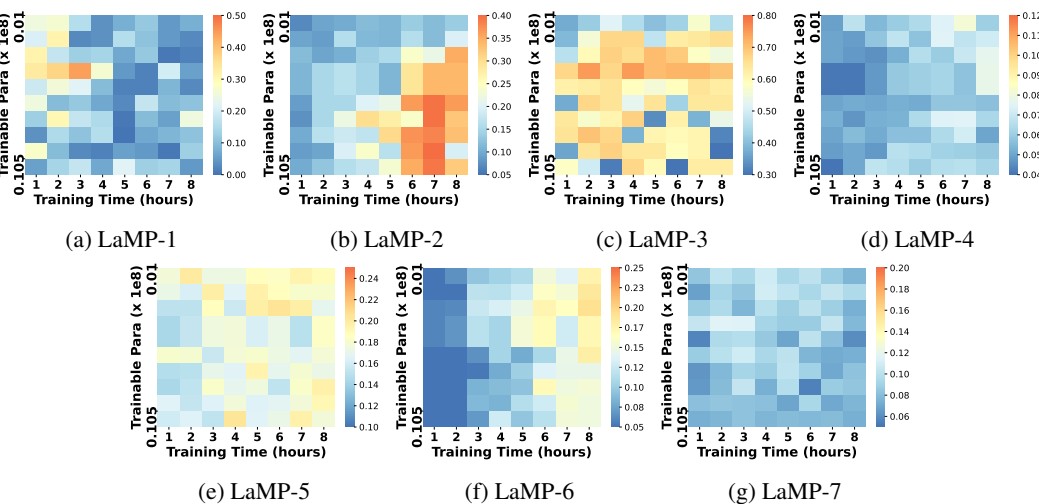

Figure 15: Performance heatmaps for Pythia-1b on dataset LaMP-1 to LaMP-7, given *alpha = rank* = 8, 16, 24, 32, 40, 48, 56, 72, 80. In each heatmap, we use the number of trainable parameters of different *rank* values as y-axis , and the training hours as the x-axis.

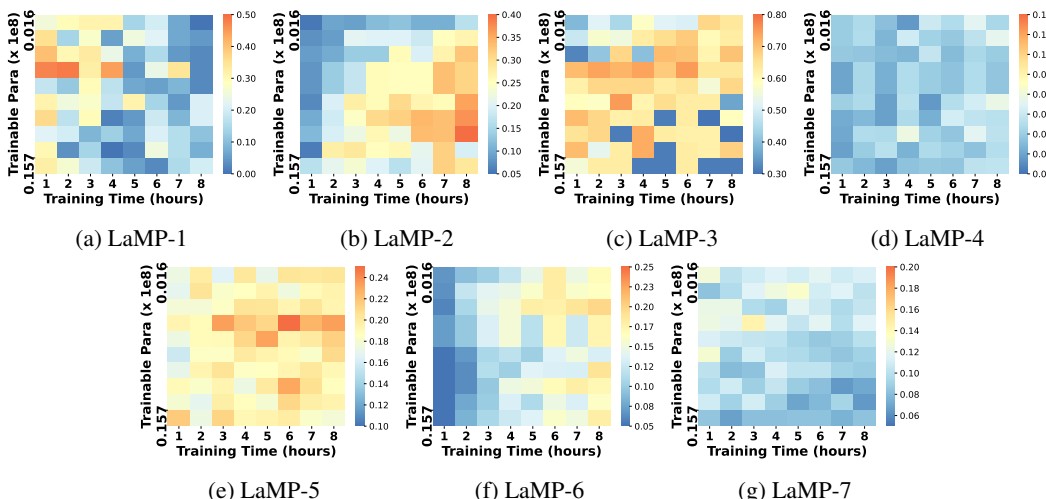

Figure 16: Performance heatmaps for Pythia-1.4b on dataset LaMP-1 to LaMP-7, given *alpha = rank* = 8, 16, 24, 32, 40, 48, 56, 72, 80. In each heatmap, we use the number of trainable parameters of different *rank* values as y-axis , and the training hours as the x-axis.

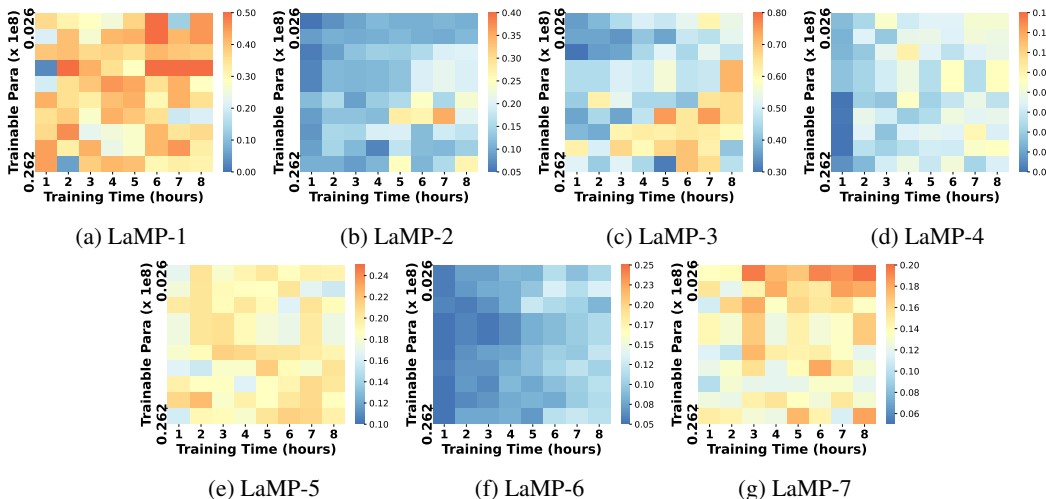

Figure 17: Performance heatmaps for Pythia-2.8b on dataset LaMP-1 to LaMP-7, given *alpha = rank* = 8, 16, 24, 32, 40, 48, 56, 72, 80. In each heatmap, we use the number of trainable parameters of different *rank* values as y-axis , and the training hours as the x-axis.

### B.1.4   EXPERIMENTS: COMBINATIONS OF SEVERAL COMMONLY USED ALPHA AND RANK

The experiments for different combinations of commonly used alpha and rank values as shown in Figure 18 and Figure 19. These experiments can correspond to section 3.2 and section 3.3.

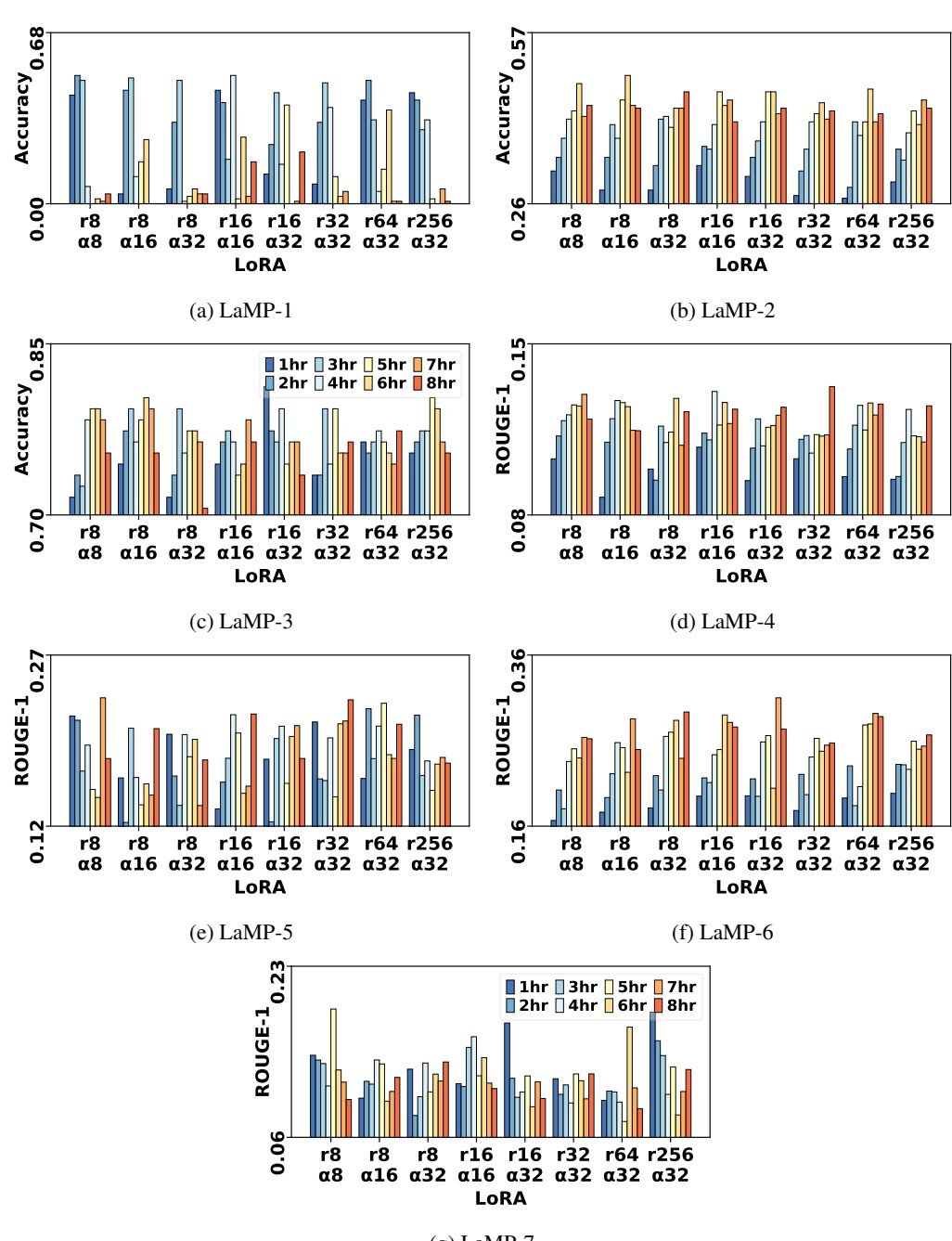

Figure 18: Performance comparison for the selected model StableLM-2-1.6b on seven datasets given different common rank and alpha combinations. The experiments are using the default settings.

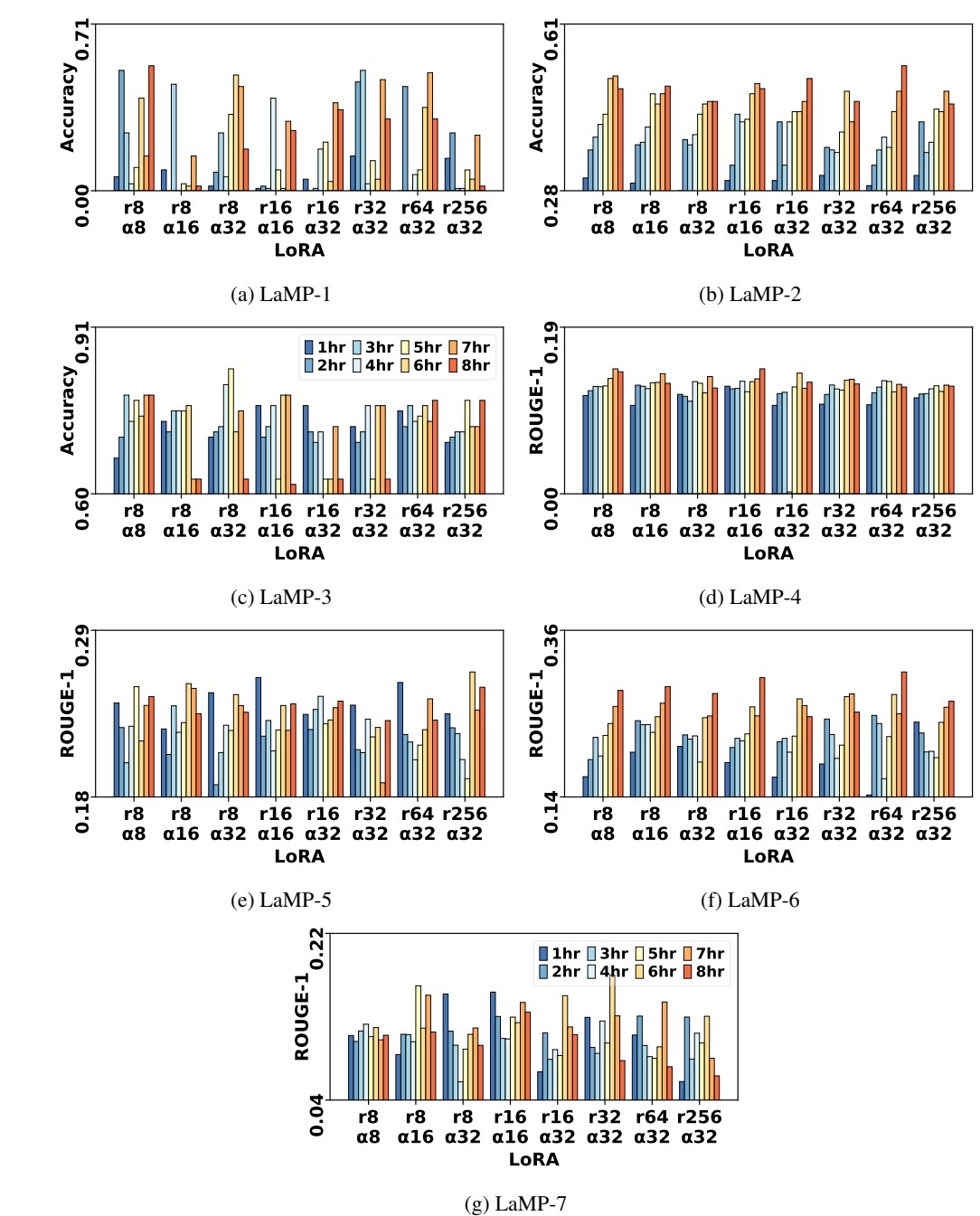

Figure 19: Performance comparison for the selected model StableLM-3b-4e1t on seven datasets given different common rank and alpha combinations. The experiments are using the default settings.

### B.1.5 EXPERIMENTS: MODEL SIZE AND TRAINING TIME

The following results shown in Figure 20 and Figure 21 correspond to Section 3.3.

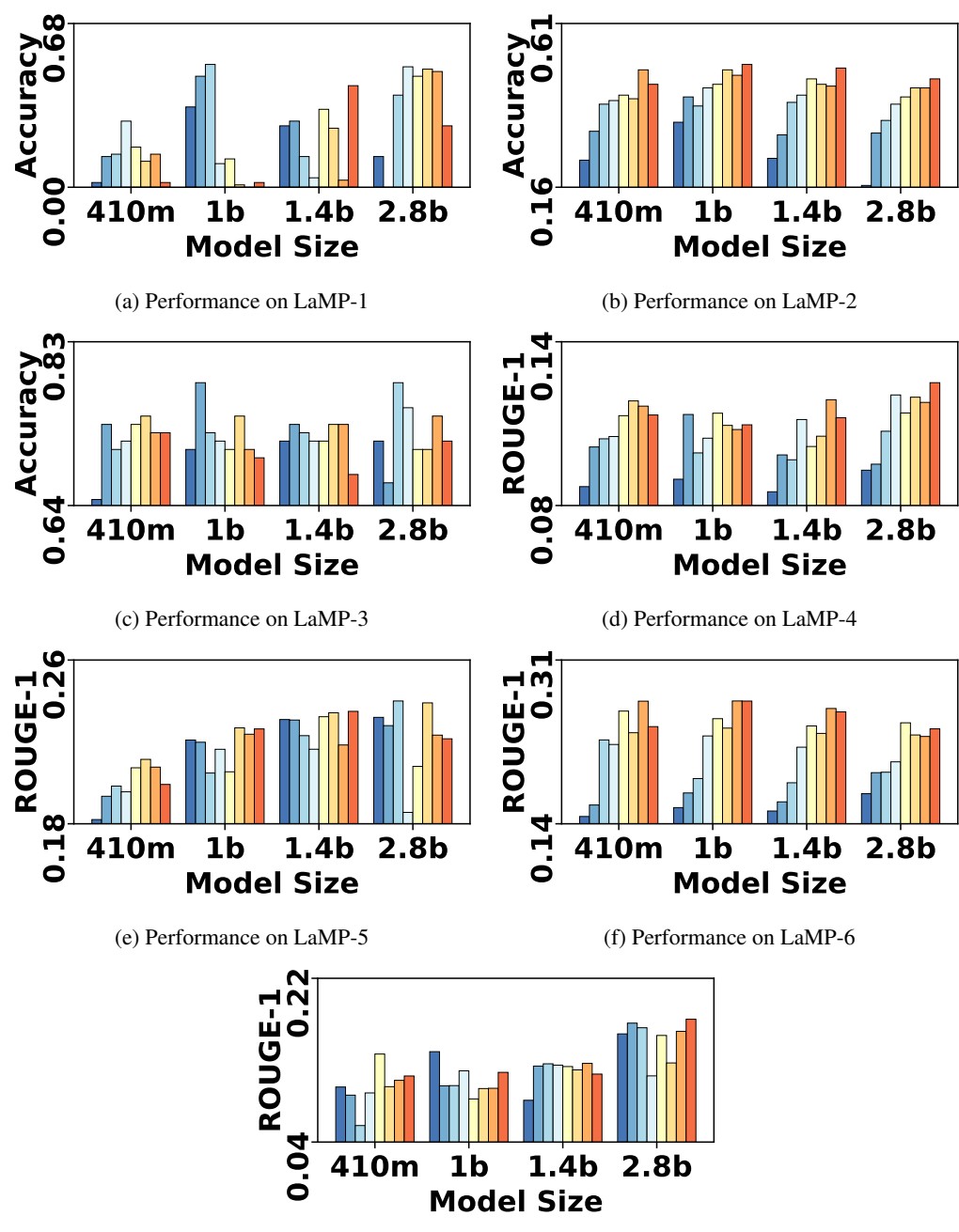

(a) Performance on LaMP-1

(b) Performance on LaMP-2

(c) Performance on LaMP-3

(d) Performance on LaMP-4

(e) Performance on LaMP-5

(f) Performance on LaMP-6

(g) Performance on LaMP-7

Figure 20: Performance comparison of selected LLMs on different RAM sizes, including Pythia(Py)-410m on RAM 4G, Pythia-1b on RAM 6G, Pythia-1.4b on RAM 8G, StableLM(Sta)-1.6b on RAM 10G, and Pythia-2.8b on RAM 16G. Examine their performance across datasets with different difficulty levels. Legend 1 to 8 represent the number of throughputs, where the detailed data size per unit hour can be found in Table 3 Experiments are based on the default settings with *rank* = 8 and *alpha* = 8.

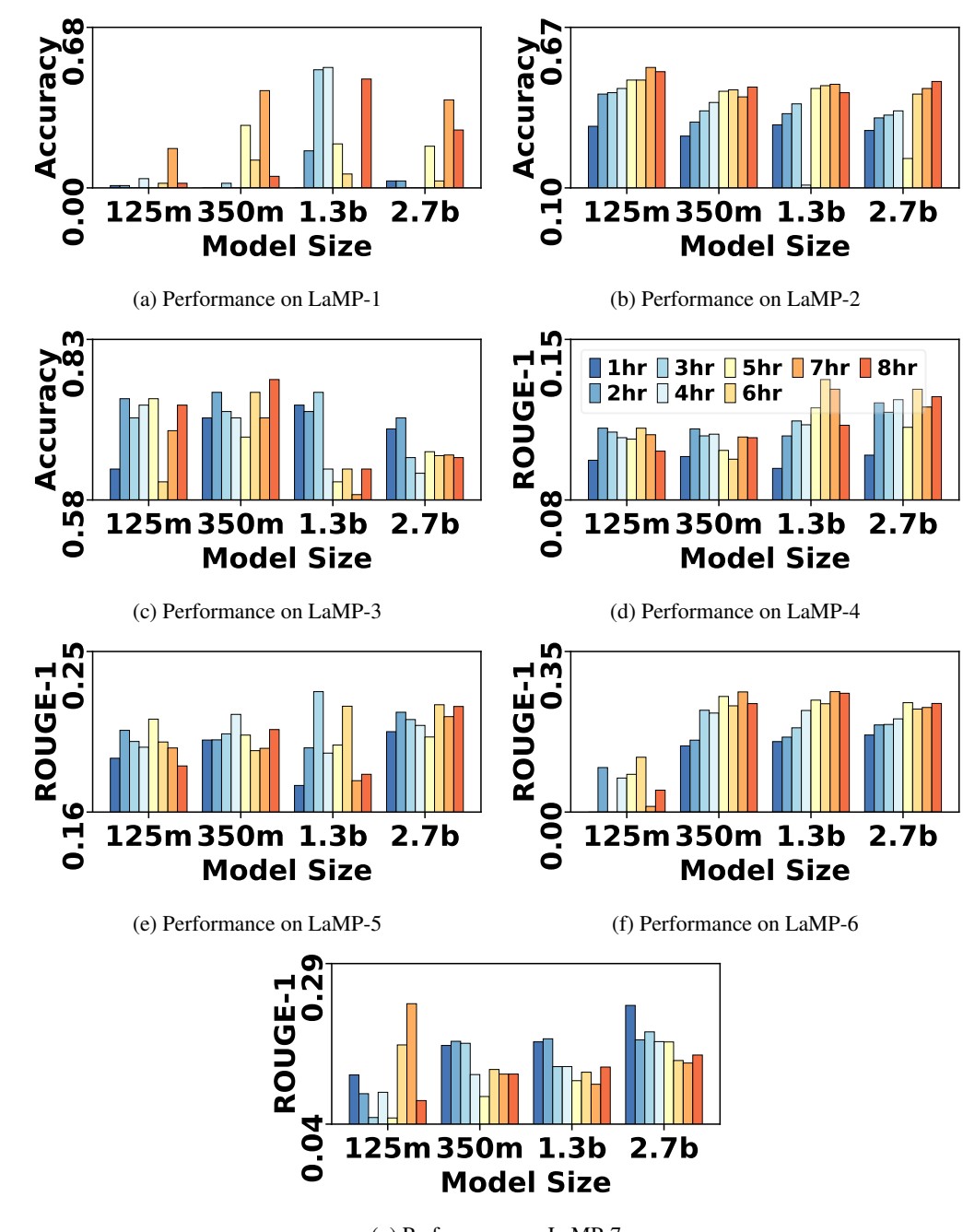

(a) Performance on LaMP-1

(b) Performance on LaMP-2

(c) Performance on LaMP-3

(d) Performance on LaMP-4

(e) Performance on LaMP-5

(f) Performance on LaMP-6

(g) Performance on LaMP-7

Figure 21: Performance comparison of selected LLMs on different RAM sizes, including OPT-125m on RAM 4G, OPT-350m on RAM 6G, OPT-1.3b on RAM 8G, and OPT-2.7b on RAM 16G. Examine their performance across datasets with different difficulty levels. Legend 1 to 8 represent the number of hours of training, where the detailed data size per unit hour can be found in Table 3 Experiments are based on the default settings with *rank* = 8 and *alpha* = 8.

## B.2    EXPERIMENTAL RESULTS FOR RAG

### B.2.1    EXPERIMENTS: RAG PERFORMANCE ON DIFFERENT SIZES OF USER HISTORY DATA

Below, we present the experiments for the impact of the amount of user history data on RAG performance. These experiments can correspond to Section 3.4

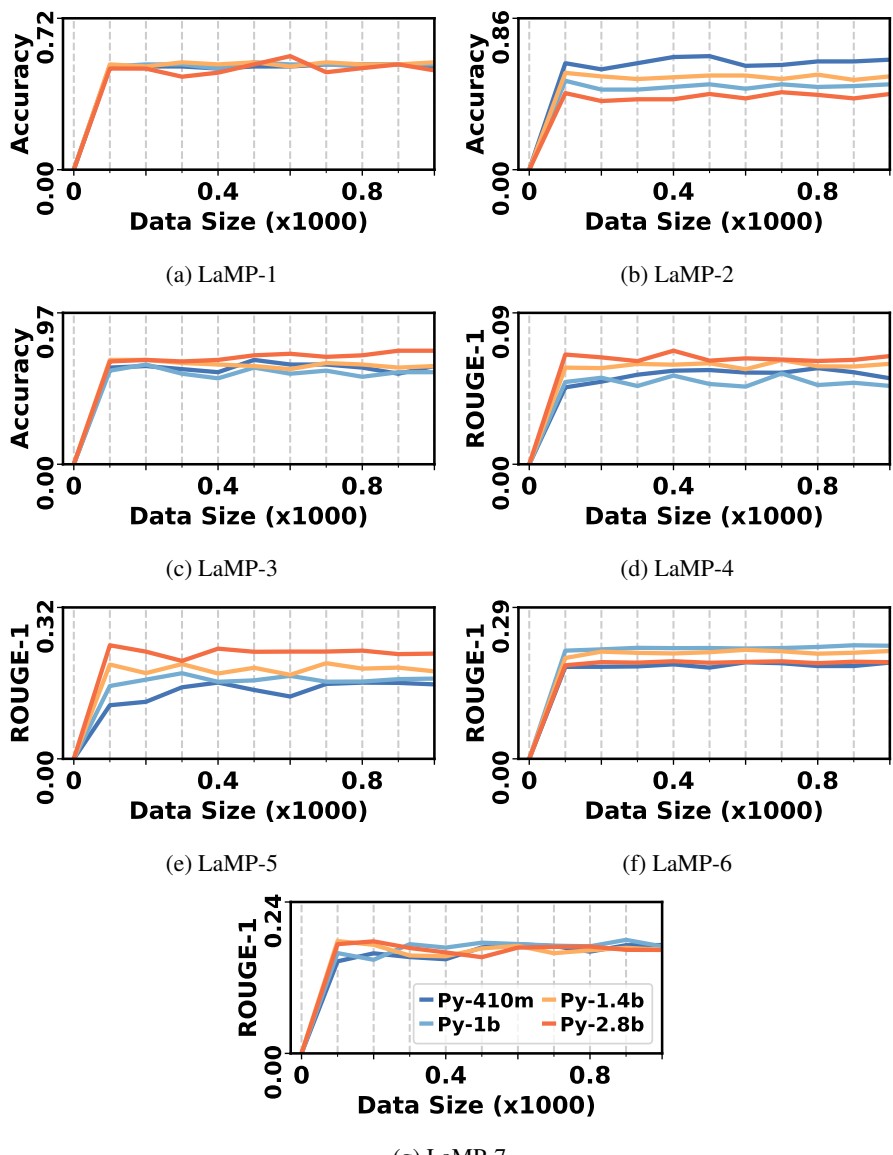

(a) LaMP-1

(b) LaMP-2

(c) LaMP-3

(d) LaMP-4

(e) LaMP-5

(f) LaMP-6

(g) LaMP-7

Figure 22: Performance improvement brought by RAG on four Pythia models of different sizes across different sizes of user history data on all seven datasets.

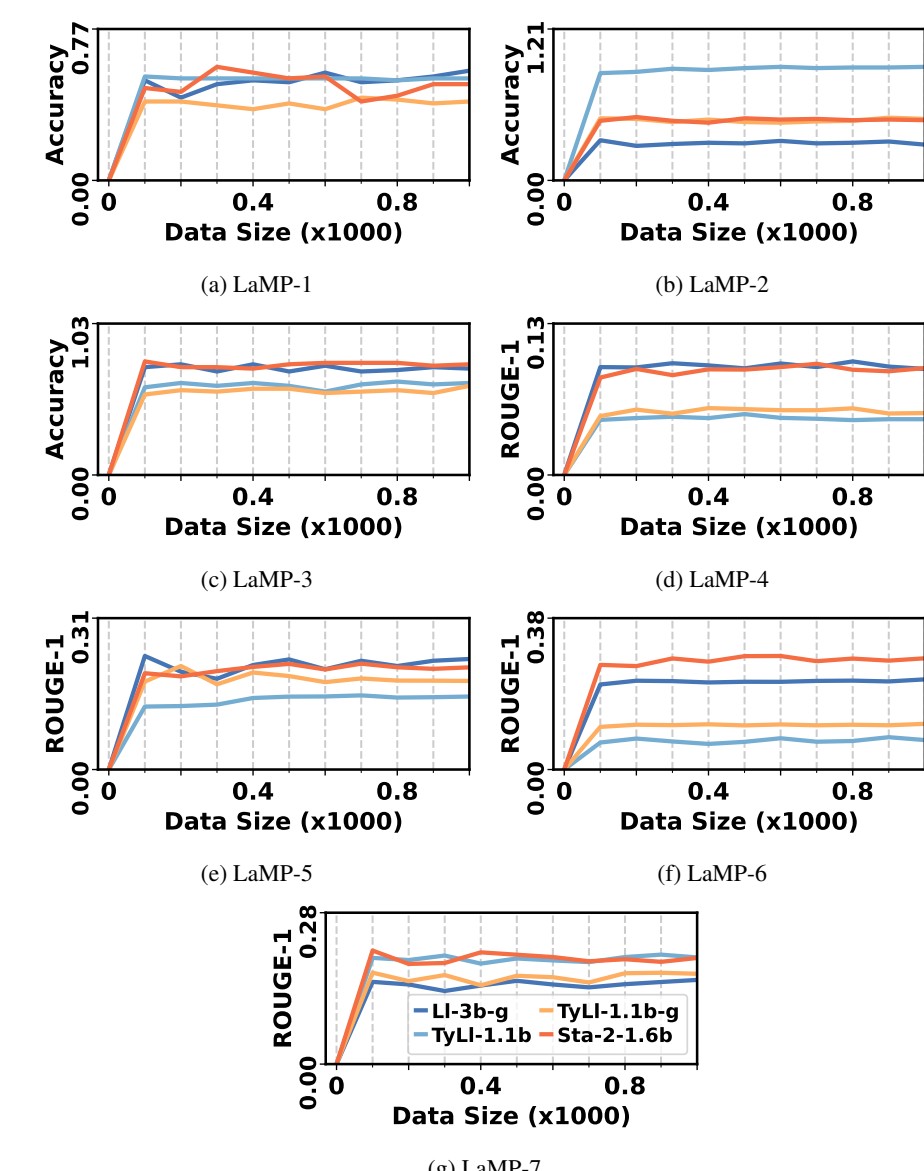

Figure 23: Performance improvement brought by RAG on four models of different sizes across different sizes of user history data on all seven datasets. The four models are Llama(Ll)-3b-GPTQ(G), StableLM(Sta)-2-1.6B, TinyLlama(TyLl)-1.1B-G, and TinyLlama(TyLl)-1.1B.

# C Results of Edge LLM Design

## C.1 Experiments: quantization models

Experiments about quantization models only. These experiments can correspond to section 3.5. We first examine quantized Gemma-2b and Gemma-7b models, showing results in Table 9 and Table 10. Then, we examine the quantized Llama family models, showing results in Figure 24. Additionally, we examine the quantized models with similar size but different architectures, showing results in Figure 25. These experimental results correspond to section 3.5, serving as the supplemental materials for LLM quantization.

Table 9: Performance comparisons between two quantized models with different sizes after training 8 hours.

| Gemma-GPTQ | LaMP-1 | | LaMP-2 | | LaMP-3 | | LaMP-4 | | LaMP-5 | | LaMP-6 | | LaMP-7 | |
|---|---|---|---|---|---|---|---|---|---|---|---|---|---|---|
| | 2b | 7b | 2b | 7b | 2b | 7b | 2b | 7b | 2b | 7b | 2b | 7b | 2b | 7b |
| R(8) A(8) | 0.520 | 0.529 | 0.490 | 0.420 | 0.814 | 0.775 | 0.124 | 0.105 | 0.240 | 0.213 | 0.201 | 0.244 | 0.227 | 0.030 |
| R(8) A(16) | 0.408 | 0.451 | 0.455 | 0.440 | 0.794 | 0.784 | 0.122 | 0.099 | 0.240 | 0.184 | 0.220 | 0.220 | 0.289 | 0.032 |
| R(8) A(32) | 0.475 | 0.304 | 0.440 | 0.425 | 0.725 | 0.784 | 0.119 | 0.095 | 0.209 | 0.187 | 0.241 | 0.228 | 0.231 | 0.093 |
| R(16) A(16) | 0.570 | 0.441 | 0.490 | 0.400 | 0.775 | 0.833 | 0.125 | 0.099 | 0.226 | 0.203 | 0.267 | 0.245 | 0.209 | 0.032 |
| R(16) A(32) | 0.545 | 0.510 | 0.470 | 0.430 | 0.765 | 0.725 | 0.129 | 0.092 | 0.221 | 0.228 | 0.224 | 0.263 | 0.300 | 0.052 |
| R(32) A(32) | 0.421 | 0.441 | 0.505 | 0.415 | 0.755 | 0.755 | 0.113 | 0.096 | 0.202 | 0.200 | 0.231 | 0.221 | 0.186 | 0.088 |
| R(64) A(32) | 0.520 | 0.480 | 0.455 | 0.435 | 0.745 | 0.745 | 0.108 | 0.094 | 0.214 | 0.190 | 0.229 | 0.208 | 0.190 | 0.090 |
| R(256) A(32) | 0.480 | 0.520 | 0.435 | 0.440 | 0.765 | 0.755 | 0.114 | 0.094 | 0.203 | 0.199 | 0.218 | 0.231 | 0.263 | 0.051 |

Table 10: Performance comparisons between two quantized models with different normalized data sizes under the LoRA setting: *rank* = 8 and *alpha* = 32.

| Gemma-GPTQ | LaMP-1 | | LaMP-2 | | LaMP-3 | | LaMP-4 | | LaMP-5 | | LaMP-6 | | LaMP-7 | |
|---|---|---|---|---|---|---|---|---|---|---|---|---|---|---|
| | 2b | 7b | 2b | 7b | 2b | 7b | 2b | 7b | 2b | 7b | 2b | 7b | 2b | 7b |
| Training 1 Hour | 0.469 | 0.480 | 0.265 | 0.240 | 0.745 | 0.706 | 0.094 | 0.074 | 0.223 | 0.222 | 0.119 | 0.160 | 0.300 | 0.037 |
| Training 2 Hours | 0.445 | 0.480 | 0.310 | 0.285 | 0.725 | 0.765 | 0.104 | 0.076 | 0.201 | 0.204 | 0.155 | 0.171 | 0.283 | 0.036 |
| Training 3 Hours | 0.520 | 0.471 | 0.360 | 0.365 | 0.784 | 0.735 | 0.105 | 0.073 | 0.226 | 0.207 | 0.205 | 0.186 | 0.294 | 0.048 |
| Training 4 Hours | 0.410 | 0.186 | 0.415 | 0.350 | 0.784 | 0.775 | 0.116 | 0.088 | 0.205 | 0.200 | 0.169 | 0.195 | 0.263 | 0.036 |
| Training 5 Hours | 0.441 | 0.373 | 0.450 | 0.305 | 0.775 | 0.765 | 0.122 | 0.089 | 0.208 | 0.215 | 0.171 | 0.188 | 0.294 | 0.023 |
| Training 6 Hours | 0.429 | 0.480 | 0.470 | 0.360 | 0.765 | 0.716 | 0.110 | 0.079 | 0.200 | 0.195 | 0.226 | 0.205 | 0.244 | 0.090 |
| Training 7 Hours | 0.408 | 0.480 | 0.465 | 0.395 | 0.775 | 0.775 | 0.114 | 0.099 | 0.243 | 0.201 | 0.208 | 0.177 | 0.277 | 0.061 |
| Training 8 Hours | 0.421 | 0.304 | 0.440 | 0.425 | 0.725 | 0.784 | 0.119 | 0.095 | 0.209 | 0.187 | 0.241 | 0.228 | 0.231 | 0.093 |

---

[1]G represents GPTQ, the quantization technique

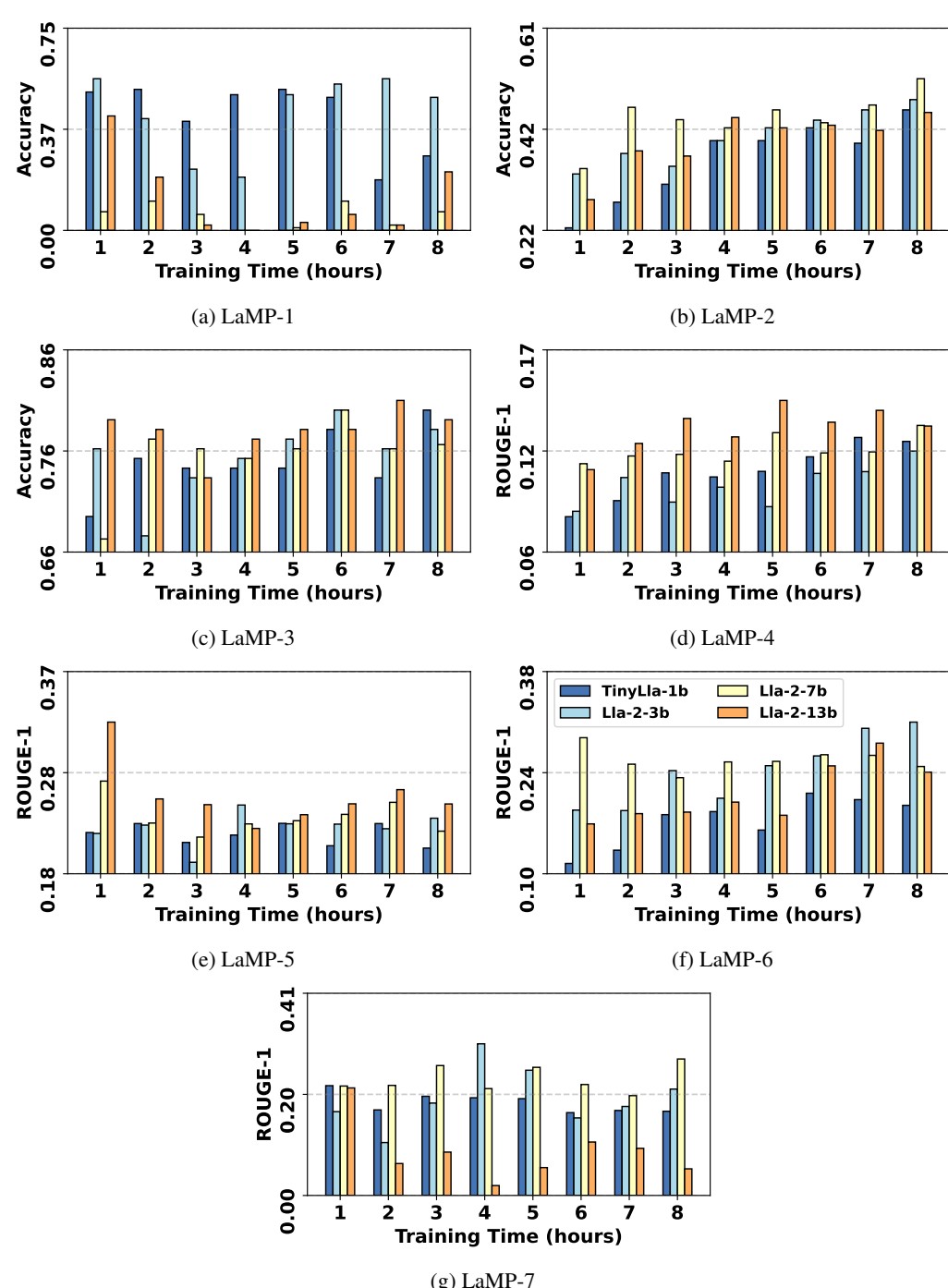

Figure 24: Performance comparison between the quantized Llama model with different sizes across seven datasets. The learning performance can be observed along with the increase of training data size. We use the default settings and set *rank* = 8 and *alpha* = 32 for LoRA.

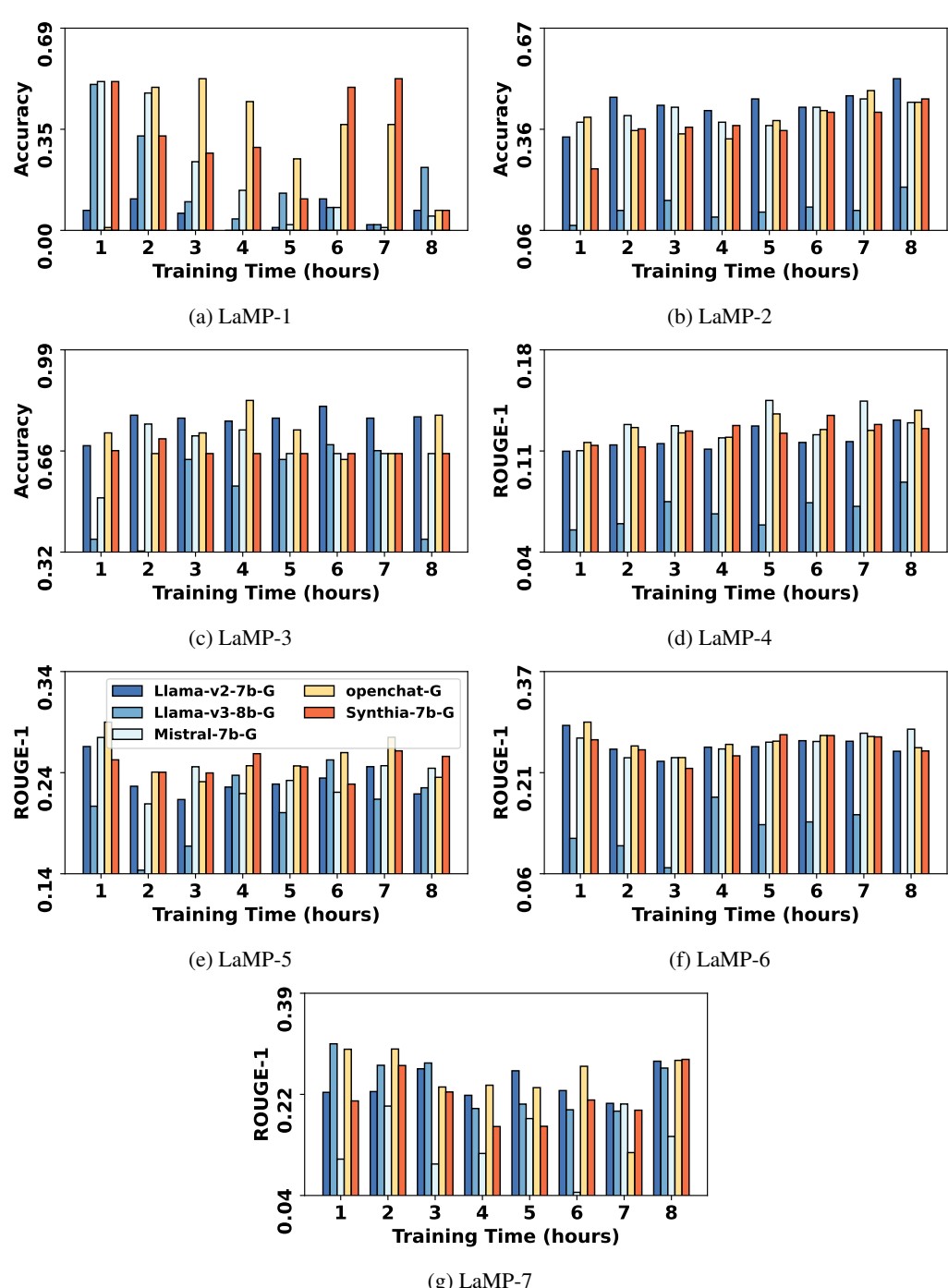

Figure 25: Performance comparison between the different quantized models with the same size across seven datasets. The learning performance can be observed along with the increase in training data size. We use the default settings and set *rank* = 8 and *alpha* = 32 for LoRA.

## C.2 EVALUATION LLMs PRUNING

Experiments to investigate the pruned models: Sheared-Llama-1.3b-Prune and Sheared-Llama-2.7b-Prune. These experiments can correspond to section 3.5 and section 3.6.

We first examine Sheared-Llama-1.3b-Prune (S-Llama-1.3b-P) and Sheared-Llama-1.3b (S-Llama-1.3b) across different data sizes and LoRA settings. The results can be seen in Table 11 and Table 12. Then, we examine Sheared-Llama-1.3b-Prune vs Sheared-Llama-1.3b vs TinyLlama-1.1b. The results can be seen in Figure 26. Additionally, we examine Sheared-Llama-2.7b-Prune vs Sheared-Llama-2.7b across different data size and different lora setting. The results can be seen in Table 13 and Table 14. Finally, we examine Sheared-Llama-2.7b-Prune vs Sheared-Llama-2.7b vs Sheared-Llama-1.3b-Prune vs Sheared-Llama-1.3b vs Llama-3b vs Llama-3b-gptq. The results can be seen in Figure 27.

Table 11: Performance comparisons between pruned (+P) and unpruned (-P) models with different LoRA settings under normalized data size of 8.

| Llama-1.1b | LaMP-1 | | LaMP-2 | | LaMP-3 | | LaMP-4 | | LaMP-5 | | LaMP-6 | | LaMP-7 | |
|---|---|---|---|---|---|---|---|---|---|---|---|---|---|---|
| | -P[2] | +P | -P | +P | -P | +P | -P | +P | -P | +P | -P | +P | -P | +P |
| R(8) A(8) | 0.461 | 0.069 | 0.510 | 0.410 | 0.784 | 0.686 | 0.113 | 0.090 | 0.205 | 0.215 | 0.247 | 0.268 | 0.126 | 0.096 |
| R(8) A(16) | 0.245 | 0.118 | 0.480 | 0.445 | 0.765 | 0.676 | 0.099 | 0.097 | 0.193 | 0.188 | 0.298 | 0.297 | 0.129 | 0.071 |
| R(8) A(32) | 0.010 | 0.186 | 0.500 | 0.450 | 0.725 | 0.696 | 0.093 | 0.091 | 0.221 | 0.184 | 0.291 | 0.281 | 0.125 | 0.084 |
| R(16) A(16) | 0.108 | 0.108 | 0.475 | 0.430 | 0.725 | 0.706 | 0.099 | 0.084 | 0.231 | 0.184 | 0.264 | 0.296 | 0.119 | 0.079 |
| R(16) A(32) | 0.029 | 0.098 | 0.520 | 0.445 | 0.784 | 0.657 | 0.098 | 0.079 | 0.194 | 0.204 | 0.258 | 0.305 | 0.154 | 0.065 |
| R(32) A(32) | 0.000 | 0.059 | 0.480 | 0.455 | 0.804 | 0.686 | 0.103 | 0.083 | 0.208 | 0.212 | 0.280 | 0.268 | 0.098 | 0.074 |
| R(64) A(32) | 0.108 | 0.157 | 0.535 | 0.435 | 0.735 | 0.706 | 0.105 | 0.093 | 0.237 | 0.212 | 0.263 | 0.297 | 0.099 | 0.084 |
| R(256) A(32) | 0.020 | 0.137 | 0.485 | 0.440 | 0.784 | 0.686 | 0.107 | 0.079 | 0.232 | 0.191 | 0.288 | 0.287 | 0.114 | 0.100 |

Table 12: Performance comparisons between pruned (+P) and unpruned (-P) models given LoRA setting where *rank* = 8 and *alpha* = 32, under training hours from 1 to 8.

| Llama-1.1b | LaMP-1 | | LaMP-2 | | LaMP-3 | | LaMP-4 | | LaMP-5 | | LaMP-6 | | LaMP-7 | |
|---|---|---|---|---|---|---|---|---|---|---|---|---|---|---|
| | -P | +P | -P | +P | -P | +P | -P | +P | -P | +P | -P | +P | -P | +P |
| Training 1 Hour | 0.069 | 0.020 | 0.325 | 0.220 | 0.735 | 0.735 | 0.091 | 0.058 | 0.239 | 0.188 | 0.172 | 0.170 | 0.192 | 0.042 |
| Training 2 Hours | 0.000 | 0.039 | 0.380 | 0.335 | 0.755 | 0.667 | 0.080 | 0.074 | 0.259 | 0.188 | 0.169 | 0.190 | 0.139 | 0.060 |
| Training 3 Hours | 0.059 | 0.069 | 0.445 | 0.370 | 0.725 | 0.755 | 0.114 | 0.071 | 0.233 | 0.176 | 0.252 | 0.232 | 0.134 | 0.126 |
| Training 4 Hours | 0.049 | 0.069 | 0.490 | 0.400 | 0.843 | 0.755 | 0.103 | 0.083 | 0.191 | 0.186 | 0.264 | 0.260 | 0.126 | 0.080 |
| Training 5 Hours | 0.049 | 0.049 | 0.495 | 0.395 | 0.775 | 0.716 | 0.107 | 0.076 | 0.208 | 0.208 | 0.234 | 0.238 | 0.141 | 0.086 |
| Training 6 Hours | 0.000 | 0.059 | 0.480 | 0.415 | 0.784 | 0.647 | 0.098 | 0.089 | 0.204 | 0.213 | 0.313 | 0.288 | 0.146 | 0.089 |
| Training 7 Hours | 0.010 | 0.078 | 0.505 | 0.440 | 0.716 | 0.686 | 0.097 | 0.087 | 0.230 | 0.196 | 0.323 | 0.293 | 0.113 | 0.097 |
| Training 8 Hours | 0.010 | 0.186 | 0.500 | 0.450 | 0.725 | 0.696 | 0.093 | 0.091 | 0.221 | 0.184 | 0.291 | 0.281 | 0.125 | 0.084 |

---

[1]P represents Pruning

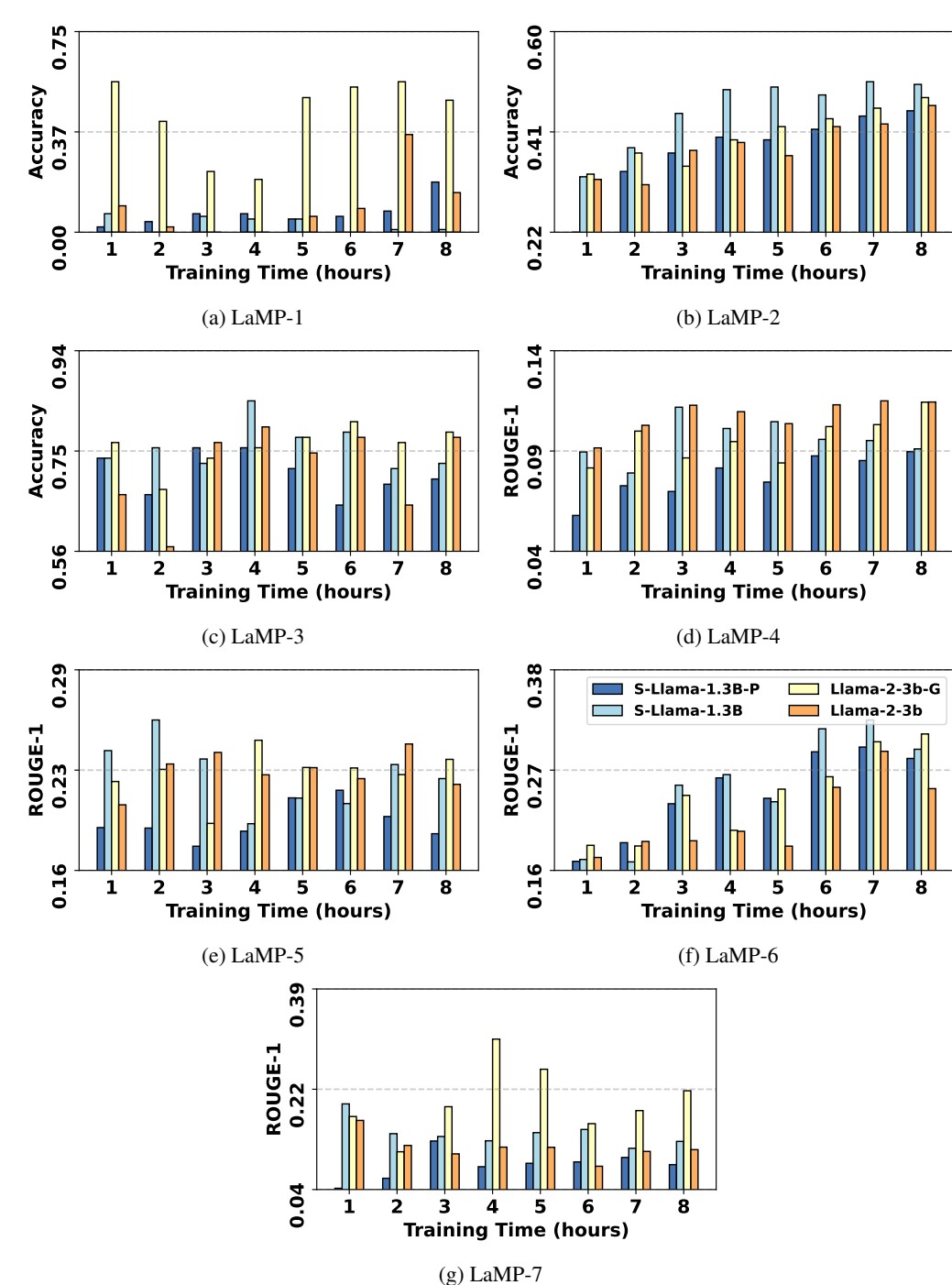

Figure 26: Performance comparison between the different quantized models with the same size across seven datasets. The learning performance can be observed along with the increase of training data size. We use the default settings and set *rank* = 8 and *alpha* = 32 for LoRA.

Table 13: Performance comparisons between pruned (+P) and unpruned (-P) models with different LoRA settings under training hours of 8.

| Sheared-Llama-2.7b | LaMP-1 | | LaMP-2 | | LaMP-3 | | LaMP-4 | | LaMP-5 | | LaMP-6 | | LaMP-7 | |
|---|---|---|---|---|---|---|---|---|---|---|---|---|---|---|
| | -P[3] | +P | -P | +P | -P | +P | -P | +P | -P | +P | -P | +P | -P | +P |
| R(8) A(8) | 0.020 | 0.167 | 0.480 | 0.455 | 0.735 | 0.725 | 0.112 | 0.100 | 0.213 | 0.221 | 0.278 | 0.319 | 0.001 | 0.124 |
| R(8) A(16) | 0.069 | 0.245 | 0.490 | 0.455 | 0.784 | 0.745 | 0.109 | 0.089 | 0.201 | 0.206 | 0.286 | 0.285 | 0.000 | 0.116 |
| R(8) A(32) | 0.078 | 0.059 | 0.495 | 0.450 | 0.755 | 0.765 | 0.112 | 0.089 | 0.208 | 0.217 | 0.332 | 0.301 | 0.147 | 0.133 |
| R(16) A(16) | 0.029 | 0.137 | 0.495 | 0.445 | 0.775 | 0.755 | 0.110 | 0.091 | 0.181 | 0.205 | 0.296 | 0.302 | 0.178 | 0.117 |
| R(16) A(32) | 0.039 | 0.314 | 0.485 | 0.460 | 0.775 | 0.745 | 0.103 | 0.085 | 0.223 | 0.209 | 0.316 | 0.302 | 0.151 | 0.100 |
| R(32) A(32) | 0.029 | 0.088 | 0.510 | 0.445 | 0.784 | 0.755 | 0.109 | 0.102 | 0.227 | 0.213 | 0.329 | 0.312 | 0.183 | 0.093 |
| R(64) A(32) | 0.039 | 0.049 | 0.495 | 0.435 | 0.725 | 0.716 | 0.103 | 0.086 | 0.208 | 0.222 | 0.307 | 0.314 | 0.144 | 0.071 |
| R(256) A(32) | 0.020 | 0.108 | 0.480 | 0.420 | 0.784 | 0.755 | 0.112 | 0.080 | 0.227 | 0.219 | 0.294 | 0.327 | 0.184 | 0.082 |

Table 14: Performance comparisons between pruned (+P) and unpruned (-P) models given LoRA setting where *rank* = 8 and *alpha* = 32, under training hours from 1 to 8.

| Sheared-Llama-1.3b | LaMP-1 | | LaMP-2 | | LaMP-3 | | LaMP-4 | | LaMP-5 | | LaMP-6 | | LaMP-7 | |
|---|---|---|---|---|---|---|---|---|---|---|---|---|---|---|
| | -P | +P | -P | +P | -P | +P | -P | +P | -P | +P | -P | +P | -P | +P |
| Training 1 Hour | 0.010 | 0.039 | 0.320 | 0.200 | 0.706 | 0.676 | 0.097 | 0.074 | 0.212 | 0.241 | 0.180 | 0.093 | 0.000 | 0.128 |
| Training 2 Hours | 0.059 | 0.137 | 0.410 | 0.285 | 0.765 | 0.755 | 0.100 | 0.079 | 0.232 | 0.219 | 0.205 | 0.211 | 0.000 | 0.113 |
| Training 3 Hours | 0.069 | 0.010 | 0.400 | 0.345 | 0.804 | 0.765 | 0.101 | 0.072 | 0.220 | 0.234 | 0.219 | 0.193 | 0.000 | 0.087 |
| Training 4 Hours | 0.294 | 0.039 | 0.415 | 0.395 | 0.775 | 0.775 | 0.108 | 0.070 | 0.216 | 0.210 | 0.218 | 0.185 | 0.234 | 0.131 |
| Training 5 Hours | 0.029 | 0.137 | 0.440 | 0.460 | 0.755 | 0.784 | 0.101 | 0.087 | 0.204 | 0.216 | 0.257 | 0.227 | 0.128 | 0.069 |
| Training 6 Hours | 0.020 | 0.069 | 0.475 | 0.440 | 0.755 | 0.686 | 0.104 | 0.083 | 0.207 | 0.188 | 0.278 | 0.246 | 0.168 | 0.123 |
| Training 7 Hours | 0.000 | 0.078 | 0.505 | 0.470 | 0.755 | 0.794 | 0.112 | 0.093 | 0.202 | 0.209 | 0.311 | 0.304 | 0.172 | 0.108 |
| Training 8 Hours | 0.078 | 0.059 | 0.495 | 0.450 | 0.755 | 0.765 | 0.112 | 0.089 | 0.208 | 0.217 | 0.332 | 0.301 | 0.147 | 0.133 |

---

[1]P represents Pruning

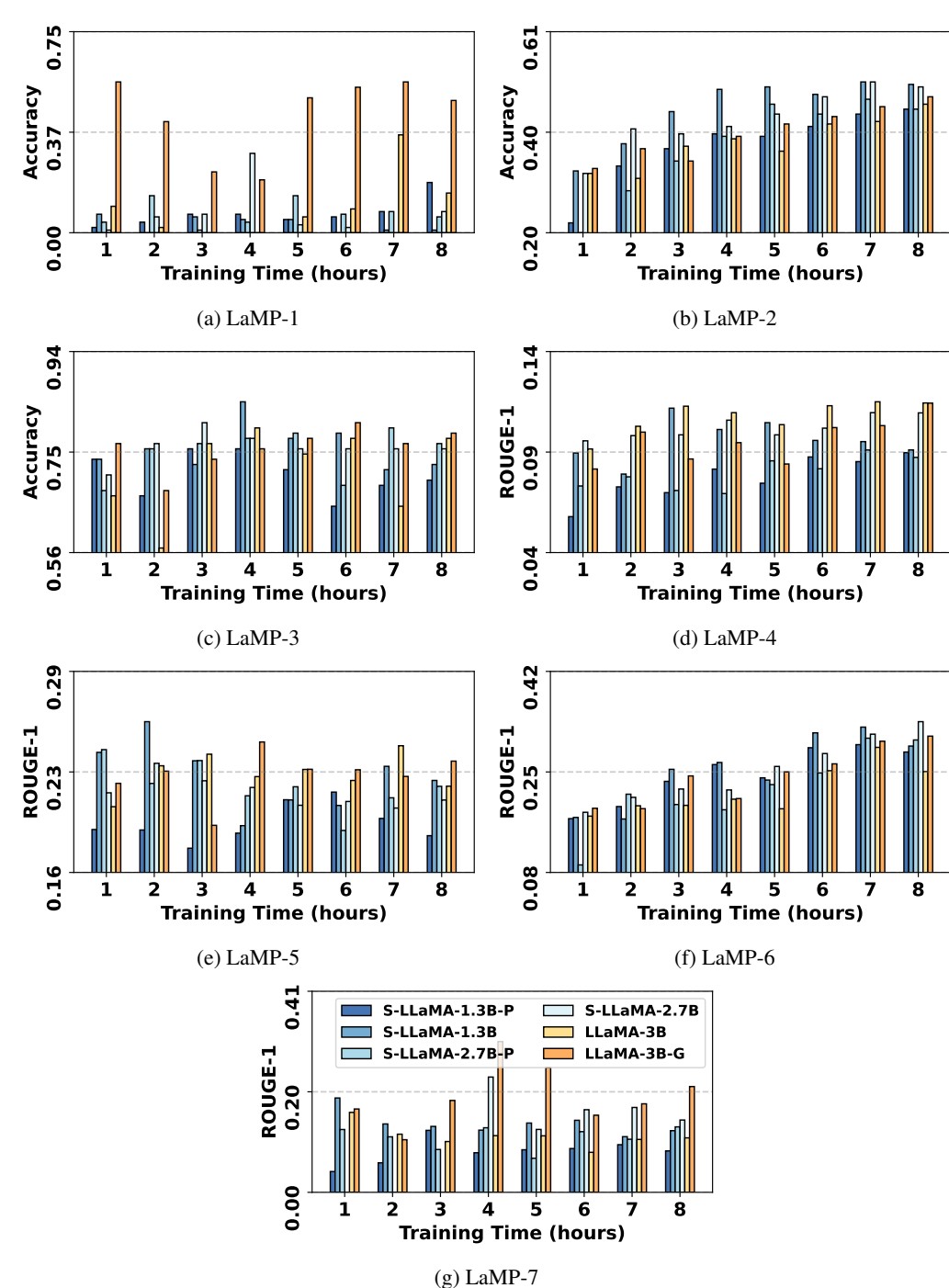

Figure 27: Performance comparison between the different quantized models with the same size across seven datasets. The learning performance can be observed along with the increase in training data size. We use the default settings and set *rank* = 8 and *alpha* = 32 for LoRA.

## C.3 EVALUATION LLMS DISTILLATION

The experiments examine distillation LLM including Phi-1.5, Phi-2, and Phi-3 across different data size and different lora setting. The results can be seen in Table 15 and Table 16. These experiments can correspond to section 3.5.

Table 15: Performance comparisons knowledge distillation models under different LoRA settings under training hours of 8.

| | | LaMP-1 | LaMP-2 | LaMP-3 | LaMP-4 | LaMP-5 | LaMP-6 | LaMP-7 |
|---|---|---|---|---|---|---|---|---|
| Phi-1.5 | R(8) A(8) | 0.255 | 0.405 | 0.696 | 0.119 | 0.252 | 0.265 | 0.181 |
| | R(8) A(16) | 0.500 | 0.430 | 0.686 | 0.115 | 0.250 | 0.261 | 0.159 |
| | R(8) A(32) | 0.441 | 0.435 | 0.716 | 0.119 | 0.224 | 0.280 | 0.112 |
| | R(16) A(16) | 0.490 | 0.410 | 0.706 | 0.115 | 0.237 | 0.277 | 0.133 |
| | R(16) A(32) | 0.137 | 0.440 | 0.706 | 0.114 | 0.247 | 0.248 | 0.171 |
| | R(32) A(32) | 0.490 | 0.420 | 0.706 | 0.120 | 0.256 | 0.243 | 0.148 |
| | R(64) A(32) | 0.441 | 0.435 | 0.706 | 0.116 | 0.220 | 0.255 | 0.104 |
| | R(256) A(32) | 0.402 | 0.430 | 0.696 | 0.117 | 0.229 | 0.273 | 0.139 |
| Phi-2 | R(8) A(8) | 0.206 | 0.450 | 0.745 | 0.133 | 0.243 | 0.241 | 0.220 |
| | R(8) A(16) | 0.529 | 0.445 | 0.725 | 0.132 | 0.237 | 0.281 | 0.221 |
| | R(8) A(32) | 0.520 | 0.465 | 0.765 | 0.137 | 0.264 | 0.273 | 0.208 |
| | R(16) A(16) | 0.373 | 0.435 | 0.745 | 0.130 | 0.253 | 0.254 | 0.219 |
| | R(16) A(32) | 0.529 | 0.455 | 0.765 | 0.141 | 0.259 | 0.285 | 0.208 |
| | R(32) A(32) | 0.529 | 0.455 | 0.775 | 0.137 | 0.247 | 0.297 | 0.209 |
| | R(64) A(32) | 0.529 | 0.450 | 0.784 | 0.134 | 0.240 | 0.290 | 0.215 |
| | R(256) A(32) | 0.520 | 0.440 | 0.745 | 0.135 | 0.232 | 0.268 | 0.191 |
| Phi-3 | R(8) A(8) | 0.471 | 0.470 | 0.784 | 0.103 | 0.211 | 0.198 | 0.199 |
| | R(8) A(16) | 0.529 | 0.450 | 0.804 | 0.102 | 0.209 | 0.248 | 0.176 |
| | R(8) A(32) | 0.520 | 0.470 | 0.794 | 0.109 | 0.224 | 0.221 | 0.159 |
| | R(16) A(16) | 0.549 | 0.455 | 0.833 | 0.100 | 0.210 | 0.226 | 0.142 |
| | R(16) A(32) | 0.529 | 0.465 | 0.745 | 0.098 | 0.210 | 0.222 | 0.197 |
| | R(32) A(32) | 0.216 | 0.505 | 0.775 | 0.094 | 0.218 | nan | 0.125 |
| | R(64) A(32) | 0.255 | 0.480 | 0.804 | 0.090 | 0.233 | 0.125 | 0.191 |
| | R(256) A(32) | 0.402 | 0.490 | 0.833 | 0.094 | 0.252 | 0.129 | 0.127 |

Table 16: Performance comparisons between knowledge distillation models under given LoRA setting where *rank* = 8 and *alpha* = 32, under training hours from 1 to 8.

| | | LaMP-1 | LaMP-2 | LaMP-3 | LaMP-4 | LaMP-5 | LaMP-6 | LaMP-7 |
|---|---|---|---|---|---|---|---|---|
| Phi-1.5 | Training 1 Hour | 0.108 | 0.255 | 0.696 | 0.084 | 0.219 | 0.163 | 0.166 |
| | Training 2 Hours | 0.010 | 0.280 | 0.676 | 0.095 | 0.226 | 0.178 | 0.165 |
| | Training 3 Hours | 0.108 | 0.390 | 0.716 | 0.100 | 0.249 | 0.180 | 0.135 |
| | Training 4 Hours | 0.441 | 0.370 | 0.686 | 0.106 | 0.239 | 0.226 | 0.163 |
| | Training 5 Hours | 0.382 | 0.415 | 0.676 | 0.119 | 0.218 | 0.234 | 0.098 |
| | Training 6 Hours | 0.216 | 0.380 | 0.716 | 0.116 | 0.232 | 0.258 | 0.145 |
| | Training 7 Hours | 0.480 | 0.435 | 0.716 | 0.114 | 0.247 | 0.285 | 0.095 |
| | Training 8 Hours | 0.441 | 0.435 | 0.716 | 0.119 | 0.224 | 0.280 | 0.112 |
| Phi-2 | Training 1 Hour | 0.539 | 0.295 | 0.706 | 0.108 | 0.228 | 0.156 | 0.179 |
| | Training 2 Hours | 0.529 | 0.315 | 0.735 | 0.116 | 0.249 | 0.157 | 0.235 |
| | Training 3 Hours | 0.529 | 0.330 | 0.706 | 0.129 | 0.228 | 0.217 | 0.156 |
| | Training 4 Hours | 0.529 | 0.350 | 0.765 | 0.126 | 0.204 | 0.190 | 0.211 |
| | Training 5 Hours | 0.529 | 0.375 | 0.794 | 0.126 | 0.236 | 0.210 | 0.164 |
| | Training 6 Hours | 0.529 | 0.420 | 0.765 | 0.136 | 0.251 | 0.249 | 0.164 |
| | Training 7 Hours | 0.471 | 0.425 | 0.755 | 0.127 | 0.231 | 0.274 | 0.227 |
| | Training 8 Hours | 0.520 | 0.465 | 0.765 | 0.137 | 0.264 | 0.273 | 0.208 |
| Phi-3 | Training 1 Hour | 0.529 | 0.325 | 0.765 | 0.094 | 0.213 | 0.158 | 0.091 |
| | Training 2 Hours | 0.431 | 0.315 | 0.775 | 0.076 | 0.199 | 0.205 | 0.107 |
| | Training 3 Hours | 0.500 | 0.395 | 0.725 | 0.100 | 0.208 | 0.200 | 0.130 |
| | Training 4 Hours | 0.569 | 0.385 | 0.775 | 0.088 | 0.202 | 0.192 | 0.111 |
| | Training 5 Hours | 0.549 | 0.385 | 0.775 | 0.095 | 0.215 | 0.186 | 0.145 |
| | Training 6 Hours | 0.510 | 0.470 | 0.814 | 0.092 | 0.221 | 0.215 | 0.121 |
| | Training 7 Hours | 0.441 | 0.425 | 0.755 | 0.102 | 0.245 | 0.260 | 0.114 |
| | Training 8 Hours | 0.520 | 0.470 | 0.794 | 0.109 | 0.224 | 0.221 | 0.159 |

C.4 EXPERIMENTS: COMPARE THE THREE COMPRESSION TECHNIQUES

The experiments examine how each compressed model can perform compared to each other. We select a wide range of models from each compression technique. For each task, we use the same heatmap scale to show the optimal model and condition. These experiments correspond to the section 3.5.

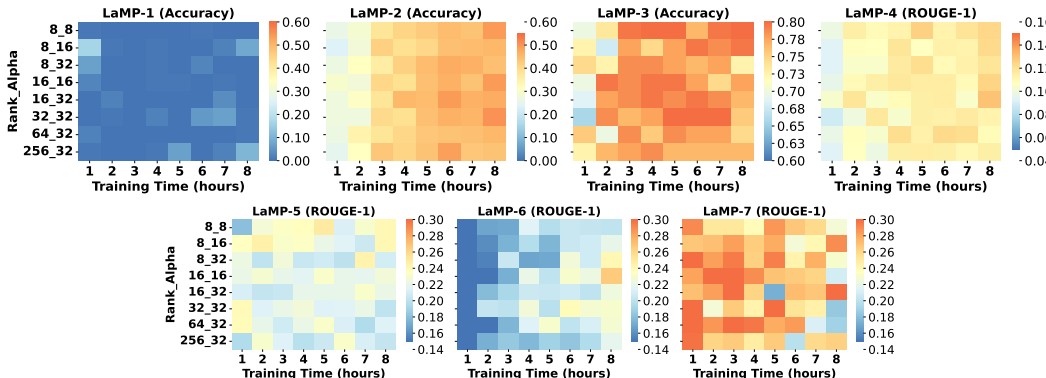

Figure 28: Performances of **Gemma-2b-GPTQ** on eight commonly used combinations of *alpha* and *rank*, over eight hours training time, across seven datasets.

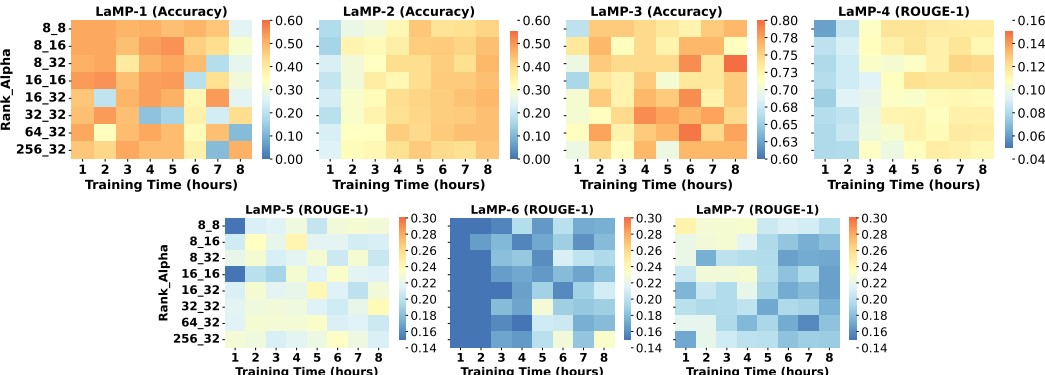

Figure 29: Performances of **TinyLlama-1.1B-GPTQ** on eight commonly used combinations of *alpha* and *rank*, over eight hours training time, across seven datasets.

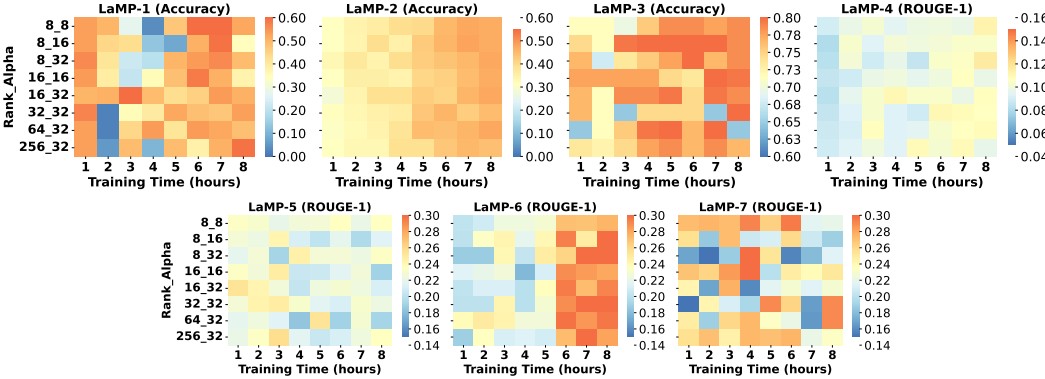

Figure 30: Performances of **Llama-2-3b-GPTQ** on eight commonly used combinations of *alpha* and *rank*, over eight hours training time, across seven datasets.

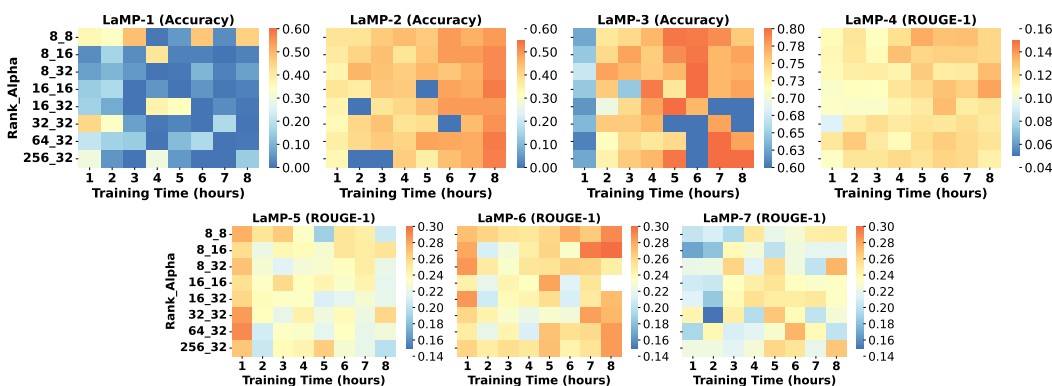

Figure 31: Performances of **Llama-2-7B-GPTQ** on eight commonly used combinations of *alpha* and *rank*, over eight hours training time, across seven datasets.

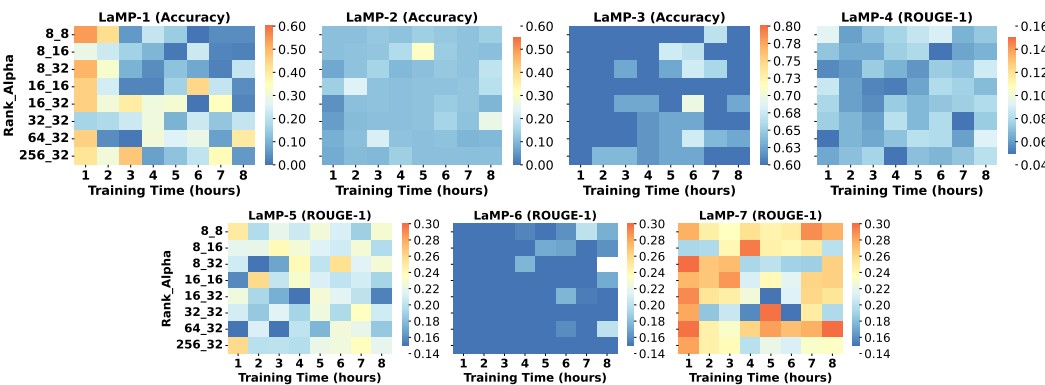

Figure 32: Performances of **Llama-3-8B-GPTQ** on eight commonly used combinations of *alpha* and *rank*, over eight hours training time, across seven datasets.

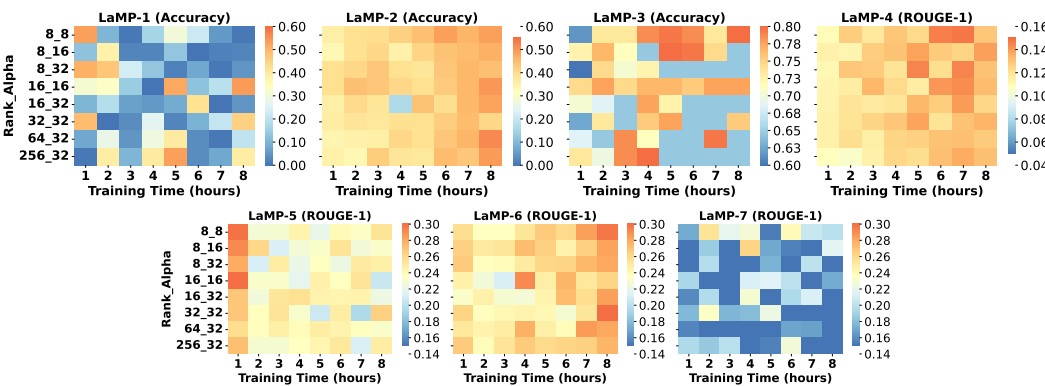

Figure 33: Performances of **Mistral-7b-GPTQ** on eight commonly used combinations of *alpha* and *rank*, over eight hours training time, across seven datasets.

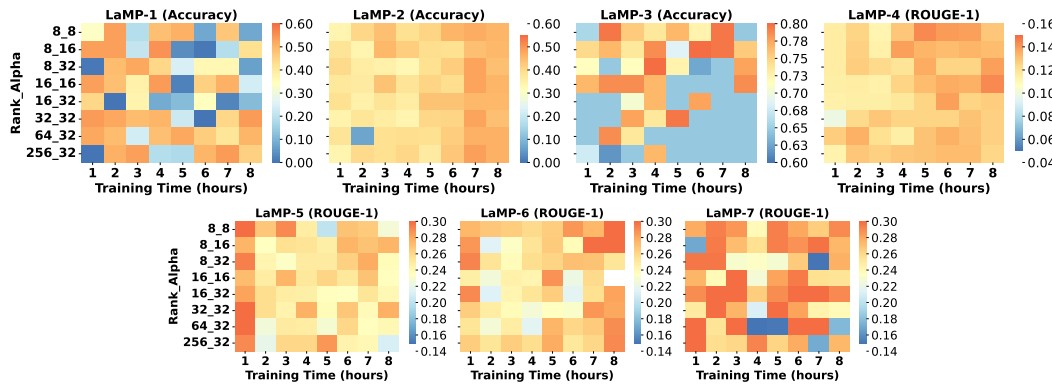

Figure 34: Performances of **OpenChat-3.5-GPTQ** on eight commonly used combinations of *alpha* and *rank*, over eight hours training time, across seven datasets.

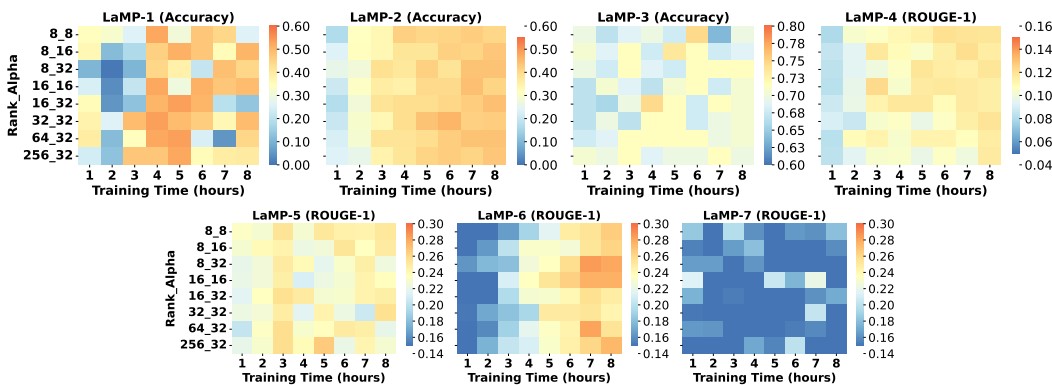

Figure 35: Performances of **Phi-1.5** on eight commonly used combinations of *alpha* and *rank*, over eight hours training time, across seven datasets.

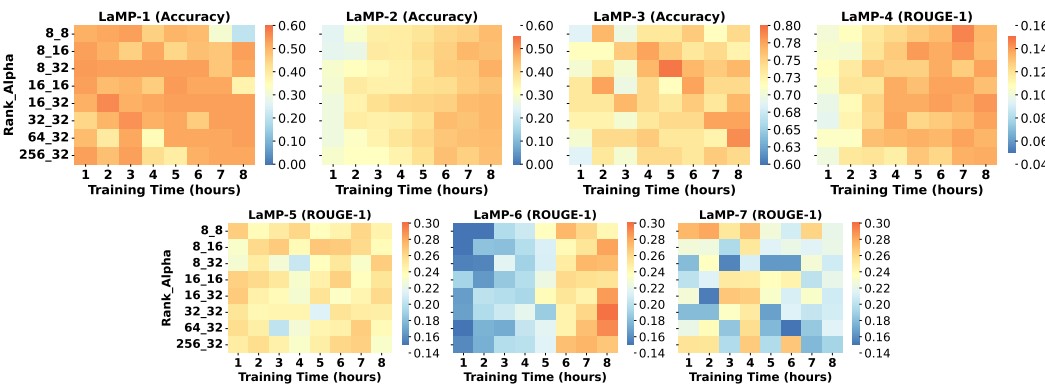

Figure 36: Performances of **Phi-2** on eight commonly used combinations of *alpha* and *rank*, over eight hours training time, across seven datasets.

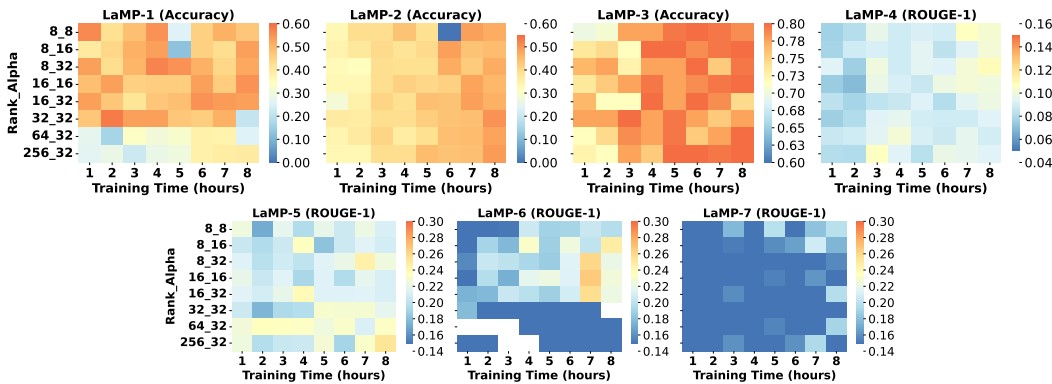

Figure 37: Performances of **Phi-3** on eight commonly used combinations of *alpha* and *rank*, over eight hours training time, across seven datasets.

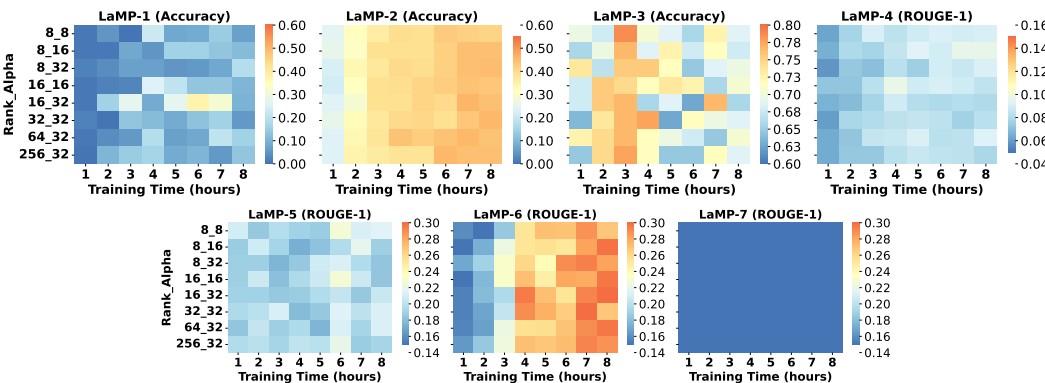

Figure 38: Performances of **Sheared-LLaMA-1.3B-Pruned** on eight commonly used combinations of *alpha* and *rank*, over eight hours training time, across seven datasets.

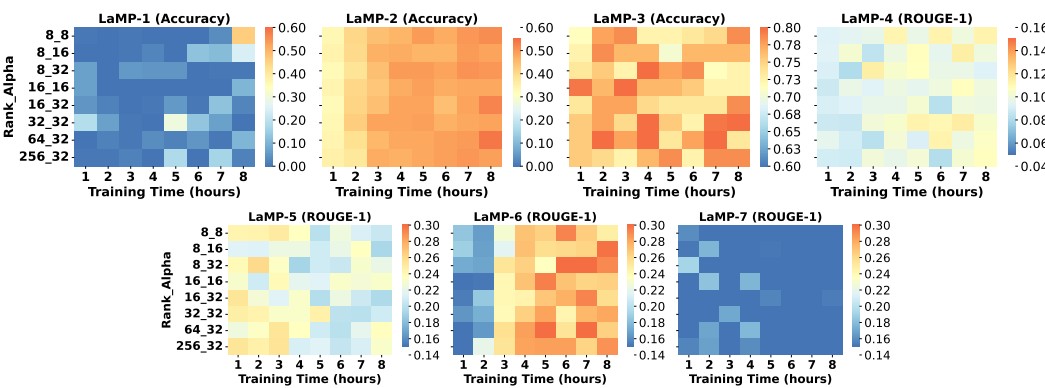

Figure 39: Performances of **Sheared-LLaMA-1.3B** on eight commonly used combinations of *alpha* and *rank*, over eight hours training time, across seven datasets.

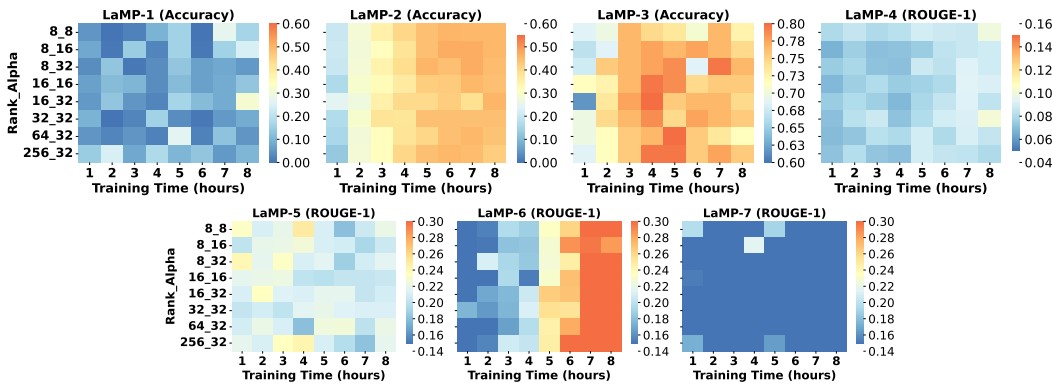

Figure 40: Performances of **Sheared-LLaMA-2.7B-Pruned** on eight commonly used combinations of *alpha* and *rank*, over eight hours training time, across seven datasets.

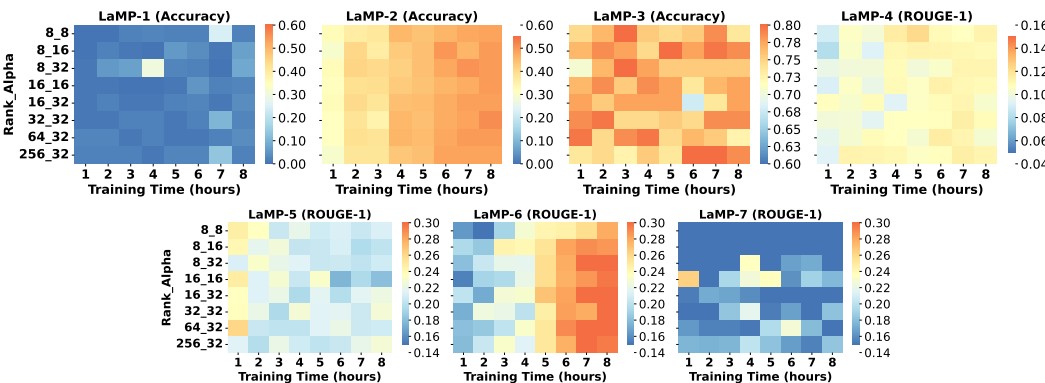

Figure 41: Performances of **Sheared-LLaMA-2.7B** on eight commonly used combinations of *alpha* and *rank*, over eight hours training time, across seven datasets.

## C.5 EXPERIMENTS: COMPARE THE COMPRESSED MODEL WITH UNCOMPRESSED MODELS

These experiments provide additional study concentrating on quantization due to the popular trend of this technique. Thse experiments can correspond to Section 3.6. This section includes Table 17, Table 18, Figure 42, Figure 43, and Figure 44

Table 17: Performance comparisons between quantized (+G) and quantized (-G) models with different LoRA settings under training hours of 8.

| Llama-v2-3b | LaMP-1 | | LaMP-2 | | LaMP-3 | | LaMP-4 | | LaMP-5 | | LaMP-6 | | LaMP-7 | |
|---|---|---|---|---|---|---|---|---|---|---|---|---|---|---|
| | -G[1] | +G | -G | +G | -G | +G | -G | +G | -G | +G | -G | +G | -G | +G |
| R(8) A(8) | 0.020 | 0.520 | 0.430 | 0.470 | 0.765 | 0.784 | 0.129 | 0.113 | 0.252 | 0.234 | 0.264 | 0.275 | 0.204 | 0.220 |
| R(8) A(16) | 0.010 | 0.343 | 0.475 | 0.470 | 0.814 | 0.784 | 0.118 | 0.104 | 0.251 | 0.216 | 0.241 | 0.312 | 0.132 | 0.189 |
| R(8) A(32) | 0.147 | 0.490 | 0.460 | 0.475 | 0.775 | 0.784 | 0.117 | 0.117 | 0.217 | 0.233 | 0.248 | 0.307 | 0.111 | 0.215 |
| R(16) A(16) | 0.294 | 0.373 | 0.450 | 0.470 | 0.833 | 0.794 | 0.116 | 0.098 | 0.230 | 0.190 | 0.270 | 0.280 | 0.097 | 0.251 |
| R(16) A(32) | 0.010 | 0.510 | 0.470 | 0.475 | 0.804 | 0.784 | 0.114 | 0.108 | 0.232 | 0.211 | 0.294 | 0.291 | 0.061 | 0.229 |
| R(32) A(32) | 0.000 | 0.529 | 0.460 | 0.450 | 0.647 | 0.804 | 0.117 | 0.106 | 0.215 | 0.223 | 0.263 | 0.311 | 0.109 | 0.291 |
| R(64) A(32) | 0.196 | 0.480 | 0.455 | 0.475 | 0.794 | 0.647 | 0.115 | 0.109 | 0.235 | 0.191 | 0.263 | 0.296 | 0.125 | 0.290 |
| R(256) A(32) | 0.324 | 0.588 | 0.465 | 0.440 | 0.765 | 0.765 | 0.112 | 0.096 | 0.244 | 0.234 | nan | 0.281 | 0.095 | 0.233 |

| Llama-v3-8b | LaMP-1 | | LaMP-2 | | LaMP-3 | | LaMP-4 | | LaMP-5 | | LaMP-6 | | LaMP-7 | |
|---|---|---|---|---|---|---|---|---|---|---|---|---|---|---|
| | -G[1] | +G | -G | +G | -G | +G | -G | +G | -G | +G | -G | +G | -G | +G |
| R(8) A(8) | 0.324 | 0.039 | 0.130 | 0.140 | 0.627 | 0.461 | 0.065 | 0.073 | 0.180 | 0.229 | 0.172 | 0.174 | 0.189 | 0.277 |
| R(8) A(16) | 0.529 | 0.020 | 0.125 | 0.115 | 0.676 | 0.559 | 0.088 | 0.067 | 0.246 | 0.195 | 0.166 | 0.162 | 0.128 | 0.199 |
| R(8) A(32) | 0.088 | 0.216 | 0.150 | 0.190 | 0.363 | 0.363 | 0.091 | 0.088 | 0.147 | 0.227 | 0.080 | 0.134 | 0.089 | 0.264 |
| R(16) A(16) | 0.265 | 0.039 | 0.115 | 0.180 | 0.637 | 0.412 | 0.094 | 0.070 | 0.186 | 0.192 | 0.179 | 0.172 | 0.155 | 0.263 |
| R(16) A(32) | 0.049 | 0.029 | 0.110 | 0.095 | 0.608 | 0.627 | 0.074 | 0.081 | 0.190 | 0.155 | 0.172 | 0.134 | 0.079 | 0.253 |
| R(32) A(32) | 0.029 | 0.216 | 0.135 | 0.260 | 0.324 | 0.363 | 0.057 | 0.075 | 0.203 | 0.207 | 0.097 | 0.110 | 0.076 | 0.194 |
| R(64) A(32) | 0.098 | 0.402 | 0.115 | 0.100 | 0.627 | 0.627 | 0.069 | 0.092 | 0.211 | 0.185 | 0.158 | 0.200 | 0.135 | 0.306 |
| R(256) A(32) | 0.010 | 0.059 | 0.115 | 0.110 | 0.588 | 0.363 | 0.089 | 0.082 | 0.199 | 0.218 | nan | 0.095 | 0.089 | 0.233 |

| Gemma-2b | LaMP-1 | | LaMP-2 | | LaMP-3 | | LaMP-4 | | LaMP-5 | | LaMP-6 | | LaMP-7 | |
|---|---|---|---|---|---|---|---|---|---|---|---|---|---|---|
| | -G | +G | -G | +G | -G | +G | -G | +G | -G | +G | -G | +G | -G | +G |
| R(8) A(8) | 0.010 | 0.000 | 0.430 | 0.490 | 0.784 | 0.814 | 0.119 | 0.124 | 0.239 | 0.240 | 0.229 | 0.201 | 0.301 | 0.227 |
| R(8) A(16) | 0.000 | 0.078 | 0.465 | 0.455 | 0.794 | 0.794 | 0.126 | 0.122 | 0.202 | 0.240 | 0.239 | 0.220 | 0.165 | 0.289 |
| R(8) A(32) | 0.000 | 0.000 | 0.445 | 0.440 | 0.794 | 0.725 | 0.125 | 0.119 | 0.242 | 0.209 | 0.239 | 0.241 | 0.135 | 0.231 |
| R(16) A(16) | 0.039 | 0.000 | 0.460 | 0.490 | 0.755 | 0.775 | 0.121 | 0.125 | 0.211 | 0.226 | 0.248 | 0.267 | 0.194 | 0.209 |
| R(16) A(32) | 0.020 | 0.000 | 0.445 | 0.470 | 0.755 | 0.765 | 0.121 | 0.129 | 0.227 | 0.221 | 0.246 | 0.224 | 0.173 | 0.300 |
| R(32) A(32) | 0.059 | 0.010 | 0.450 | 0.505 | 0.804 | 0.755 | 0.121 | 0.113 | 0.234 | 0.202 | 0.220 | 0.231 | 0.164 | 0.186 |
| R(64) A(32) | 0.029 | 0.010 | 0.485 | 0.455 | 0.794 | 0.745 | 0.125 | 0.108 | 0.208 | 0.214 | nan | 0.229 | 0.238 | 0.190 |
| R(256) A(32) | 0.000 | 0.108 | 0.440 | 0.435 | 0.794 | 0.765 | 0.121 | 0.114 | 0.225 | 0.203 | nan | 0.218 | 0.193 | 0.263 |

| StabLM-3b | LaMP-1 | | LaMP-2 | | LaMP-3 | | LaMP-4 | | LaMP-5 | | LaMP-6 | | LaMP-7 | |
|---|---|---|---|---|---|---|---|---|---|---|---|---|---|---|
| | -G | +G | -G | +G | -G | +G | -G | +G | -G | +G | -G | +G | -G | +G |
| R(8) A(8) | 0.529 | 0.490 | 0.480 | 0.440 | 0.784 | 0.725 | 0.136 | 0.118 | 0.246 | 0.234 | 0.280 | 0.226 | 0.108 | 0.200 |
| R(8) A(16) | 0.020 | 0.529 | 0.485 | 0.485 | 0.627 | 0.725 | 0.123 | 0.101 | 0.234 | 0.205 | 0.285 | 0.245 | 0.112 | 0.176 |
| R(8) A(32) | 0.176 | 0.520 | 0.455 | 0.440 | 0.627 | 0.775 | 0.118 | 0.105 | 0.235 | 0.251 | 0.276 | 0.269 | 0.098 | 0.218 |
| R(16) A(16) | 0.255 | 0.471 | 0.480 | 0.425 | 0.618 | 0.755 | 0.139 | 0.115 | 0.241 | 0.207 | 0.297 | 0.266 | 0.133 | 0.231 |
| R(16) A(32) | 0.343 | 0.431 | 0.500 | 0.485 | 0.627 | 0.725 | 0.124 | 0.104 | 0.243 | 0.232 | 0.245 | 0.262 | 0.109 | 0.237 |
| R(32) A(32) | 0.304 | 0.549 | 0.455 | 0.420 | 0.627 | 0.627 | 0.122 | 0.105 | 0.230 | 0.184 | 0.251 | 0.269 | 0.082 | 0.219 |
| R(64) A(32) | 0.304 | 0.520 | 0.525 | 0.435 | 0.775 | 0.775 | 0.119 | 0.109 | 0.230 | 0.218 | 0.304 | 0.246 | 0.075 | 0.201 |
| R(256) A(32) | 0.020 | 0.392 | 0.450 | 0.450 | 0.775 | 0.627 | 0.120 | 0.111 | 0.252 | 0.226 | 0.266 | 0.268 | 0.066 | 0.204 |

---

[1]G represents GPTQ, the quantization technique

Table 18: Performance comparisons between quantized (+G) and quantized (-G) models given LoRA setting where *rank* = 8 and *alpha* = 32, under training hours from 1 to 8.

| Llama-v2-3b | LaMP-1 | | LaMP-2 | | LaMP-3 | | LaMP-4 | | LaMP-5 | | LaMP-6 | | LaMP-7 | |
|---|---|---|---|---|---|---|---|---|---|---|---|---|---|---|
| | -G[1] | +G | -G | +G | -G | +G | -G | +G | -G | +G | -G | +G | -G | +G |
| Training 1 Hour | 0.098 | 0.559 | 0.320 | 0.330 | 0.667 | 0.765 | 0.093 | 0.083 | 0.203 | 0.219 | 0.174 | 0.187 | 0.162 | 0.169 |
| Training 2 Hours | 0.020 | 0.412 | 0.310 | 0.370 | 0.569 | 0.676 | 0.105 | 0.102 | 0.230 | 0.227 | 0.191 | 0.186 | 0.118 | 0.107 |
| Training 3 Hours | 0.000 | 0.225 | 0.375 | 0.345 | 0.765 | 0.735 | 0.115 | 0.088 | 0.238 | 0.191 | 0.192 | 0.241 | 0.103 | 0.187 |
| Training 4 Hours | 0.000 | 0.196 | 0.390 | 0.395 | 0.794 | 0.755 | 0.112 | 0.097 | 0.223 | 0.246 | 0.202 | 0.203 | 0.115 | 0.306 |
| Training 5 Hours | 0.059 | 0.500 | 0.365 | 0.420 | 0.745 | 0.775 | 0.106 | 0.086 | 0.228 | 0.228 | 0.186 | 0.248 | 0.115 | 0.253 |
| Training 6 Hours | 0.088 | 0.539 | 0.420 | 0.435 | 0.775 | 0.804 | 0.116 | 0.104 | 0.221 | 0.228 | 0.250 | 0.261 | 0.081 | 0.157 |
| Training 7 Hours | 0.363 | 0.559 | 0.425 | 0.455 | 0.647 | 0.765 | 0.118 | 0.105 | 0.243 | 0.223 | 0.289 | 0.299 | 0.107 | 0.180 |
| Training 8 Hours | 0.147 | 0.490 | 0.460 | 0.475 | 0.775 | 0.784 | 0.117 | 0.117 | 0.217 | 0.233 | 0.248 | 0.307 | 0.111 | 0.215 |

| Llama-v3-8b | LaMP-1 | | LaMP-2 | | LaMP-3 | | LaMP-4 | | LaMP-5 | | LaMP-6 | | LaMP-7 | |
|---|---|---|---|---|---|---|---|---|---|---|---|---|---|---|
| | -G[1] | +G | -G | +G | -G | +G | -G | +G | -G | +G | -G | +G | -G | +G |
| Training 1 Hour | 0.353 | 0.500 | 0.065 | 0.075 | 0.569 | 0.363 | 0.077 | 0.055 | 0.263 | 0.208 | 0.052 | 0.114 | 0.087 | 0.306 |
| Training 2 Hours | 0.304 | 0.324 | 0.095 | 0.120 | 0.569 | 0.324 | 0.066 | 0.060 | 0.239 | 0.144 | 0.067 | 0.103 | 0.107 | 0.268 |
| Training 3 Hours | 0.098 | 0.098 | 0.110 | 0.150 | 0.696 | 0.627 | 0.060 | 0.075 | 0.174 | 0.168 | 0.106 | 0.069 | 0.115 | 0.272 |
| Training 4 Hours | 0.020 | 0.039 | 0.120 | 0.100 | 0.627 | 0.539 | 0.066 | 0.066 | 0.157 | 0.240 | 0.112 | 0.177 | 0.089 | 0.193 |
| Training 5 Hours | 0.020 | 0.127 | 0.125 | 0.115 | 0.627 | 0.627 | 0.083 | 0.059 | 0.253 | 0.202 | 0.088 | 0.135 | 0.103 | 0.200 |
| Training 6 Hours | 0.029 | 0.078 | 0.100 | 0.130 | 0.559 | 0.676 | 0.088 | 0.074 | 0.155 | 0.255 | 0.151 | 0.139 | 0.077 | 0.190 |
| Training 7 Hours | 0.039 | 0.020 | 0.115 | 0.120 | 0.627 | 0.657 | 0.064 | 0.072 | 0.132 | 0.216 | 0.130 | 0.150 | 0.082 | 0.188 |
| Training 8 Hours | 0.088 | 0.216 | 0.150 | 0.190 | 0.363 | 0.363 | 0.091 | 0.088 | 0.147 | 0.227 | 0.080 | 0.134 | 0.089 | 0.264 |

| Gemma-2b | LaMP-1 | | LaMP-2 | | LaMP-3 | | LaMP-4 | | LaMP-5 | | LaMP-6 | | LaMP-7 | |
|---|---|---|---|---|---|---|---|---|---|---|---|---|---|---|
| | -G[1] | +G | -G | +G | -G | +G | -G | +G | -G | +G | -G | +G | -G | +G |
| Training 1 Hour | 0.010 | 0.069 | 0.275 | 0.265 | 0.696 | 0.745 | 0.102 | 0.094 | 0.231 | 0.223 | 0.127 | 0.119 | 0.154 | 0.300 |
| Training 2 Hours | 0.098 | 0.000 | 0.320 | 0.310 | 0.755 | 0.725 | 0.113 | 0.104 | 0.226 | 0.201 | 0.159 | 0.155 | 0.140 | 0.283 |
| Training 3 Hours | 0.000 | 0.000 | 0.390 | 0.360 | 0.706 | 0.784 | 0.112 | 0.105 | 0.208 | 0.226 | 0.176 | 0.205 | 0.165 | 0.294 |
| Training 4 Hours | 0.186 | 0.010 | 0.390 | 0.415 | 0.794 | 0.784 | 0.114 | 0.116 | 0.233 | 0.205 | 0.207 | 0.169 | 0.139 | 0.263 |
| Training 5 Hours | 0.010 | 0.000 | 0.460 | 0.450 | 0.784 | 0.775 | 0.116 | 0.122 | 0.213 | 0.208 | 0.197 | 0.171 | 0.167 | 0.294 |
| Training 6 Hours | 0.010 | 0.029 | 0.435 | 0.470 | 0.784 | 0.765 | 0.115 | 0.110 | 0.223 | 0.200 | 0.213 | 0.226 | 0.141 | 0.244 |
| Training 7 Hours | 0.039 | 0.000 | 0.480 | 0.465 | 0.824 | 0.775 | 0.118 | 0.114 | 0.200 | 0.243 | 0.217 | 0.208 | 0.153 | 0.277 |
| Training 8 Hours | 0.000 | 0.000 | 0.445 | 0.440 | 0.794 | 0.725 | 0.125 | 0.119 | 0.242 | 0.209 | 0.239 | 0.241 | 0.135 | 0.231 |

| StableLM-3b | LaMP-1 | | LaMP-2 | | LaMP-3 | | LaMP-4 | | LaMP-5 | | LaMP-6 | | LaMP-7 | |
|---|---|---|---|---|---|---|---|---|---|---|---|---|---|---|
| | -G[1] | +G | -G | +G | -G | +G | -G | +G | -G | +G | -G | +G | -G | +G |
| Training 1 Hour | 0.020 | 0.480 | 0.280 | 0.325 | 0.706 | 0.676 | 0.111 | 0.088 | 0.248 | 0.200 | 0.206 | 0.191 | 0.152 | 0.137 |
| Training 2 Hours | 0.078 | 0.510 | 0.380 | 0.285 | 0.716 | 0.716 | 0.108 | 0.094 | 0.188 | 0.179 | 0.221 | 0.210 | 0.113 | 0.128 |
| Training 3 Hours | 0.245 | 0.539 | 0.370 | 0.370 | 0.725 | 0.716 | 0.103 | 0.103 | 0.209 | 0.223 | 0.216 | 0.211 | 0.098 | 0.152 |
| Training 4 Hours | 0.059 | 0.500 | 0.390 | 0.365 | 0.804 | 0.784 | 0.125 | 0.094 | 0.227 | 0.193 | 0.220 | 0.227 | 0.059 | 0.149 |
| Training 5 Hours | 0.324 | 0.510 | 0.430 | 0.380 | 0.833 | 0.735 | 0.123 | 0.101 | 0.223 | 0.208 | 0.186 | 0.234 | 0.094 | 0.235 |
| Training 6 Hours | 0.490 | 0.510 | 0.450 | 0.380 | 0.716 | 0.765 | 0.112 | 0.113 | 0.247 | 0.238 | 0.244 | 0.244 | 0.109 | 0.180 |
| Training 7 Hours | 0.441 | 0.471 | 0.455 | 0.390 | 0.755 | nan | 0.130 | 0.105 | 0.240 | 0.229 | 0.246 | 0.242 | 0.116 | 0.227 |
| Training 8 Hours | 0.176 | 0.520 | 0.455 | 0.440 | 0.627 | 0.775 | 0.118 | 0.105 | 0.235 | 0.251 | 0.276 | 0.269 | 0.098 | 0.218 |

[1]G represents GPTQ, the quantization technique

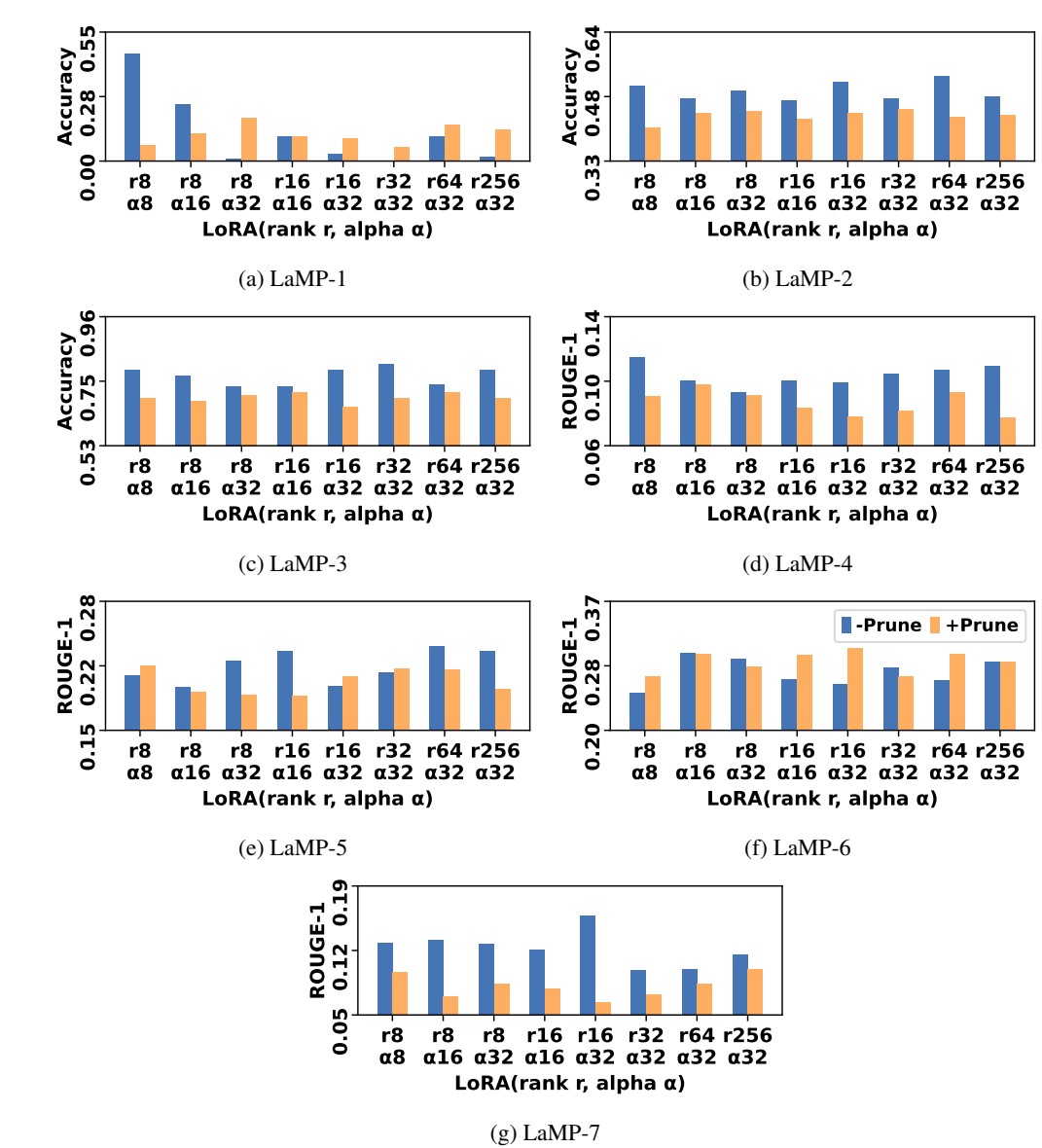

(a) LaMP-1

(b) LaMP-2

(c) LaMP-3

(d) LaMP-4

(e) LaMP-5

(f) LaMP-6

(g) LaMP-7

Figure 42: Performance comparison between pruned and unpruned **Sheared-LLaMA-1.3B** under training hours from 1 to 8

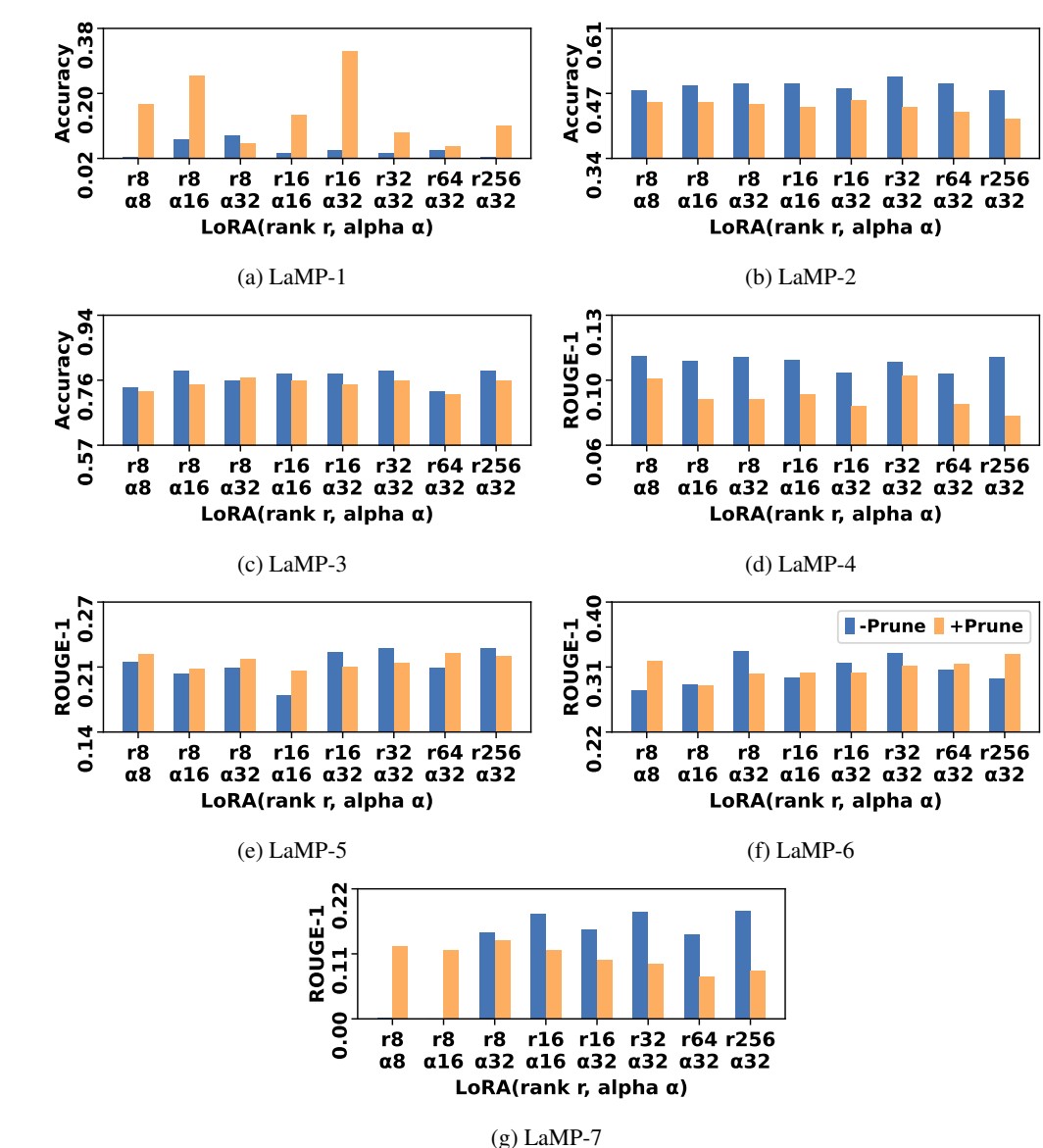

Figure 43: Performance comparison between pruned and unpruned **Sheared-LLaMA-2.7B** under training hours from 1 to 8

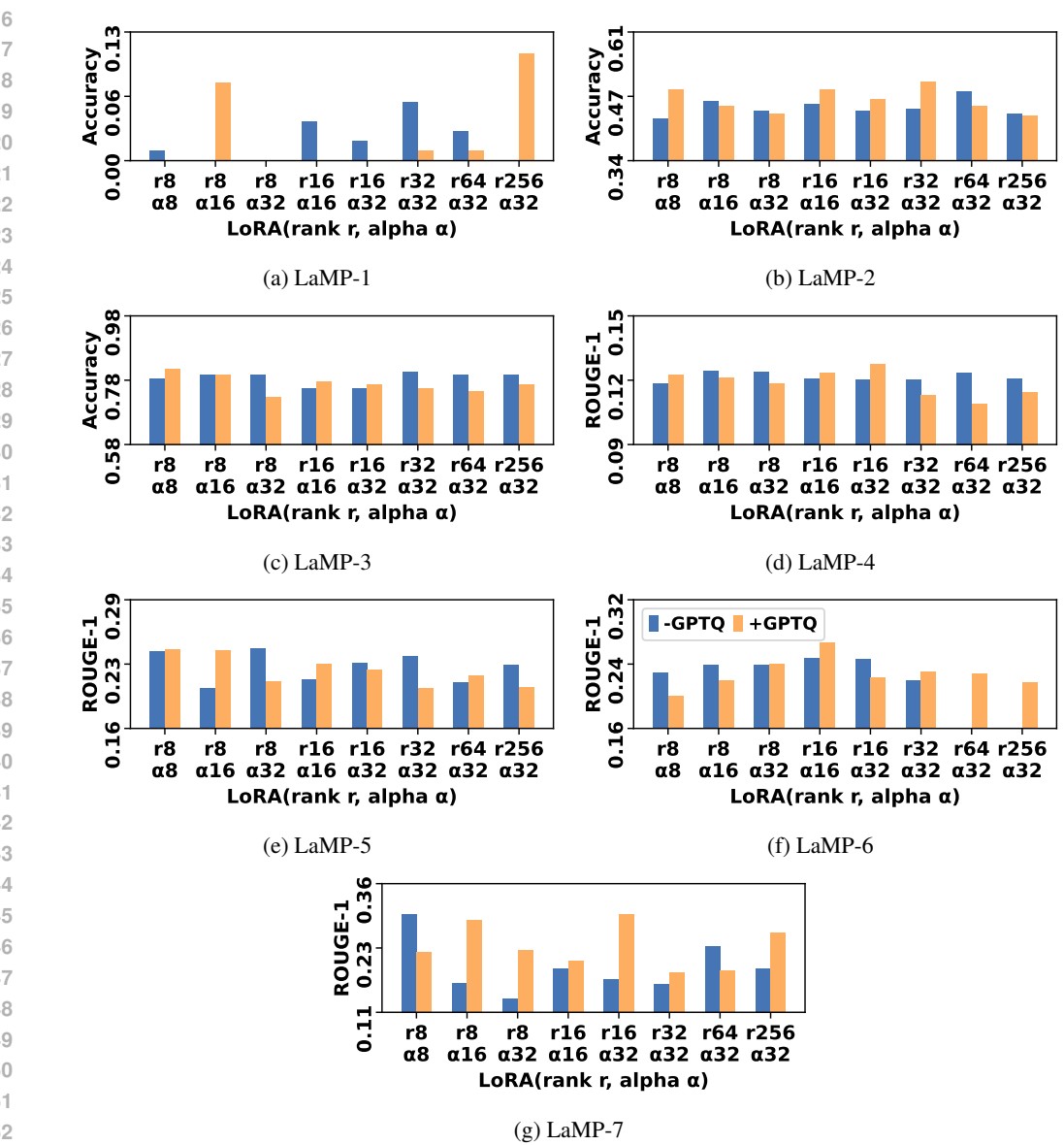

(a) LaMP-1

(b) LaMP-2

(c) LaMP-3

(d) LaMP-4

(e) LaMP-5

(f) LaMP-6

(g) LaMP-7

Figure 44: Performance comparison between quantized and unquantized **Gemma-2b** under training hours from 1 to 8

## D  Background and Related Works

### D.1  Large Language Models Customization for Downstream Tasks

Different from BERT family language models (Devlin et al., 2019; Sanh, 2019; Liu, 2019; Qin et al., 2021), Large Language Models (LLMs), such as GPT-4 (Achiam et al., 2023), Llama (Touvron et al., 2023), and T5 (Raffel et al., 2023), are pre-trained on vast amounts of text data (Pan et al., 2024). This pretraining enables them to learn a wide range of language patterns, structures, and knowledge. However, to achieve optimal performance on specific downstream tasks such as classification, summarization, or translation, these models often require further customizations (Zhang et al., 2024b; Hu et al., 2024). Several approaches can be employed to tailor LLMs for specific downstream tasks, and we can generally group them into two categories: fine-tuning and prompt engineering. Fine-tuning uses targetted datasets to update the weights of LLMs so that the models can be familiar with the desired knowledge and structure of the outputs (Zhang et al., 2024b). The dominant approach uses variations of low-rank approximation (LoRA) techniques (Hu et al., 2021; Xu et al., 2024; Dettmers et al., 2023; Babakniya et al., 2023; Sun et al., 2024), which freeze the original LLM weights and add additional low-rank decomposition weights that simulate weight updates. On the other hand, prompt engineering achieves downstream optimization without changing the LLM's weights. Typical approaches include few-shot in-context learning (Coda-Forno et al., 2023) and retrieval-augmented generation (RAG) (Lewis et al., 2020). In particular, RAG retrieves relevant documents from a knowledge base and adds the knowledge pertinent to the prompt. It grounds the generated answer with the retrieved knowledge and significantly increases response quality on commercial Cloud LLMs.

### D.2  Large Language Model Evaluations

The field of Large Language Models (LLMs) has seen a significant amount of research focusing on various aspects of model evaluation (Guo et al., 2023; Chang et al., 2024; Wei et al., 2024). Numerous studies have proposed benchmarks to systematically evaluate the capabilities of LLMs across different tasks. On the benchmark side, notable benchmarks include MMLU (Hendrycks et al., 2020), GPQA (Rein et al., 2023), HumanEval (Peng et al., 2024b), and GSM-8K (Cobbe et al., 2021), but they are not suitable for edge LLMs due to the reasons explained in the introduction section. In this paper, we focus on the LaMP dataset (Salemi et al., 2023), which closely aligns with the common use cases of edge LLMs. On the side of optimizations, research has explored the impact of quantization on LLM performance, aiming to make these models more efficient without significantly degrading their accuracy (Li et al., 2024). Optimal fine-tuning techniques have been extensively studied by works like Pu et al. (2023) that provides a framework for choosing the optimal fine-tuning techniques given the task type and data availability, Liu et al. (2024) that optimizes transformer architecture for edge devices, and Lin et al. (2024a) that focuses on the selection of LLM. However, to the best of our knowledge, those existing evaluation papers either focus on cloud LLMs or do not fully address the constraints outlined in the introduction, thus unable to provide a guidebook for edge LLMs.

