# OpenReview forum: "Empirical Guidelines for Deploying LLMs onto Resource-constrained Edge Devices"
_ICLR.cc/2025/Conference — ICLR 2025 Conference Withdrawn Submission_

### Official Review · Reviewer_PpSq · 2024-10-27

**Soundness:** 4
**Presentation:** 3
**Contribution:** 3
**Rating:** 6
**Confidence:** 3

**Summary:**

This paper addresses various design and deployment choices for running LLMs on edge devices. I enjoyed reading this paper, as the authors provide insights with clear and empirical experimental results.

**Strengths:**

- As LLM deployment differs significantly on edge devices due to limited computing resources and the use of private data, the comprehensive evaluation and criteria for optimization techniques are crucial topics
- The authors conduct comprehensive and well-defined experiments
- The authors’ findings are novel, as they can guide future research directions

**Weaknesses:**

- In the introduction, many statements and findings seem to pertain to training (fine-tuning). Clearly stating the focus and deployment scenario would make the paper’s sections clearer (fine-tuning, inference, or both?)
- Instead of focusing on individual techniques for LLM deployment, what about combining two or three methods? For example, RAG and LoRA could potentially be applied together
- I understand the page limit and the authors’ efforts to address this issue, but many supporting results are in the Appendix, making it somewhat difficult to follow thoroughly.
- The authors experiment with specified models on particular devices, considering memory constraints. What about testing similar models on different devices? Training a model for one hour on different devices might yield different insights
- I believe many aspects, including the combination of design choices, remain unexplored in this study. Clearly specifying which aspects or potential experiment cases are covered and which are not would help readers better understand the study’s scope

**Questions:**

Including some curiosity in weakness, I also raise the following questions.

- Appendix A.1 shows different edge devices. I suggest the authors include and compare the computing resources, not only RAM, of the devices to give readers a better understanding of the range of resources considered in the study
- In Table 3, the data samples per hour measured by the A10 GPU are illustrated. However, as stated earlier, processors like CPUs, GPUs, and mobile-targeted SoCs show significantly different computation times. Is the focus on computing resources solely on memory?
- RAG and LoRA are quite different techniques, with one altering the model structure and the other not. I also think they serve different purposes. Can they be considered as alternatives to one another?
- (minor) Rather than “performance,” please specify the exact metrics measured
- Figure 9 shows five models, not 6, as mentioned in the caption.
- The results suggest that insights vary significantly based on task type or complexity. Is there a way to categorize or quantify task characteristics or difficulty before running the tasks?

---

> ### Author Response · Authors · 2024-11-24
>
> We sincerely thank the reviewer for the constructive feedback and recognition of our work. The suggestions you provided are truly valuable and will help us further improve our research quality and presentation.
>
> We appreciate your interest in exploring combined techniques like RAG+PEFT. While this is definitely an interesting direction, there was a practical consideration behind our choice to study these methods separately. In real-world edge deployments, we often encounter scenarios where resource constraints make parameter updating impossible, forcing users to rely solely on RAG-based solutions. Additionally, these two methods serve fundamentally different purposes - RAG allows knowledge integration without touching the model parameters, while PEFT enables model adaptation through careful parameter updates. Understanding their individual characteristics helps users make informed choices based on their specific constraints and needs.
>
> Regarding the extensive use of appendices - we hear your concern about reading convenience, and we genuinely apologize for any difficulties this may have caused. We faced a challenging trade-off between providing comprehensive experimental evidence and maintaining a focused main narrative. While we chose to keep detailed results in the appendices for interested readers, we understand this could be better structured. Your feedback helps us recognize the need to better highlight key findings in the main sections while keeping supporting details accessible but not overwhelming.
>
> On the hardware evaluation front, we acknowledge that our primary focus on memory constraints could be expanded. Memory often represents the first bottleneck in edge deployment, which guided our initial focus. However, your suggestion about broader hardware metrics and cross-device comparisons opens up valuable directions for future investigation. This kind of comprehensive evaluation would indeed provide even more practical insights for the community.
>
> The ultimate goal of our work is to provide empirically-grounded guidance for deploying LLMs on edge devices, moving beyond intuitive assumptions to concrete evidence. We believe our extensive experimental results offer valuable reference points for practitioners while raising important questions for future research. Your thoughtful feedback has helped us identify areas where we can expand our investigation to make the work even more valuable to the community. We're excited to explore these directions in our future research.

---

### Official Review · Reviewer_n7jF · 2024-10-31

**Soundness:** 1
**Presentation:** 2
**Contribution:** 1
**Rating:** 3
**Confidence:** 4

**Summary:**

The paper studies LLM personalization at resource-constrained edge devices, investigating the effect of design choices (e.g. what model to use, what personalization technique to apply) on the performance (e.g. accuracy). This is done by running a set of experiments to observe the effect of each choices. According to these observations, a set of guidelines are proposed.

**Strengths:**

This is an important and timely topic. If done correctly, such study can provide practical guidelines for researchers and developer.

**Weaknesses:**

The study does not have the necessary depth. In particular, it reports only single-run experiments. It is however essential for an experimental study of this nature to perform multiple experiments per setting, enabling statistical comparison of different design choices. For instance, by providing means/medias and confidence intervals, one can assess if design choice A achieves a statically significant improvement over design choice B. I recommend the authors to follow approaches such as experimental design [1] to enhance the robustness of the study.

Besides, the paper fails to provide clear guidelines on how select the design choices. For example, the text below is from Section 3.1.:

“As task difficulty increases, such as with complex classification tasks and simple summarization tasks, the choice should gradually shift to RAG with the strongest model. Here, the strongest models are (quantized) LLMs that excel at general benchmarks and fit within the RAM constraint.”

What are specific criteria on deciding what is a complex classification task? Does it depend on the number of classes? Or on the task? Is it possible to provide some quantitative measures on what is a complex classification task? Also, what does mean gradually? What I get from this guideline is some general roles, but it does not help me to make a clear decision.

Finally, as I understand, the selected datasets for the fine-tuning process are available online, at Github. So, there is a possibility that the models which are studied in this paper have been already exposed to these datasets during the pre-training. This could change how we interpret the results, as fine-tuning a model over a subset of its training data is usually an easier task than fine-tuning over new (unseen) dataset. Please elaborate more on this aspect.

[1]. R. A. Fisher et al. The Design of Experiments. Number 5th ed. Oliver and Boyd, London and Edinburgh, 1949.

**Questions:**

Please see above.

---

### Official Review · Reviewer_1txd · 2024-11-01

**Soundness:** 2
**Presentation:** 2
**Contribution:** 1
**Rating:** 5
**Confidence:** 5

**Summary:**

This paper focuses on providing empirical guidance on LLM deployment on resource-constrained edge devices. The research focuses on how to optimize the design choices of LLMs in a resource-limited environment, balancing the computational demands with the capabilities of the devices.

**Strengths:**

1. Adequate experimental evaluation is carried out in this paper.
2. The topic of deploying LLM at the edge is interesting.

**Weaknesses:**

1. The paper primarily restates existing strategies for model deployment and optimization, lacking substantial innovation. The guidelines and strategies discussed, such as model compression, parameter-efficient fine-tuning (PEFT), and retrieval-augmented generation (RAG), are already well-documented methods in the machine learning field. The paper offers an empirical evaluation rather than a novel methodological contribution.

2. The experiments are largely confined to synthetic and benchmark datasets, which may not adequately represent the diversity of real-world scenarios where edge LLMs are deployed. This limits the applicability of the guidelines to practical use cases involving more dynamic environments.

**Questions:**

The paper primarily evaluates popular pre-trained LLMs like Llama and OPT, with various modifications. There is little exploration of alternative architectures that could be inherently better suited for edge deployment.

---

### Official Review · Reviewer_PkWA · 2024-11-03

**Soundness:** 3
**Presentation:** 3
**Contribution:** 2
**Rating:** 5
**Confidence:** 3

**Summary:**

The authors conducted extensive experiments and benchmarking to provide empirical guidelines for deploying large language models (LLMs) on edge devices, with a focus on fine-tuning models in private settings. However, the final conclusions of the paper align largely with common sense, and in some areas, the study lacks novelty.

**Strengths:**

This study is very comprehensive, employing a wide range of models and constructing various scenarios and methods for comparison.

**Weaknesses:**

Although this paper provides an empirical guideline through extensive experimentation, many of the conclusions are quite intuitive, lacking some innovative findings.

**Questions:**

1.	The remark in Section 3.1 suggests that increasingly complex tasks require stronger models. This is a commonly understood point and lacks novelty.
2.	In Section 3.1, it is suggested that RAG is more suitable for tasks of moderate difficulty. However, according to Figure 2 and Table 1, there is no significant difference between LoRA and RAG in terms of performance.
3.	Five fine-tuning methods are compared in Section 3.1, but the first three are weaker than LoRA and RAG, offering limited practical insight.
4.	Section 3.2 suggests that smaller values for rank and alpha are more suitable for resource-constrained environments, but this finding also lacks innovation. Additionally, the models discussed are relatively small, making them inherently more compatible with LoRA in limited-resource scenarios, which somewhat disconnects the findings from real-world edge limitations.
5.	The discussion on training duration in Section 3.2 does not specify which type of device is being considered. In the Appendix, the devices listed range from 4GB to 16GB of RAM, which would result in significantly different feasible training times.
6.	Section 3.4 proposes using only a limited amount of historical data in RAG, yet given the privatized edge LLM scenario suggested by the authors, it is realistic that users would only have access to a finite amount of data rather than unlimited data.
7.	The performance loss due to model compression is a well-known trade-off rather than a novel finding specific to edge LLMs.
8.	Although the paper is framed as addressing the deployment of edge LLMs, it mainly focuses on the private fine-tuning of models. Important aspects of deployment, such as inference, are not covered.

---

### Note · Authors · 2024-12-03

**Comment:**

We would like to thank the reviewers for their time and effort in reviewing our paper. In this work, we systematically studied LLM deployment on edge devices and provided guidelines and insights to advance research in this field. However, given the certain differences between our perspective and the reviewers' feedback, we have carefully decided to withdraw our paper.

**Withdrawal Confirmation:**

I have read and agree with the venue's withdrawal policy on behalf of myself and my co-authors.